# ProteinZero: Self-Improving Protein Generation via Online Reinforcement Learning

## Abstract

Protein generative models have shown remarkable promise in protein design, yet their success rates remain constrained by reliance on curated sequence-structure datasets and by misalignment between supervised objectives and real design goals. We present ProteinZero, an online reinforcement learning framework for inverse folding models that enables scalable, automated, and continuous self-improvement with computationally efficient feedback. ProteinZero employs a reward pipeline that combines structural guidance from ESMFold with a novel self-derived ddG predictor, providing stable multi-objective signals while avoiding the prohibitive cost of physics-based methods. To ensure robustness in online RL, we further introduce a novel embedding-level diversity regularizer that mitigates mode collapse and promotes sequence-level diversity among generated designs. Within a general RL formulation balancing multi-reward optimization, KL-divergence from a reference model, and diversity regularization, ProteinZero achieves robust improvements across designability, predicted stability, recovery, and diversity. On the CATH-4.3 benchmark, it consistently outperforms state-of-the-art baselines including ProteinMPNN, ESM-IF, and InstructPLM, reducing design failure rates by 36-48% and achieving success rates above 90% across diverse folds. Importantly, a complete RL run can be executed on a single 8×GPU node within three days, including reward computation and data generation. These results indicate that efficient online RL fine-tuning can complement supervised pretraining by allowing protein generative models to evolve continuously from their own outputs and optimize multiple design objectives without labeled data, opening new possibilities for exploring the vast protein design space. Sample designed sequences are provided in the supplementary material, and full source code and model checkpoints will be released upon publication.

## 1 Introduction

Protein design and engineering represent one of the most promising frontiers in computational biology, with applications spanning drug discovery to novel enzymes (Dauparas et al., 2022; Hsu et al., 2022; Wang et al., 2023a). A central challenge is protein inverse folding: generating amino acid sequences that fold into desired three-dimensional structures (Jing et al., 2021; Zhang & Skolnick, 2005), serving as the foundation for fixed backbone sequence design. This task is distinct from de novo structure generation or sequence-structure co-design, as it optimizes sequences for predetermined backbone geometries: the dominant setting in enzyme engineering, therapeutic scaffold optimization, and structure-based drug design. This task is crucial as protein backbone structure and side-chain conformation jointly determine functionalities like binding and catalytic interactions. However, optimizing functional properties requires first establishing high designability (designed sequences correctly folding into target structures) and thermodynamic stability (free energy difference favoring folded over unfolded states) as foundational prerequisites. Rocklin et al. (2017) demonstrated that 70-80% of computationally designed proteins fail due to misfolding or instability, with failures persisting in state-of-the-art AI methods (Bennett et al., 2023; Tsuboyama et al., 2023). Moreover, tiny (≈1-2 Å) atomic shifts at binding interfaces can disrupt hydrogen-bond geometry and packing, causing large affinity and specificity losses (failure to distinguish intended from off-target binders) (Clackson & Wells, 1995; Bogan & Thorn, 1998; Bajusz et al., 2021).

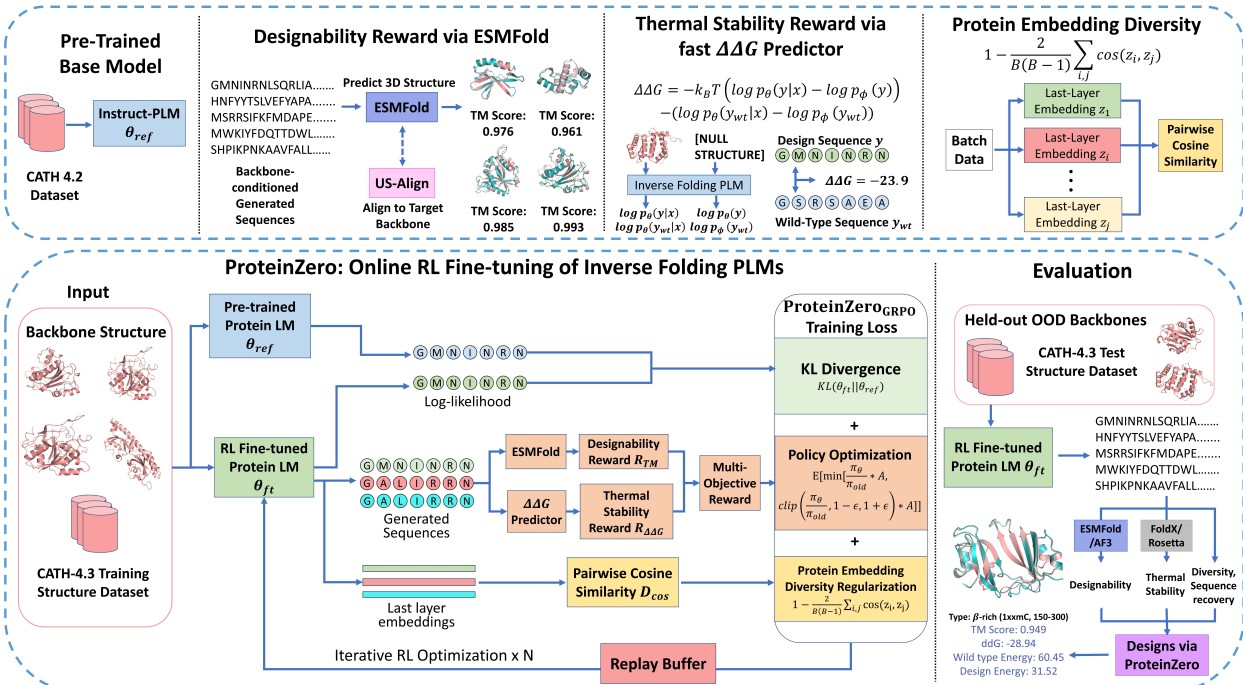

Figure 1: **ProteinZero framework. Upper:** Online RL components: ESMFold-based designability (TM-score via US-Align), $\Delta\Delta G$ predictor using backbone-conditioned likelihoods, and embedding diversity regularization. **Lower:** Iterative training where inverse folding models generate sequences, receive multi-objective rewards, and update with KL constraints and diversity regularization. Held-out CATH-4.3 evaluation demonstrates substantial improvements across all key design metrics.

Recent deep learning breakthroughs including ProteinMPNN (Dauparas et al., 2022), ESM-IF (Hsu et al., 2022), and graph-based methods (Jing et al., 2021; Wang et al., 2023a) have significantly improved inverse folding accuracy. However, these methods train on paired sequence-structure data from the Protein Data Bank (PDB) which, while valuable, represent a minuscule fraction of the protein sequence space (Dauparas et al., 2022; Hsu et al., 2022; Qiu et al., 2024) and exhibit limited diversity and natural biases. This data scarcity creates a ceiling for model performance and restricts exploration of novel protein designs beyond known natural and synthetic settings (Fujimoto & Gu, 2021; Shumailov et al., 2024). Moreover, there is a misalignment between the supervised learning task of inverse folding and actual objectives in real-world protein design, where applications require high designability, thermal stability, and sequence diversity (providing numerous reliable candidates for experimental validation rather than converging to known patterns) (Watson et al., 2023; Ingraham et al., 2023). Existing alignment efforts have focused on RL-finetuning structural generative models (Campbell et al., 2024; Huguet et al., 2024; Zhou et al., 2024; Gasser et al., 2024; Park et al., 2024), achieving only single- or few-round alignment with curated offline datasets, limiting exploration to known successes rather than discovering novel design principles through iterative feedback.

We propose ProteinZero, an online RL fine-tuning framework that addresses multi-objective optimization challenges in protein design, enabling automated self-improvement of inverse folding models while balancing designability, stability, and diversity. Our contributions are:

1. We present ProteinZero, achieving stable multi-round self-improvement in protein sequence design through continuous exploration without curated preference datasets.

2. We introduce a self-derived $\Delta\Delta G$ estimator computed from the inverse folding model using backbone-conditioned likelihoods normalized by unconditional priors. Combined with ESMFold-

based designability rewards, this enables computationally tractable multi-objective online RL optimization (see Table 4).

3. We develop a novel diversity regularizer operating in protein embedding space rather than sequence space, preventing mode collapse (Shumailov et al., 2024; Alemohammad et al., 2024; Holtzman et al., 2020) while preserving designability and predicted stability.

4. We elucidate the design space of RL fine-tuning by examining algorithms (GRPO, RAFT, DPO, multi-round DPO), rewards, and regularization strategies, identifying optimal configurations for stable multi-objective optimization without mode collapse.

5. Extensive experiments demonstrate that ProteinZero outperforms existing methods across all key metrics, achieving 36-48% reduction in design failure rates versus ProteinMPNN (Dauparas et al., 2022), ESM-IF (Hsu et al., 2022), and InstructPLM (Qiu et al., 2024), with significant improvements in structural accuracy, predicted stability, and diversity across diverse protein folds including challenging long chains, validated across multiple independent computational oracles (ESMFold, AlphaFold3, FoldX, Rosetta) to assess whether improvements transfer consistently across independent in-silico evaluators.

## 2 Related Work

**Protein Inverse Folding Models.** Inverse folding generates amino acid sequences $y = (y_1, ..., y_L)$ for target structures $x$, formulated as conditional generation $p_\theta(y|x)$ with model parameters $\theta$ trained via supervised loss on PDB pairs. Ingraham et al. (2019) pioneered graph neural networks for this task, extended by ProteinMPNN (Dauparas et al., 2022) with noise-aware training. ESM-IF (Hsu et al., 2022) leveraged pretrained language models, while GVP-GNN (Jing et al., 2021), StructTrans (Wang et al., 2023a), Pi-Fold (Gao et al., 2023), and GraDe-IF (Yi et al., 2023) introduced geometric representations, transformers, co-design, and diffusion respectively. InstructPLM (Qiu et al., 2024) achieved SOTA by adapting frozen language models via structural prompts (our base architecture). While achieving strong benchmarks, supervised approaches face inherent constraints: limited PDB datasets restrict exploration of the vast sequence space, and their objectives, optimizing sequence recovery, may not align with real design goals of maximizing stability, designability, and diversity. We extend these foundations through online RL with efficient proxy rewards, enabling continuous learning from self-generated sequences to directly optimize these multiple design objectives.

**RLHF of Protein Generative Models.** Classical RL approaches to biological sequence design (Angermueller et al., 2020; Runge et al., 2019) train task-specific policies from scratch via unconditional generation or local mutations, a different paradigm detailed in Appendix D.1. With powerful pre-trained protein models, Reinforcement Learning from Human Feedback (RLHF) has emerged for fine-tuning generative models. RLHF transforms models through online methods like PPO (Schulman et al., 2017), GRPO (Shao et al., 2024), and RAFT (Dong et al., 2023) that optimize rewards directly, and offline methods like DPO (Rafailov et al., 2023) using static preference datasets. While successful in LLMs, applying online RLHF to protein models faces both computational (Table 4) and reward modeling infrastructure challenges. Current protein RLHF work encompasses diverse architectures: Campbell et al. (2024) and Huguet et al. (2024) enhance structural generation with ReFT, Zhou et al. (2024) applies DPO for antibody design, Xu et al. (2025) employ multi-round DPO for inverse folding with structural feedback, ResiDPO implements residue-level DPO with pLDDT scores (Xue et al., 2025), and Wang et al. (2025) introduces online fine-tuning for discrete diffusion sequence models through direct reward backpropagation via Gumbel-Softmax approximations, which requires differentiable rewards. These methods primarily address structure generation or co-design tasks, with most operating offline. Offline approaches rely on pre-collected rewards without iterative learning from self-generated sequences, limiting exploration of protein design space. We introduce online RL for inverse folding models using policy gradients with non-differentiable reward proxies, enabling self-improvement in designability, stability, and diversity.

**Diversity Regularization.** Promoting diversity in protein generative models is crucial for increasing downstream success rates and maintaining exploration capability in online RL to avoid mode collapse and reward

hacking (Ouyang et al., 2022; Fan et al., 2025; Shumailov et al., 2024) (see Appendix D.2). Prior work explored sequence-level metrics: Park et al. (2024) employ Hamming distance as diversity regularizer, operating on raw sequences instead of structure-aware representations. The DPO-based approach faces challenges in simultaneously optimizing rewards and diversity. We introduce embedding-level diversity regularization that operates in the model's embedding space, promoting sequence-level diversity while preventing mode collapse and maintaining structural coherence (theoretical derivation in Appendix F, empirical dynamics in Appendix C.3).

## 3 Method

We propose ProteinZero, a framework that fine-tunes protein generative models through online reinforcement learning. Our approach optimizes a reward-based objective $\mathcal{J}_{\mathrm{RL}}(\theta) = \mathbb{E}_{x \sim \mathcal{D}_x, y \sim p_\theta(\cdot|x)}[r(x,y)] - \alpha_{\mathrm{KL}} \cdot \mathrm{KL}(p_\theta(\cdot \mid x) \| p_{\mathrm{ref}}(\cdot \mid x))$, where $r(x,y)$ combines multiple design objectives including designability and stability, $p_{\mathrm{ref}}$ is a reference model (typically the pre-trained model), and $\alpha_{\mathrm{KL}}$ controls divergence from the reference.

### 3.1 ProteinZero Framework: Addressing Mode Collapse in Online RL for Proteins

To realize this objective while preventing mode collapse, ProteinZero couples reward optimization with novel diversity constraints, enabling stable and effective online learning. The framework enables continuous exploration beyond pre-collected datasets, discovering novel design principles through iterative feedback.

Reinforcement learning for protein design requires optimizing a model to generate sequences that maximize a reward function while maintaining reasonable proximity to a reference model. In practice, recent RL fine-tuning methods can be unified in this general objective through specialized algorithms that balance exploitation and exploration: $\mathcal{L}(\theta) = \mathcal{L}_{\mathrm{RL}}(\theta) + \mathcal{L}_{\mathrm{KL}}(\theta)$. For instance, in the Group Relative Policy Optimization (GRPO) algorithm, this objective is realized as $\mathcal{L}_{\mathrm{GRPO}}(\theta) = -\mathbb{E}_{(x,y) \sim \mathcal{B}} \left[ \min(A^* \frac{p_\theta}{p_{\theta_{\mathrm{old}}}}, A^* \mathrm{clip}(\frac{p_\theta}{p_{\theta_{\mathrm{old}}}}, 1\text{-}\epsilon, 1+\epsilon)) \right] + \alpha_{\mathrm{KL}} \cdot \mathrm{KL}(p_\theta \| p_{\mathrm{ref}})$ (Shao et al., 2024), where $A$ is the advantage function, $p_{\theta_{old}}$ is learned policy of last iteration. However, we observed that protein generative models suffer from mode collapse in online RL fine-tuning, converging to a narrow set of solutions that maximize rewards without diversity (see Appendix C.3). Thus, we incorporate a diversity regularization term, resulting in a more comprehensive objective:

$$\mathcal{L}(\theta) = \mathcal{L}_{\mathrm{RL}}(\theta) + \mathcal{L}_{\mathrm{KL}}(\theta) + \mathcal{L}_{\mathrm{Div}}(\theta) \tag{1}$$

While diversity can be promoted by incorporating it directly into the reward, our experiments show this often causes training instability and performance degradation (see Table 3). Thus, ProteinZero applies diversity regularization at the representation level through a separate loss $\mathcal{L}_{\mathrm{Div}}(\theta)$, encouraging diversity while preserving the integrity of the main reward optimization (Table 1 and Figure 6).

To enable practical online RL fine-tuning, we address two critical challenges: (1) the lack of effective diversity regularization for protein models, and (2) the prohibitive computational cost of reward evaluation, which can extend training to months. We therefore propose embedding-level diversity regularization and fast reward modeling to make online fine-tuning practically achievable.

#### 3.1.1 Embedding-Level Diversity Regularization for Mode Collapse Mitigation

To address mode collapse in protein generative models during online RL fine-tuning, we propose a novel diversity regularization operating at the protein embedding level. Unlike token-level diversity metrics which can compromise functional properties, our approach leverages learned representations shown to encode hierarchical biological information from local patterns to functional domains (Simon & Zou, 2024), with embedding distances reflecting functional relationships (Schmirler et al., 2024; Corso et al., 2021; Blaabjerg et al., 2024). For each protein sequence in a batch, we compute a fixed-dimensional embedding vector by aggregating the

last-layer decoder activations:

$$z_i(\theta) = \frac{\sum_t m_{i,t} h_{i,t}}{\sum_t m_{i,t}} \in \mathbb{R}^d, \quad 1 \le i \le B \tag{2}$$

where $h_{i,t} \in \mathbb{R}^d$ is the decoder activation at position $t$ for sample $i$, and $m_{i,t} \in \{0,1\}$ an attention mask. These protein embeddings are $\ell_2$-normalized before computing a cosine-based diversity score: $D_{\cos}(\theta; \mathcal{B}) = 1 - \overline{\cos} \in [0,1]$, where $\overline{\cos} = \frac{2}{B(B-1)} \sum_{1 \le i < j \le B} \cos(z_i, z_j)$. The diversity regularization term is incorporated as:

$$\mathcal{L}_{\mathrm{Div}}(\theta) = -\alpha_{\mathrm{div}} \cdot D_{\cos}(\theta; \mathcal{B}) \tag{3}$$

Since $z_i$ depends on $\theta$, this provides informative gradients that foster the generation of diverse candidate sequences that retain high designability and predicted stability. Our theoretical analysis (Appendix F) demonstrates how this embedding-based approach mitigates mode collapse in online RL, a contribution applicable beyond protein design. Note that while we optimize embedding-level diversity during training, our evaluation employs standard Hamming distance between sequences to provide an orthogonal assessment of sequence-level diversity. The embedding-level formulation achieves diversity preservation and training stability, validated in ablation studies (Table 3) and training dynamics (Appendix C.3).

### 3.1.2 Fast Proxy Rewards: Enabling Practical Online RL Training

AlphaFold's MSA and template searches and FoldX's physics calculations (Table 4) require minutes to hours per protein, making online RL infeasible. We address this with two fast proxy rewards:

**Designability Reward:** We use ESMFold (Hsu et al., 2022) for structural inference, leveraging its alignment-free, single-pass architecture instead of AlphaFold2/3's MSA searches and recycling steps (Jumper et al., 2021; Abramson et al., 2024). Our designability reward $r_{\mathrm{TM}}(x,y)$ specifically uses the TM-score from US-Align Zhang et al. (2022), an updated version of TM-Align (Zhang & Skolnick, 2005), computed between ESMFold-predicted and target structures, explicitly not ESMFold's internal confidence score pTM, ensuring our optimization targets actual structural alignment through length-normalized distance-weighted $C_\alpha$ overlaps, not prediction confidence.

**Thermal Stability Reward:** We propose a novel thermal stability reward $r_{\Delta\Delta G}(x,y)$, serving as a backbone-specific folding-energy surrogate for single-chain proteins, referenced to the PDB wild-type. Because our monomeric setting lacks an inter-chain interface, the unbound-state term required by the Boltzmann-aligned estimator (BA-DDG) (Jiao et al., 2025) is unevaluable. Instead, drawing on evidence that backbone-conditioned likelihoods reflect folding stability (Shanker et al., 2024; Widatalla et al., 2024; Cagiada et al., 2025; Zheng et al., 2023; Ingraham et al., 2019), we normalize this likelihood with an unconditional sequence prior and anchor it to the wild-type baseline:

$$\Delta\Delta G(x,y) = -k_B T[(\log p_\theta(y \mid x) - \log p_\varphi(y)) - (\log p_\theta(y_{\mathrm{wt}} \mid x) - \log p_\varphi(y_{\mathrm{wt}}))], \tag{4}$$

where $p_\theta(y \mid x)$ is the backbone-conditioned inverse-folding likelihood, $p_\varphi(\cdot)$ the unconditional sequence prior, $y_{\mathrm{wt}}$ the PDB wild-type sequence, and $k_B T$ the thermal energy at 298 K ($0.593\,\mathrm{kcal\,mol^{-1}}$). The prior $p_\varphi(\cdot)$ is obtained by running the same inverse-folding network (e.g., ProteinMPNN or InstructPLM) with coordinate channels masked, converting it into a sequence-only language model capturing residue-frequency and chain-length distributions of proteins. Subtracting $\log p_\varphi(y)$ from $\log p_\theta(y \mid x)$ removes background amino-acid composition and chain-length preferences, isolating backbone-specific excess compatibility of candidate sequence $y$. Hence, using $y_{\mathrm{wt}}$ as reference yields a computationally efficient $\Delta\Delta G$ surrogate for monomeric stability optimization, with the likelihood ratio isolating backbone-specific stabilization that correlates with experimental measurements (PCC = 0.60–0.62 on Ssym; Section C.4).

**Multi-objective reward:** Our final reward combines both scores after min-max normalization to balance scale differences. Normalization is performed across the candidate pool of inverse folding sequences generated for the same backbone within each reinforcement learning iteration: $\tilde{r}_{\mathrm{TM}} = (r_{\mathrm{TM}} - r_{\mathrm{TM}}^{\min})/(r_{\mathrm{TM}}^{\max} - r_{\mathrm{TM}}^{\min})$ and $\tilde{r}_{\Delta\Delta G}$ analogously, giving $r(x,y) = \lambda_{\mathrm{TM}} \tilde{r}_{\mathrm{TM}}(x,y) + \lambda_{\Delta\Delta G} \tilde{r}_{\Delta\Delta G}(x,y)$. This reward accelerates evaluation 25-100× depending on protein length (Table 4), reducing training time from months to days. Our experiments show substantial predicted thermodynamic stability improvements with high structural fidelity (see Figure 6 and Table 1).

## 3.2 ProteinZero Algorithms: Diversity-Regularized RAFT and GRPO

Building upon our general framework, we implement two online RL algorithms for fine-tuning inverse folding models: RAFT and GRPO. We adapt both methods to incorporate our dual-objective reward system, designability scores from ESMFold structures evaluated by US-Align, and self-derived $\Delta\Delta G$ for thermodynamic stability, alongside embedding-level diversity regularization. These adaptations enable different optimization strategies (detailed in Sections 3.2.1 and 3.2.2).

### 3.2.1 ProteinZero_RAFT: Reward-ranked Fine-tuning with Embedding Diversity

RAFT (Dong et al., 2023) transforms RL into a supervised learning problem by iteratively filtering model outputs based on rewards. Our adaptation generates multiple candidate sequences per target structure, evaluates them with our efficient reward, and retains only the best to form a filtered dataset. Unlike conventional RAFT that incorporates KL-divergence into the reward, we separate the KL term and add our embedding-based diversity regularization ($\mathcal{L}_{\text{CE}}$ is the cross entropy loss):

$$\mathcal{L}_{\text{ProteinZero}_{\text{RAFT}}}(\theta) = \mathcal{L}_{\text{CE}}\left(\theta; \mathcal{D}_{\text{filtered}}\right) + \alpha_{\text{KL}} \cdot \text{KL}\left(p_\theta \| p_{\text{ref}}\right) - \alpha_{\text{div}} \cdot D_{\cos}\left(\theta; \mathcal{D}_{\text{filtered}}\right) \tag{5}$$

### 3.2.2 ProteinZero_GRPO: Embedding-Diversified Policy Optimization

GRPO (Shao et al., 2024) directly optimizes the policy via a trust-region objective:

$$\mathcal{J}_{\text{GRPO}}(\theta) = \mathbb{E}_{x \sim P(X), \{y_i\}_{i=1}^{G} \sim \pi_{\theta_{\text{old}}}(Y|x)} \frac{1}{G} \sum_{i=1}^{G} \frac{1}{|y_i|} \sum_{t=1}^{|y_i|} \min\left[ \frac{\pi_\theta\left(y_{i,t} \mid x, y_{i,<t}\right)}{\pi_{\theta_{\text{old}}}\left(y_{i,t} \mid x, y_{i,<t}\right)} \hat{A}_{i,t}, \right.$$
$$\left. \text{clip}\left( \frac{\pi_\theta\left(y_{i,t} \mid x, y_{i,<t}\right)}{\pi_{\theta_{\text{old}}}\left(y_{i,t} \mid x, y_{i,<t}\right)}, 1 - \varepsilon, 1 + \varepsilon \right) \hat{A}_{i,t} \right] - \beta \mathbb{D}_{KL}\left[\pi_\theta \| \pi_{ref}\right], \tag{6}$$

where $\varepsilon$ and $\beta$ are hyperparameters, and $\hat{A}_{i,t}$ is the advantage calculated from relative rewards within each group. The group relative advantage calculation aligns well with our reward models. Unlike methods that add KL penalty to rewards, GRPO directly adds KL divergence to the loss. We extend this formulation by incorporating our embedding-level diversity regularization ($\mathcal{L}_{\text{GRPO}} = -\mathcal{J}_{\text{GRPO}}$):

$$\mathcal{L}_{\text{ProteinZero}_{\text{GRPO}}}(\theta) = \mathcal{L}_{\text{GRPO}}(\theta) - \alpha_{\text{div}} \cdot D_{\cos}(\theta; \mathcal{B}) \tag{7}$$

Both algorithms effectively implement our ProteinZero framework but approach optimization differently. Our experiments demonstrate that both methods significantly outperform baselines, with ProteinZero_GRPO consistently achieving superior performance across evaluated metrics.

## 4 Experiment

### 4.1 Experimental Setup

**Evaluation.** We evaluated ProteinZero on CATH-4.3 (Orengo et al., 1997), maintaining the train-test-validation split from Hsu et al. (2022). Our test set excluded structures with $> 40\%$ sequence identity to training proteins of 0-150 residues and $> 30\%$ identity to proteins of 150-300 residues, enabling assessment of out-of-distribution generalization. We trained and evaluated models separately for each length category (0-150, 150-300). We evaluate the models with a comprehensive set of metrics, including designability metrics measured by both ESMFold and AF3 (TM Score, PLDDT, scRMSD), stability measured with our fast-ddG predictor and physics-based FoldX/Rosetta ddG (Schymkowitz et al., 2005)), sequence recovery, and sequence Diversity — see Appendix B.3 for detailed definitions. Overall Success Rate was defined as achieving both scRMSD $< 2\,\text{Å}$ and FoldX ddG $< 0$, inspired by Wang et al. (2025).

**ProteinZero implementation.** We perform online RL from the publicly released InstructPLM checkpoint, which consists of a ProGen2-xlarge decoder (32 transformer blocks, hidden size 4096, 16 attention

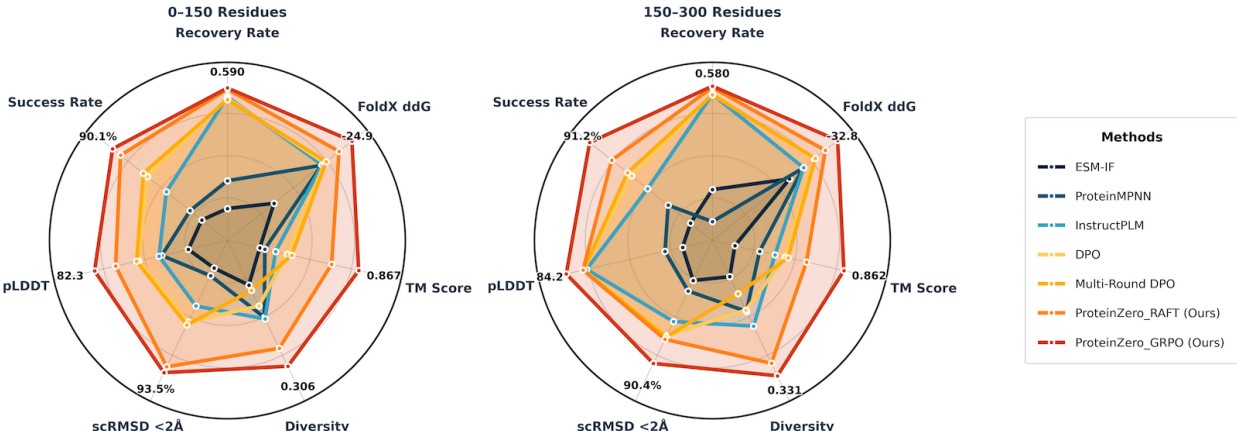

Figure 2: Performance comparison across seven evaluation metrics for 0–150 residue proteins (left) and 150–300 residue proteins (right). ProteinZero uses TM Score and predicted $\Delta\Delta G$ as rewards, which measure designability and thermal stability, and additionally promotes diversity through the embedding-level regularizer in the loss. ProteinZero$_{\text{GRPO}}$ and ProteinZero$_{\text{RAFT}}$ lead on the closely related success-defining metrics, scRMSD < 2Å% and FoldX ddG, while also leading on pLDDT and Recovery Rate.

heads, approximately 6.57B parameters) conditioned on protein structure through a ProteinMPNN-based structural encoder. Per-residue structural embeddings of dimension 1152 are mapped by a learned structural projection module into 256 prefix embeddings of dimension 4096, which are prepended to the decoder's token embeddings. During online RL, all base-model parameters are frozen and only LoRA adapters are trained, applied to the self-attention and feed-forward projections of every decoder block as well as the structural projection module, yielding approximately 33.8M trainable parameters (0.51% of the base model). We implement two algorithms: ProteinZero$_{\text{RAFT}}$, which selects the best-rewarded sequences for policy updates, and ProteinZero$_{\text{GRPO}}$, which directly optimizes the policy using group-relative rewards. For ProteinZero$_{\text{GRPO}}$, we adopt the GRPO update rule from the HuggingFace TRL library (von Werra et al., 2020) and implement the remaining components of the online RL pipeline, including the multi-objective reward computation (Section 3.1.2), the min-max combination of the multi-objective rewards, the embedding-level diversity regularizer (Section 3.1.1), and the rollout procedure for inverse folding under the formulation described in Section 3. ProteinZero$_{\text{RAFT}}$ is implemented from scratch, strictly following the original RAFT formulation (Dong et al., 2023). Both methods employ embedding-level diversity regularization ($\alpha_{\text{div}} = 0.05$) and KL constraints ($\alpha_{\text{KL}} = 0.1$) against a frozen reference policy. All experiments use 8 NVIDIA A100 GPUs. Per-algorithm hyperparameters and sampling settings for ProteinZero$_{\text{GRPO}}$ and ProteinZero$_{\text{RAFT}}$ are provided in Appendix B, together with ablations over the KL and diversity coefficients.

**Baselines.** We compared against state-of-the-art inverse folding models (ProteinMPNN (Dauparas et al., 2022), ESM-IF (Hsu et al., 2022), InstructPLM (Qiu et al., 2024)). For RL-finetuniung algorithms, we compare with widely used offline RL baselines including DPO (Rafailov et al., 2023) and multi-round DPO (Xu et al., 2025).

## 4.2 Main Results

**Overall Performance Analysis.** Table 1 shows ProteinZero consistently outperforms existing methods. Both ProteinZero$_{\text{GRPO}}$ and ProteinZero$_{\text{RAFT}}$ surpass all baselines, with ProteinZero$_{\text{GRPO}}$ achieving best results across metrics (Figure 2). Our approach balances sequence recovery, structural accuracy, and stability while learning from self-generated outputs without additional labels. Importantly, although we only use TM-score (ESMFold/US-Align) and self-derived $\Delta\Delta G$ as rewards (Section 3.1.2), our evaluation uses orthogonal metrics, FoldX ddG for stability, pLDDT/scRMSD for structure, recovery/diversity for sequences, suggesting the gains are not solely attributable to reward hacking. Independent AlphaFold3 evaluation also shows these improvements transfer across computational structure predictors (Table 2), with Figure 4

Table 1: Comparison of protein sequence design methods for 0-150 and 150-300 residue proteins. Success Rate is defined as scRMSD < 2Å and FoldX ddG < 0. Best scores are highlighted in blue, second-best in green. Designability metrics computed by ESMFold (independent AF3 evaluations confirm the same trend, see Table 2). All results are mean ± s.e. over 10 independent runs.

| Length | Method | InverseFold Acc. Recovery Rate ↑ | Thermal Stability Metrics Fast-ddG ↓ | FoldX ddG ↓ | TM Score ↑ | PLDDT ↑ | Diversity ↑ | scRMSD ↓ (scRMSD <2Å% ↑) | Overall Success (%) ↑ |
|---|---|---|---|---|---|---|---|---|---|
| | *Base Model* | | | | | | | | |
| 0-150 residues | InstructPLM | $0.574_{\pm0.009}$ | $-21.543_{\pm1.330}$ | $-20.878_{\pm1.445}$ | $0.812_{\pm0.011}$ | $79.983_{\pm0.614}$ | $0.281_{\pm0.007}$ | $1.484_{\pm0.044}$ ($85.71\%_{\pm0.2\%}$) | $84.45\%_{\pm0.02\%}$ |
| | *SOTA Inverse Folding Models* | | | | | | | | |
| | ProteinMPNN | $0.426_{\pm0.006}$ | $-21.509_{\pm1.230}$ | $-20.792_{\pm1.207}$ | $0.805_{\pm0.009}$ | $79.883_{\pm0.502}$ | $0.280_{\pm0.005}$ | $1.500_{\pm0.037}$ ($82.14\%_{\pm0.2\%}$) | $81.95\%_{\pm0.02\%}$ |
| | ESM-IF | $0.377_{\pm0.006}$ | $-17.900_{\pm1.235}$ | $-14.328_{\pm1.269}$ | $0.802_{\pm0.009}$ | $78.918_{\pm0.534}$ | $0.263_{\pm0.005}$ | $1.515_{\pm0.038}$ ($81.25\%_{\pm0.2\%}$) | $80.71\%_{\pm0.02\%}$ |
| | *RL Baseline Method* | | | | | | | | |
| | DPO | $0.571_{\pm0.008}$ | $-21.713_{\pm1.260}$ | $-21.191_{\pm1.332}$ | $0.820_{\pm0.010}$ | $80.716_{\pm0.571}$ | $0.274_{\pm0.005}$ | $1.473_{\pm0.041}$ ($87.58\%_{\pm0.2\%}$) | $86.44\%_{\pm0.02\%}$ |
| | Multi-Round DPO | $0.569_{\pm0.008}$ | $-21.797_{\pm1.312}$ | $-21.423_{\pm1.398}$ | $0.823_{\pm0.011}$ | $80.797_{\pm0.585}$ | $0.266_{\pm0.005}$ | $1.468_{\pm0.043}$ ($87.95\%_{\pm0.2\%}$) | $86.89\%_{\pm0.03\%}$ |
| | *Our Online RL Methods* | | | | | | | | |
| | ProteinZero$_{RAFT}$ (Ours) | $0.587_{\pm0.008}$ | $-22.236_{\pm1.272}$ | $-23.168_{\pm1.356}$ | $0.849_{\pm0.011}$ | $81.560_{\pm0.613}$ | $0.296_{\pm0.007}$ | $1.393_{\pm0.044}$ ($92.86\%_{\pm0.3\%}$) | $89.29\%_{\pm0.02\%}$ |
| | ProteinZero$_{GRPO}$ (Ours) | $0.590_{\pm0.008}$ | $-22.616_{\pm1.327}$ | $-24.924_{\pm1.382}$ | $0.867_{\pm0.011}$ | $82.326_{\pm0.612}$ | $0.306_{\pm0.007}$ | $1.373_{\pm0.044}$ ($93.55\%_{\pm0.3\%}$) | $90.13\%_{\pm0.02\%}$ |
| | *Base Model* | | | | | | | | |
| 150-300 residues | InstructPLM | $0.570_{\pm0.009}$ | $-36.362_{\pm2.451}$ | $-27.145_{\pm1.797}$ | $0.824_{\pm0.014}$ | $83.783_{\pm0.568}$ | $0.305_{\pm0.008}$ | $1.448_{\pm0.048}$ ($88.24\%_{\pm0.2\%}$) | $86.38\%_{\pm0.02\%}$ |
| | *SOTA Inverse Folding Models* | | | | | | | | |
| | ProteinMPNN | $0.405_{\pm0.007}$ | $-35.778_{\pm2.280}$ | $-27.057_{\pm1.581}$ | $0.816_{\pm0.012}$ | $82.361_{\pm0.548}$ | $0.297_{\pm0.006}$ | $1.469_{\pm0.040}$ ($86.64\%_{\pm0.2\%}$) | $84.67\%_{\pm0.02\%}$ |
| | ESM-IF | $0.446_{\pm0.008}$ | $-32.125_{\pm2.207}$ | $-24.816_{\pm1.548}$ | $0.802_{\pm0.013}$ | $82.042_{\pm0.536}$ | $0.279_{\pm0.006}$ | $1.487_{\pm0.042}$ ($86.09\%_{\pm0.2\%}$) | $82.81\%_{\pm0.02\%}$ |
| | *RL Baseline Method* | | | | | | | | |
| | DPO | $0.570_{\pm0.009}$ | $-36.417_{\pm2.325}$ | $-28.915_{\pm1.571}$ | $0.830_{\pm0.013}$ | $83.837_{\pm0.506}$ | $0.296_{\pm0.008}$ | $1.441_{\pm0.042}$ ($88.97\%_{\pm0.2\%}$) | $87.70\%_{\pm0.02\%}$ |
| | Multi-Round DPO | $0.569_{\pm0.009}$ | $-36.483_{\pm2.402}$ | $-29.087_{\pm1.612}$ | $0.831_{\pm0.014}$ | $83.840_{\pm0.519}$ | $0.288_{\pm0.008}$ | $1.437_{\pm0.044}$ ($89.04\%_{\pm0.3\%}$) | $88.05\%_{\pm0.02\%}$ |
| | *Our Online RL Methods* | | | | | | | | |
| | ProteinZero$_{RAFT}$ (Ours) | $0.578_{\pm0.009}$ | $-37.575_{\pm2.391}$ | $-30.755_{\pm1.661}$ | $0.841_{\pm0.013}$ | $83.850_{\pm0.542}$ | $0.324_{\pm0.008}$ | $1.427_{\pm0.046}$ ($89.17\%_{\pm0.2\%}$) | $89.36\%_{\pm0.02\%}$ |
| | ProteinZero$_{GRPO}$ (Ours) | $0.580_{\pm0.009}$ | $-40.626_{\pm2.422}$ | $-32.805_{\pm1.694}$ | $0.862_{\pm0.013}$ | $84.154_{\pm0.539}$ | $0.331_{\pm0.009}$ | $1.393_{\pm0.045}$ ($90.43\%_{\pm0.2\%}$) | $91.19\%_{\pm0.02\%}$ |

providing qualitative examples of representative complex protein architectures evaluated with AlphaFold3. This cross-oracle consistency, training on ESMFold/Fast-ddG while improving on AF3/FoldX, suggests the improvements are not merely predictor-specific artifacts; confirming that they reflect genuine biophysical principles would require prospective experimental validation. (Further per-backbone analyses against the InstructPLM base (AlphaFold3 refolding, BLOSUM62 substitutions, ESM-2 plausibility, biophysical descriptors, UniRef90 identity) are reported in Appendix C.) For example, ProteinZero$_{GRPO}$ achieves success rates 90.13% and 91.19% for 0-150 and 150-300 residues, respectively, reducing failure rates by 45% (from 18.05% to 9.87%) compared to ProteinMPNN for small proteins. Notably, compared to InstructPLM, we simultaneously improve recovery (0.574 → 0.590) and diversity (0.281 → 0.306), two traditionally conflicting objectives, demonstrating its ability to balance sequence conservation with exploration.

**Comparison with DPO-based fine-tuning.** We next compare ProteinZero with widely used DPO variants to illustrate the advantages of online RL. Regular DPO improves InstructPLM's success rate modestly (84.45% →86.44% for 0-150 residues), while Multi-Round DPO further raises it slightly to 86.89%. However, both variants reduce sequence diversity below the baseline: DPO lowers it from 0.281 to 0.274 and Multi-Round DPO further to 0.266. In contrast, ProteinZero$_{GRPO}$ reaches 90.13% success and enhances diversity to 0.306. This divergence reflects a broader trend: offline methods progressively converge toward narrower solution spaces, limiting exploration of novel sequences. Online RL with diversity regularization maintains an exploration-exploitation balance, yielding not only higher diversity but also better structural generalization, as seen in improved scRMSD (1.373Å vs. 1.473Å for DPO). Similar patterns hold for larger proteins, where Multi-Round DPO increases success rate only modestly (86.38% → 88.05%) but still reduces diversity to 0.288, whereas ProteinZero achieves both higher success (91.19%) and greater diversity (0.331).

**Comparison with SOTA Inverse Folding Models.** We further compare ProteinZero against state-of-the-art inverse folding models. Starting from InstructPLM (Qiu et al., 2024) as our base model, ProteinZero$_{GRPO}$ improves TM-score (0.812 → 0.867), predicted stability (FoldX ddG: –20.878 → –24.924 kcal/mol), diversity (0.281 → 0.306), and success rate (84.45% → 90.13%) for short proteins. Similar gains are observed for longer proteins, where success rate increases from 86.38% to 91.19% and predicted stability improves by 21% (–27.145 → –32.805 kcal/mol). Compared with other leading inverse folding models, ProteinZero achieves consistently higher success rates, outperforming ProteinMPNN (Dauparas et al., 2022) (81.95%) and ESM-IF (Hsu et al., 2022) (80.71%) across both size ranges. Qualitative visualizations (Fig-

Table 2: Independent validation of 0-150 and 150-300 residue proteins using AlphaFold3 versus ESMFold. Best scores are highlighted in  blue , second-best in  green .

| Length | Method | TM Score ↑ | | PLDDT ↑ | | scRMSD ↓ | | scRMSD <2Å (%) ↑ | | Success Rate (%) ↑ | |
|---|---|---|---|---|---|---|---|---|---|---|---|
| | | ESMFold | AF3 | ESMFold | AF3 | ESMFold | AF3 | ESMFold | AF3 | ESMFold | AF3 |
| | | *Base Model* | | | | | | | | | |
| | InstructPLM | 0.8121 | 0.8356 | 79.98 | 82.45 | 1.4842 | 1.4287 | 85.71 | 88.32 | 84.45 | 86.98 |
| | | *Offline RL Baselines* | | | | | | | | | |
| 0-150 residues | DPO | 0.8198 | 0.8401 | 80.72 | 82.93 | 1.4727 | 1.4218 | 87.58 | 89.43 | 86.44 | 88.12 |
| | Multi-Round DPO | 0.8228 | 0.8436 | 80.80 | 83.07 | 1.4678 | 1.4176 | 87.95 | 89.95 | 86.89 | 88.71 |
| | | *Our Online RL Methods* | | | | | | | | | |
| | ProteinZero$_{RAFT}$ | 0.8494 | 0.8612 | 81.56 | 83.48 | 1.3929 | 1.3587 | 92.86 | 93.89 | 89.29 | 90.42 |
| | ProteinZero$_{GRPO}$ | 0.8674 | 0.8798 | 82.33 | 84.09 | 1.3727 | 1.3406 | 93.55 | 94.67 | 90.13 | 91.56 |
| | | *Base Model* | | | | | | | | | |
| | InstructPLM | 0.8241 | 0.8418 | 83.78 | 85.86 | 1.4476 | 1.4018 | 88.24 | 90.12 | 86.38 | 88.41 |
| | | *Offline RL Baselines* | | | | | | | | | |
| 150-300 residues | DPO | 0.8296 | 0.8454 | 83.84 | 85.92 | 1.4407 | 1.3978 | 88.97 | 90.58 | 87.70 | 89.31 |
| | Multi-Round DPO | 0.8313 | 0.8467 | 83.84 | 85.94 | 1.4372 | 1.3953 | 89.04 | 90.67 | 88.05 | 89.56 |
| | | *Our Online RL Methods* | | | | | | | | | |
| | ProteinZero$_{RAFT}$ | 0.8413 | 0.8548 | 83.85 | 86.03 | 1.4271 | 1.3891 | 89.17 | 90.72 | 89.36 | 90.64 |
| | ProteinZero$_{GRPO}$ | 0.8617 | 0.8718 | 84.15 | 86.19 | 1.3925 | 1.3598 | 90.43 | 91.76 | 91.19 | 92.27 |

ure 6) further support these findings, highlighting ProteinZero's ability to generate stable designs with high structural fidelity.

**Effectiveness of fast-ddg reward.** ProteinZero$_{GRPO}$ achieves substantial gains in predicted thermostability compared to InstructPLM, improving FoldX ddG by 19% (from $-20.878$ to $-24.924$ kcal/mol) for 0-150 residues and 21% (from $-27.145$ to $-32.805$ kcal/mol) for 150-300 residues. Unlike single-objective methods that trade predicted stability for other properties, ProteinZero simultaneously improves TM-score (0.812 to 0.867), diversity (0.281 to 0.306), recovery (0.574 to 0.590), and success rate (84.45% to 90.13%) for small proteins, with similar improvements for larger ones (see Table 1; extended metrics with wild-type and generated absolute energies in Table 10, Appendix C.5). We note that while Fast-ddG is optimized for full-sequence redesign, its correlation with experimental single-point mutations on Ssym (Section C.4) provides evidence that it captures thermodynamic signal beyond proxy-specific patterns, although it remains a computational surrogate rather than a substitute for experimental measurement.

### 4.3 Case Study on Diverse Protein Folds and Complex Protein Design Tasks

**Stabilization of Natural Proteins Across Diverse Folds:** Our visual comparison in Figure 6 shows ProteinZero converts naturally unstable proteins into designs with substantially improved predicted stability while maintaining structural fidelity. For challenging targets like membrane proteins and $\beta$-barrels, for example, our method achieves substantial stability improvements. The $\beta$-barrel structure (4FD5 chain A) transforms from unstable wild-type (FoldX ddG: 25.75 kcal/mol) to a design with improved predicted stability (FoldX ddG: $-34.18$ kcal/mol), while the membrane protein (2W7T chain A) improves from 42.01 to $-36.09$ kcal/mol. These results show ProteinZero's optimization of sequence-structure relationships, generating predicted stability profiles that may be of interest for therapeutic and industrial applications, pending experimental validation. By consistently producing designs with high predicted structural accuracy and predicted thermodynamic stability across $\alpha$-helical, $\beta$-sheet, and mixed $\alpha/\beta$ folds, our approach expands the design space. While these computational evaluation metrics are promising, experimental validation remains essential to confirm functional properties.

**Performance Scaling Across Protein Complexity:** When compared with InstructPLM (our base model), ProteinZero demonstrates consistent improvements across diverse protein architectures. For challenging $\beta$-rich structures, our approach achieves higher structural accuracy (TM-score: 0.949 vs 0.910 for 1XXM chain C) and improved predicted stability (FoldX ddG: $-28.94$ vs $-8.94$ kcal/mol). These gains

Table 3: Ablation studies for 0-150 and 150-300 residue proteins across three design dimensions: reward models, learning objectives, and diversity regularization strategies. Best results highlighted in  blue .

| Length | Design Configuration | InverseFold Acc. Recovery Rate ↑ | Thermal Stability Metrics Fast-ddG ↓ | FoldX ddG ↓ | Designability Metrics TM Score ↑ | PLDDT ↑ | Diversity ↑ | scRMSD ↓ (scRMSD <2Å% ↑) | Overall Success (%) ↑ |
|---|---|---|---|---|---|---|---|---|---|
| | *Design Dimension 1: Reward Model Formulation* | | | | | | | | |
| | Only TM-score as Reward | 0.582 | -21.598 | -21.271 | 0.874 | 82.827 | 0.293 | 1.372 (93.62%) | 89.52% |
| | Only ddG as Reward | 0.580 | -22.996 | -25.381 | 0.831 | 82.270 | 0.299 | 1.466 (87.75%) | 85.15% |
| | Full ProteinZero (TM+ddG) | 0.590 | -22.616 | -24.924 | 0.867 | 82.326 | 0.306 | 1.373 (93.55%) | 90.13% |
| | *Design Dimension 2: Learning Objective Components* | | | | | | | | |
| | Without Diversity Term | 0.584 | -22.526 | -24.877 | 0.861 | 82.308 | 0.268 | 1.397 (92.75%) | 90.23% |
| 0-150 residues | Without KL Term | 0.564 | -22.352 | -24.264 | 0.841 | 80.979 | 0.316 | 1.429 (90.53%) | 86.41% |
| | Full ProteinZero (All Terms) | 0.590 | -22.616 | -24.924 | 0.867 | 82.326 | 0.306 | 1.373 (93.55%) | 90.13% |
| | *Design Dimension 3: Diversity Regularization Strategies* | | | | | | | | |
| | Diversity as Reward | 0.579 | -19.738 | -18.681 | 0.836 | 81.107 | 0.284 | 1.439 (87.77%) | 78.65% |
| | Hamming Distance as Reward | 0.565 | -14.137 | -11.135 | 0.831 | 81.785 | 0.276 | 1.466 (88.70%) | 74.63% |
| | Full ProteinZero (Embedding Diversity) | 0.590 | -22.616 | -24.924 | 0.867 | 82.326 | 0.306 | 1.373 (93.55%) | 90.13% |
| | *Design Dimension 1: Reward Model Formulation* | | | | | | | | |
| | Only TM-score as Reward | 0.577 | -35.793 | -25.905 | 0.870 | 84.237 | 0.333 | 1.384 (91.25%) | 89.76% |
| | Only ddG as Reward | 0.574 | -42.769 | -35.907 | 0.831 | 83.540 | 0.327 | 1.447 (88.52%) | 87.38% |
| | Full ProteinZero (TM+ddG) | 0.580 | -40.626 | -32.805 | 0.862 | 84.154 | 0.331 | 1.393 (90.43%) | 91.19% |
| | *Design Dimension 2: Learning Objective Components* | | | | | | | | |
| 150-300 residues | Without Diversity Term | 0.580 | -37.905 | -31.185 | 0.860 | 84.065 | 0.281 | 1.401 (89.71%) | 91.32% |
| | Without KL Term | 0.569 | -40.688 | -33.193 | 0.835 | 83.008 | 0.328 | 1.440 (89.08%) | 87.92% |
| | Full ProteinZero (All Terms) | 0.580 | -40.626 | -32.805 | 0.862 | 84.154 | 0.331 | 1.393 (90.43%) | 91.19% |
| | *Design Dimension 3: Diversity Regularization Strategies* | | | | | | | | |
| | Diversity as Reward | 0.558 | -33.904 | -23.967 | 0.849 | 83.326 | 0.315 | 1.421 (89.32%) | 81.71% |
| | Hamming Distance as Reward | 0.568 | -32.128 | -23.228 | 0.836 | 83.668 | 0.294 | 1.432 (89.14%) | 80.29% |
| | Full ProteinZero (Embedding Diversity) | 0.580 | -40.626 | -32.805 | 0.862 | 84.154 | 0.331 | 1.393 (90.43%) | 91.19% |

extend across $\beta$-sheets, $\alpha/\beta$ mixed domains, and $\alpha$-helical structures, as shown in Figure 6. ProteinZero delivers substantial improvements for both protein size categories: for 0-150 residues, success rate increases from 84.45% to 90.13%, predicted stability improves from $-20.878$ to $-24.924$ kcal/mol, and diversity rises from 0.281 to 0.306. For 150-300 residues, we observe comparable gains: success rate from 86.38% to 91.19%, predicted stability from $-27.145$ to $-32.805$ kcal/mol, and diversity from 0.305 to 0.331. The maintained performance improvements for larger proteins suggest our reinforcement learning framework handles increased structural complexity effectively.

### 4.4 Exploring the Design Space of Online RL for Fine-tuning Protein Generative Models

**Reward Model Designs:** Our ablation studies demonstrate that combining TM-score and stability rewards yields the highest overall success rates, consistently outperforming single-objective settings. For proteins of 0-150 residues, the combined reward achieves 90.13% success, compared to 89.52% with TM-score only and 85.15% with stability only. For larger proteins (150-300 residues), success rates are 91.19% for the combined setting, versus 89.76% and 87.38%, respectively. Examining the individual objectives explains this gap: optimizing only TM-score achieves the best structural accuracy (TM: 0.874 vs. 0.867 for the combined setting, 0-150 residues) but reduces stability, while optimizing only stability improves FoldX ddG ($-25.381$ vs. $-24.924$ kcal/mol) but compromises structural accuracy (TM: 0.831 vs. 0.867). By contrast, the combined reward balances both criteria, closing the trade-off and substantially reducing design failures. A full sweep over intermediate $\lambda_{\text{TM}}/\lambda_{\Delta\Delta G}$ weightings is reported in Appendix C.11.

**Learning Objective Components:** We ablate the diversity regularization and KL divergence to assess their contributions (Table 3). Removing the diversity regularization marginally improves success rate (90.23% vs. 90.13% for 0-150 residues, 91.32% vs. 91.19% for 150-300 residues), but significantly reduces sequence diversity from 0.306 to 0.268 for 0-150 residues and from 0.331 to 0.281 for 150-300 residues. This 12-15% reduction in diversity indicates convergence to a narrower solution space, limiting its ability to explore functionally diverse sequences, a key concern with offline RL methods. By contrast, removing KL divergence causes severe degradation: success rate drops by nearly 4%, TM-score declines by around 0.03, and pLDDT decreases by around 1.3, reflecting both reduced structural accuracy and confidence. These results show KL regularization is essential for stable optimization and preventing catastrophic forgetting, while diversity regularization, though slightly reducing peak performance, preserves exploration crucial for discovering novel protein designs beyond the training distribution.

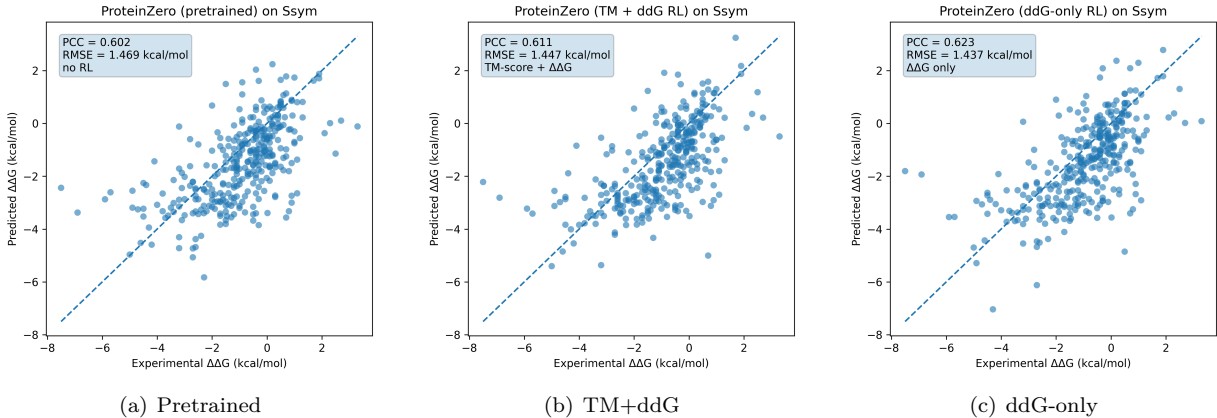

(a) Pretrained        (b) TM+ddG        (c) ddG-only

Figure 3: Fast-ddG predictor performance on the Ssym dataset with 342 wet-lab validated single-point mutations (wild-type → mutant). Each subfigure shows predicted versus experimental $\Delta\Delta G$ values for different model variants: (a) pretrained model before RL fine-tuning, (b) fine-tuned with joint TM-score + Fast-ddG rewards, (c) fine-tuned with Fast-ddG reward only. All variants achieve comparable correlation with experimental measurements (PCC ≈ 0.60–0.62, RMSE ≈ 1.44–1.47 kcal/mol).

**Diversity Regularization Strategies:** We compare three strategies for incorporating diversity into the optimization process (Table 3; detailed results in Appendix Table 17): embedding-based diversity as a reward, Hamming distance as a reward, and embedding-based diversity as a regularization term in the loss. Introducing diversity directly into the reward sharply reduces performance, with success rates falling to 78.65% and 81.71%, and stability values deteriorating relative to the baseline. Using Hamming distance performs even worse, lowering success rates to 74.63% and 80.29% and further degrading stability and structural accuracy. By contrast, applying embedding-based diversity as a regularizer maintains success rates of 90.13% and 91.19%, preserves sequence diversity at 0.306 and 0.331, and avoids losses in stability or accuracy. Thus, reward-based diversity introduces conflicting signals that destabilize training, whereas regularization provides consistent gradients that encourage exploration while safeguarding functional objectives.

The ablation studies validate our design choices and highlight the importance of balancing multiple objectives in online RL for protein design. ProteinZero navigates these trade-offs through separated optimization signals, multi-objective rewards for primary objectives, and regularization for exploration and stability, yielding a robust approach generalizing across protein sizes and architectures.

### 4.5 Fast-ddG Accuracy for Predicting Mutational $\Delta\Delta G$ Using Wet-Lab Validated Data

We assess Fast-ddG correlation with experimental measurements on the Ssym benchmark (Pucci et al., 2018), comprising 684 single-point mutations with calorimetrically measured $\Delta\Delta G$ values. Consistent with Eq. 4, we evaluate 342 wild-type→mutant transitions by computing stability changes on wild-type backbones. Table 8 (Appendix C.4) compares our predictor against physics-based oracles (FoldX, Rosetta) and supervised predictors (ThermoMPNN (Dieckhaus et al., 2024), ThermoNet (Li et al., 2020), PROSTATA (Umerenkov et al., 2022)). Across three configurations, pretrained, Fast-ddG-only, and TM-score + Fast-ddG, our predictor achieves RMSE 1.44–1.47 kcal/mol and PCC 0.60–0.62, matching FoldX (RMSE: 1.56, PCC: 0.63) while operating 236–760× faster (Tables 4–5). This represents 56% RMSE reduction versus ProteinMPNN (3.38 kcal/mol, PCC: 0.26), demonstrating gains from specialized optimization. While ThermoMPNN achieves superior performance (1.12, 0.72), it requires supervised training and handles only single-residue perturbations, whereas our unsupervised, self-derived predictor generalizes to full sequence redesigns. Per-protein analysis (Table 7) confirms improvements generalize across diverse targets rather than reflecting outlier bias. For the four largest proteins (1L63, 2LZM, 1LZ1, 1BNI; 72.5% of mutations), fine-tuning consistently reduces RMSE, with Fast-ddG-only achieving best correlation on three of four. On 1L63 (118 mutations), RMSE improves from 1.36 to 1.30 kcal/mol and PCC from 0.58 to 0.66. Representative cases with substantial error reductions appear in Table 9 (Appendix C.4). Figure 3 shows linear correspondence

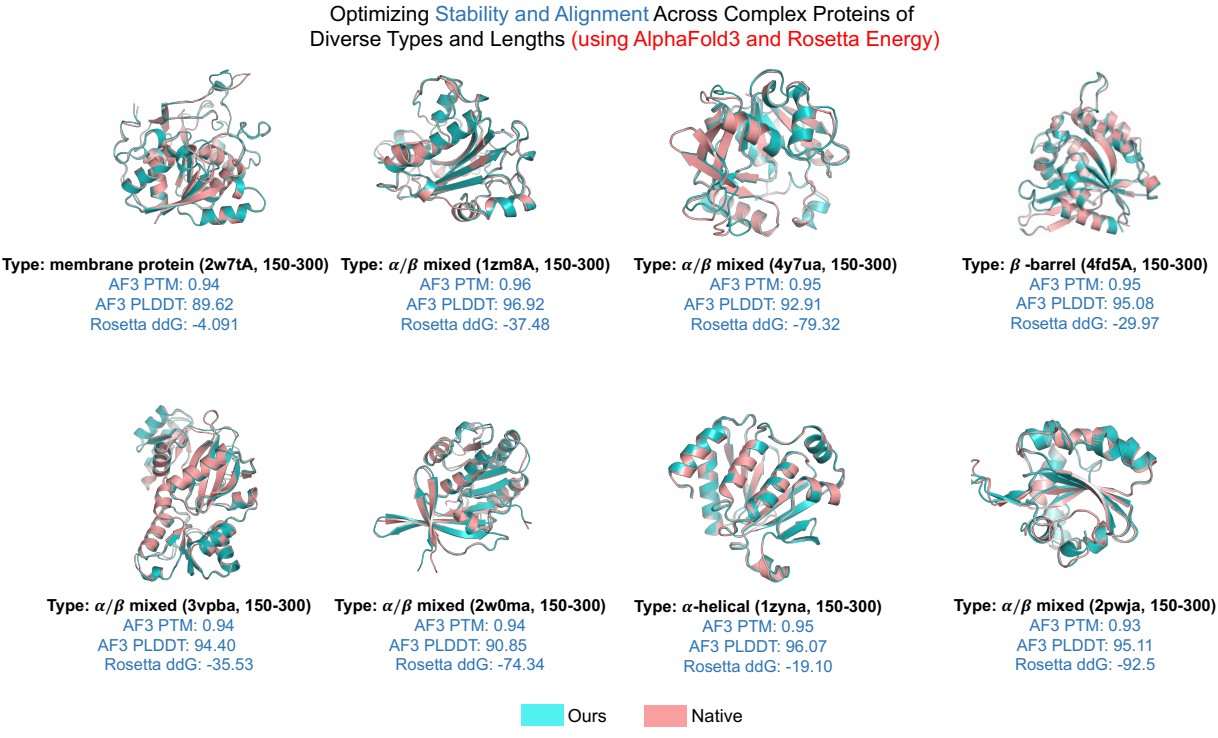

Figure 4: **Evaluation of ProteinZero designs under AlphaFold3 and Rosetta Energy.** Eight backbones (150–300 residues) spanning membrane proteins, $\alpha/\beta$-mix domains, $\alpha$-helical structures, and $\beta$-barrels, evaluated with AlphaFold3 (structure prediction) and Rosetta Energy (predicted stability). Designed sequences (cyan) are superimposed on the native target (pink). Across the eight designs, AF3 PTM scores fall in 0.93–0.96, pLDDT in 89.62–96.92, and Rosetta ddG in $-4.091$ to $-92.5$. These computational measurements are consistent with the FoldX/ESMFold observations in the main results.

(PCC $\approx$ 0.60–0.62) with reduced scatter post-tuning. These results indicate that Fast-ddG achieves accuracy comparable to physics-based benchmarks on single-point mutation data, while maintaining computational efficiency for online RL. We emphasize, however, that this experimental validation is moderate in strength (PCC $\approx$ 0.60–0.62) and is restricted to single-mutation transitions, whereas ProteinZero applies Fast-ddG to full-sequence redesign. Fast-ddG should therefore be regarded as a useful and computationally efficient training proxy rather than a calibrated predictor of absolute folding stability for redesigned sequences.

## 5 Conclusion

We presented ProteinZero, an online reinforcement learning framework that enables protein generative models to improve beyond supervised pretraining by learning from their own outputs. It integrates two methodological advances: a fast, unsupervised ddG predictor for efficient stability signals and an embedding-level diversity regularizer that prevents collapse while broadening sequence-level diversity among candidate designs. These components make online RL tractable for protein design and offer insights for broader RLHF by addressing efficiency and diversity collapse. Experiments show consistent multi-objective gains across structural accuracy, predicted stability, recovery, and diversity, including on challenging folds such as $\beta$-barrels and membrane proteins. While evaluation relies on computational oracles, validated against experimental Ssym data and cross-checked across independent predictors (ESMFold, AlphaFold3, FoldX), prospective wet-lab validation remains essential to confirm that improvements translate to experimental outcomes. Nevertheless, the results demonstrate that efficient online RL can complement supervised methods through scalable feedback, expanding the accessible design space with the potential to support future applications in therapeutics, enzymes, and synthetic biology.

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

# A    Discussion

## A.1    Broader Impact

ProteinZero represents a methodological advancement in computational protein design by enabling autonomous improvement of generative models through online reinforcement learning. As illustrated in Figure 5, our framework integrates within the broader protein design pipeline, bridging computational optimization and experimental validation. By reducing reliance on manually curated datasets from repositories like the Protein Data Bank, which capture only a fraction of viable sequence space, our approach offers new possibilities for exploring protein designs beyond naturally occurring examples.

The computational efficiency gains (achieving comparable results with substantially reduced computational time compared to physics-based methods) and improved success rates demonstrated in our experiments could accelerate research in therapeutic development, enzyme engineering, and industrial biotechnology. The reduced computational requirements potentially improve accessibility of advanced protein design capabilities for research groups with limited resources. Applications span from developing novel biologics and vaccines to engineering enzymes for sustainable manufacturing and bioremediation.

However, we emphasize that our computational metrics, while encouraging, require experimental validation to confirm biological functionality. The stability and foldability improvements we demonstrate computationally may not directly translate to enhanced catalytic activity, binding affinity, or other functional properties critical for real-world applications. Furthermore, the path from computational design to practical application involves multiple validation stages. Each designed protein undergoes synthesis, experimental characterization, functional testing, and regulatory approval before deployment. This established multi-step process provides checkpoints for safety and efficacy verification. Our computational improvements represent the initial stage of this pipeline, with subsequent experimental validation remaining essential for confirming biological relevance.

The self-improving nature of ProteinZero, learning continuously from generated outputs rather than requiring new experimental data, represents a shift toward more autonomous systems in computational biology. While this offers exciting possibilities for accelerating discovery, the comprehensive experimental validation pipeline ensures that computational predictions are rigorously tested before practical application. We envision this work contributing to a new generation of AI systems that can explore biological design spaces more efficiently, ultimately advancing our understanding and engineering capabilities in protein science.

Like other advances in computational protein design, ProteinZero is dual-use: techniques that improve the designability and predicted stability of beneficial proteins could in principle be applied to harmful ones, including toxins or pathogen-related components. We take this risk seriously and do not consider it negligible. At the same time, several properties of our method limit the concern: ProteinZero is an inverse-folding fine-tuning method that optimizes sequences for predetermined backbones rather than generating novel folds or functions, and its rewards (designability, predicted stability, and diversity) are not function- or virulence-specific, so the method confers no targeted capability toward harmful function. To further reduce risk, we adopt responsible-release practices. Code and model checkpoints will be released under an intended-use license that permits academic research, reproduction, and benchmarking without restriction, while prohibiting use intended to design harmful proteins; legitimate scientific use is therefore unaffected. We additionally encourage all users to screen designed sequences against established biosecurity tools, such as synthesis-provider screening or the IBBIS Common Mechanism, prior to synthesis. We further emphasize that ProteinZero's outputs are computational predictions only: no designed sequence should proceed to wet-lab synthesis or deployment without prior experimental characterization and institutional biosafety and biosecurity review. We see methodological progress in protein design and investment in screening, governance, and safety review as necessary complements, and encourage the community to advance them together.

From an ethical perspective, all experiments in this work were conducted using publicly available datasets (CATH 4.3) and computational simulations without any wet-lab experimentation or use of biological materials. We emphasize that any deployment of designed proteins must follow established safety protocols, regulatory frameworks, and ethical guidelines for biological research. We commit to making our code pub-

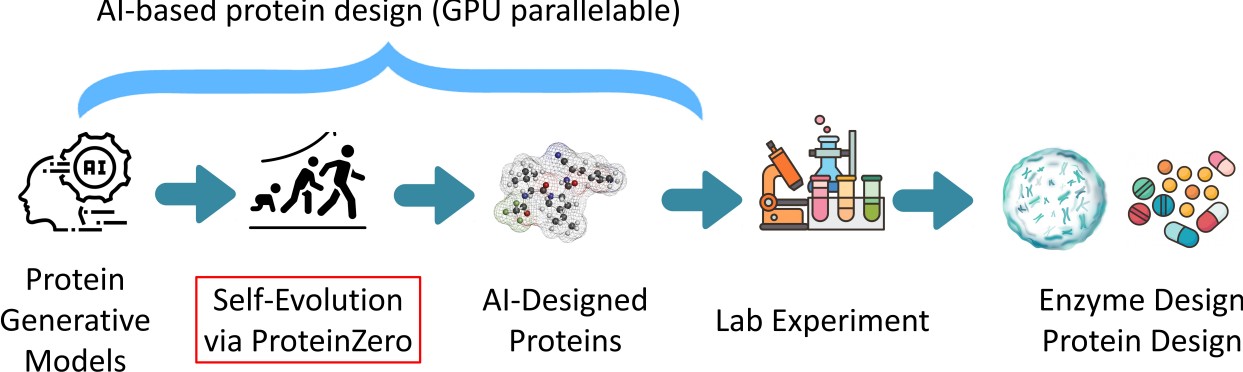

Figure 5: Integration of ProteinZero within the AI-driven protein design pipeline. Pre-trained generative models evolve through ProteinZero's online reinforcement learning framework to produce optimized protein sequences. These AI-designed candidates proceed to laboratory synthesis and experimental characterization, enabling applications in diverse biotechnological domains such as enzyme engineering and therapeutic development. The computational stages (blue) can leverage GPU parallelization for efficient large-scale processing.

licly available upon publication to ensure transparency and enable the research community to build upon and scrutinize our work.

## A.2 Limitations and Future Directions

**Restriction to Monomeric Scaffolds.** Our experiments target monomeric proteins, a practically significant class encompassing critical therapeutic modalities: *de novo* miniprotein inhibitors (Cao et al., 2020), antigen-display architectures (Lutz et al., 2023), and cyclic peptide binders (Rettie et al., 2025). Leading generative methods including RFdiffusion (Watson et al., 2023), ProteinMPNN (Dauparas et al., 2022), and Chroma (Ingraham et al., 2023) have similarly demonstrated advances on single-chain scaffolds. However, many drug discovery applications require multimeric complexes and protein-protein interfaces. The core framework components—online RL optimization, embedding-level diversity regularization, and fast proxy rewards—are architecture-agnostic and naturally extend to assemblies by incorporating interface-aware structural rewards and multimer-capable stability predictors to optimize binding affinity and interface packing simultaneously.

**Reliance on Computational Proxies.** ProteinZero employs computational predictors (Fast-ddG, ESM-Fold TM-score) as reward signals. While these metrics act as proxies for biological properties rather than substitutes for experimental validation, computational screening remains standard in protein engineering pipelines. Empirical studies demonstrate that computational stability predictors enrich for mutations that experimentally increase protein thermodynamic stability: 30–40% of computationally predicted stabilizing mutations are confirmed stable in wet-lab validation, compared to near-zero success rates for random amino acid substitutions (Wijma et al., 2014; Broom et al., 2017; Buß et al., 2018). However, individual oracles encode systematic preferences—FoldX, for instance, favors mutations that increase hydrophobic core packing, which may trade off against solubility (Broom et al., 2017). A specific scope limitation concerns Fast-ddG: its experimental validation (Section 4.5) is moderate (PCC ≈ 0.60–0.62 on Ssym) and based solely on single-point mutations, while it is deployed here as a reward for full-sequence redesign. We therefore treat it as a training proxy that provides an efficient stability signal, not as a validated stand-in for experimental $\Delta\Delta G$ measurement on redesigned sequences.

We address over-optimization to single-oracle patterns through two mechanisms. First, multi-objective optimization with diversity regularization enforces complementary constraints (structural designability, stability, KL-regularization, embedding diversity), preventing the policy from satisfying one objective at the expense of others. Second, independent evaluation rigorously separates training rewards

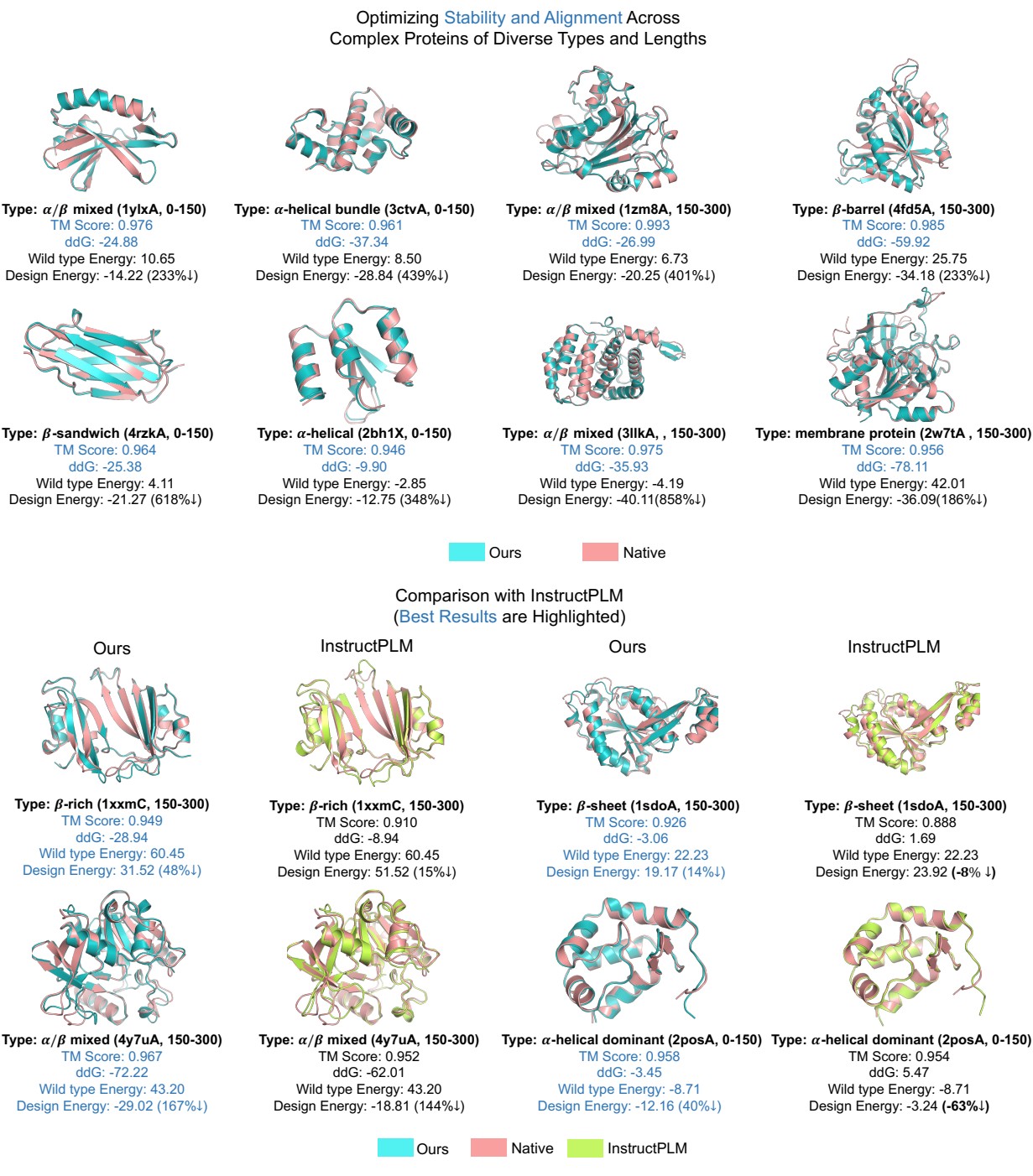

Figure 6: **Representative cases of protein structure designs from held-out test set.** Visual comparison between ProteinZero (cyan), native proteins (pink), and InstructPLM (lime green). Top panels show selected cases where naturally unstable proteins are redesigned by ProteinZero. In these examples, predicted stability improvements range from 233% to 858% (based on FoldX ddG calculations) while maintaining structural similarity (TM-scores > 0.95). Bottom panels present comparative examples with InstructPLM for challenging β-rich structures and complex architectures. In the shown cases, ProteinZero generates designs with negative predicted ddG values while InstructPLM produces positive values indicating predicted instability. These visualizations represent individual design outcomes; comprehensive quantitative results are provided in Table 1.

(Fast-ddG, ESMFold) from evaluation metrics (FoldX, AlphaFold3). Transferability to these independent oracles (Section 4) suggests the model captures signal that generalizes across computational oracles rather than oracle-specific patterns; confirming that this corresponds to genuine biophysical principles requires prospective experimental validation.

# B  Experimental Details

## B.1  Prompt/Task Datasets

We utilized the CATH-4.3 dataset for training and evaluation, which contains protein domains classified according to Class, Architecture, Topology, and Homology. The dataset was stratified into two categories based on sequence length: 0-150 residues and 150-300 residues to evaluate performance across different structural complexity levels. For rigorous evaluation, we constructed held-out test sets with sequence identity thresholds of <40% for 0-150 residue proteins and <30% for 150-300 residue proteins, ensuring assessment on genuinely out-of-distribution structures. This stringent filtering prevents overlap with both the training data and the pre-training datasets used by baseline models (e.g., InstructPLM was pre-trained on CATH-4.2).

During online reinforcement learning, our approach generates training signals entirely from model outputs evaluated by reward functions, without requiring labeled sequence-structure pairs. The model iteratively improves through self-generated examples assessed by our computational reward pipeline. This self-improving paradigm represents a fundamental departure from supervised methods that depend on curated datasets, enabling continuous learning without additional experimental data collection.

## B.2  Implementation Details

### B.2.1  Hyperparameter Settings

**ProteinZero$_{\textbf{RAFT}}$:** We optimize our model with AdamW using an initial learning rate of $3\times10^{-5}$ ($\beta_1 = 0.9$, $\beta_2 = 0.999$, $\epsilon = 1 \times 10^{-8}$, weight decay = 0.01) over all RAFT iterations. For each RAFT iteration, we apply a linear learning-rate decay (with zero warm-up) over the epochs. We apply rank-16 LoRA adapters ($\alpha_{\text{LoRA}} = 16$, dropout = 0.05) to all self-attention and feed-forward projections. During each iteration, we partition the CATH 4.3 training set across GPUs, generating $K = 8$ candidate sequences per backbone on each of the 8 GPUs via nucleus sampling (temperature = 0.8, $p = 0.9$), yielding 64 candidate sequences per backbone in total, and retain only the highest-reward sequence for fine-tuning. Gradient updates are performed only for backbones where at least 50% of the generated sequences achieve pLDDT > 80. Our policy updates incorporate a KL regularizer with a coefficient of 0.1 against a frozen reference policy, whereas the original RAFT implementation used a grid search to explore different KL term weights (0 (disabled), 0.005, 0.01, 0.1). We conduct extensive ablation studies on the KL weight used in the original RAFT in Table 6 within Section C.2. Our empirical analysis reveals that this specific KL weight parameterization of 0.1 is critical for achieving superior performance within the **ProteinZero$_{\textbf{RAFT}}$** framework. We additionally employ an embedding-space diversity penalty with a coefficient of 0.05, which was not included in the original RAFT. The reward function equally weights TM-score and predicted $\Delta\Delta G$. All experiments utilize mixed-precision FP16 (or BF16 where available) with two-step gradient accumulation per update. Our results suggest that stronger KL regularization helps mitigate instability in pretrained protein language models during fine-tuning.

**ProteinZero$_{\textbf{GRPO}}$:** We optimize our model using AdamW with an initial learning rate of $1\times10^{-6}$ ($\beta_1 = 0.9$, $\beta_2 = 0.999$, $\epsilon = 1 \times 10^{-8}$, weight decay = 0), and employ a linear learning-rate scheduler (no warm-up) over all 20 GRPO iterations. We apply LoRA adapters with rank $r = 16$ (scaling factor $\alpha_{\text{LoRA}} = 16$, dropout = 0.05) to all self-attention and feed-forward projections. Each episode samples from the CATH 4.3 training set (distributed across GPUs) and generates $K = 8$ candidate sequences per backbone on each of the 8 GPUs via nucleus sampling (temperature = 0.8, $p = 0.9$), yielding 64 candidate sequences per backbone in total, consistent with the original GRPO implementation (Shao et al., 2024). Policy updates proceed only when at least 50% of the generated sequences achieve pLDDT > 80. For policy optimization, we employ a GRPO clipping coefficient $\varepsilon = 0.1$ with KL regularization against a frozen reference policy with

Table 4: Total wall-clock time (including MSA and template search) required to generate reward for eight inverse-folding sequences conditioned on the same structural backbone. Best results are highlighted in blue. The wall-clock time without MSA search can be found in Appx. Table 5

| Length range | Structural and Designability Reward | | | | | Thermal Stability Reward (ddG) | |
|---|---|---|---|---|---|---|---|
| | ESMFold | AlphaFold 2 | ColabFold | OpenFold | AlphaFold 3 | Fast-ddG(ours) | FoldX |
| **0-150 aa** | 18.7 s | 1632.6 s ($\sim$**87.3**$\times$) | 576.2 s ($\sim$**30.8**$\times$) | 674.9 s ($\sim$**36.1**$\times$) | 705.4 s ($\sim$**37.7**$\times$) | $\sim$**2 s (GPU)** | 472.3 s ($\sim$**236.2**$\times$) |
| **150-300 aa** | 47.5 s | 4112.5 s ($\sim$**86.6**$\times$) | 1272.8 s ($\sim$**26.8**$\times$) | 1424.7 s ($\sim$**30.0**$\times$) | 1920.5 s ($\sim$**40.4**$\times$) | $\sim$**2 s (GPU)** | 1520.6 s ($\sim$**760.3**$\times$) |

Table 5: Total wall-clock time (excluding Multiple Sequence Alignment) required to generate eight inverse-folding sequences conditioned on the same structural backbone for each reward component in our fine-tuning pipeline (often used for De novo design tasks, but we focus on inverse folding tasks.). Best results are highlighted in blue.

| Length range | Structural Alignment Reward (Prediction) | | | | | Design Stability Reward (ddG) | |
|---|---|---|---|---|---|---|---|
| | ESMFold | AlphaFold 2 | ColabFold | OpenFold | AlphaFold 3 | Predicted-ddG | FoldX |
| **0-150 aa** | 18.7 s | 197.6 s ($\sim$**10.6**$\times$) | 193.6 s ($\sim$**10.4**$\times$) | 189.6 s ($\sim$**10.1**$\times$) | 199.2 s ($\sim$**10.7**$\times$) | $\sim$**2 s (GPU)** | 472.3 s ($\sim$**236.2**$\times$) |
| **150-300 aa** | 47.5 s | 223.2 s ($\sim$**4.7**$\times$) | 217.6 s ($\sim$**4.6**$\times$) | 200.8 s ($\sim$**4.2**$\times$) | 237.6 s ($\sim$**5.0**$\times$) | $\sim$**2 s (GPU)** | 1520.6 s ($\sim$**760.3**$\times$) |

a coefficient of 0.1, complemented by an embedding-space diversity penalty with a coefficient of 0.05. The reward function equally weights TM-score and predicted $\Delta\Delta G$. All experiments use mixed-precision FP16, with no gradient accumulation to ensure each episode constitutes a complete policy update. We note that our KL regularization weight of 0.1 differs from the original GRPO implementation (Shao et al., 2024), which uses 0.04. We conduct extensive ablation studies on the KL weight used in the original GRPO in Table 6 within Section C.2. These experiments demonstrate that the KL weight configuration is essential for optimal performance in our **ProteinZero$_{\textbf{GRPO}}$** setting, which establishes our configuration as the optimal solution. Our ablation studies reveal that decreasing KL regularization strength leads to performance degradation across multiple metrics, including sequence recovery, Fast-ddG, FoldX DDG, TM-score, pLDDT, scRMSD, and success rate. These findings indicate that stronger KL regularization may help stabilize pretrained protein language models during fine-tuning.

**Direct Preference Optimization (Baseline):** For each target structure, we sample $K = 8$ candidate sequences at a temperature of $T = 0.1$ to form chosen-rejected pairs according to our reward model, and we optimize the DPO loss over 20 epochs using AdamW with $\beta_1 = 0.9$, $\beta_2 = 0.999$, and a weight decay of 0.01. Training proceeds only for backbones where at least 50% of the sampled sequences achieve pLDDT > 80. We apply a KL divergence regularization term against a frozen reference policy with a coefficient of 0.1, and we incorporate an embedding-space diversity penalty with a coefficient of 0.05. All experiments are conducted in mixed-precision FP16.

**Multi-Round Direct Preference Optimization (Baseline):** We extend DPO to iterative refinement across multiple rounds. For each round, we sample $K = 8$ candidate sequences per target structure at temperature $T = 0.1$ from the current policy to form new chosen-rejected pairs according to our reward model, optimizing the DPO loss for 5 epochs per round using AdamW with $\beta_1 = 0.9$, $\beta_2 = 0.999$, and weight decay of 0.01. Gradient updates are performed only when at least 50% of the generated sequences for a backbone achieve pLDDT > 80. We apply KL divergence regularization against the frozen reference policy (coefficient 0.1) and embedding-space diversity penalty (coefficient 0.05). All experiments are conducted in mixed-precision FP16.

### B.2.2 Hardware Usage

All experiments are conducted using eight NVIDIA A100 GPUs. We fine-tune the pretrained model by stratifying protein sequences into two length categories: 0-150 amino acids and 150-300 amino acids. Our training protocol divides each complete dataset pass into 20 iterations for granular optimization control. We report that processing one full epoch requires approximately 3.23 hours for the 0-150 amino acid category and 25.58 hours for the 150-300 amino acid category. The number of training epochs can be flexibly adjusted

based on desired performance improvements and available computational resources. This modular approach enables researchers to balance training thoroughness with computational constraints, making online RL fine-tuning feasible on a single multi-GPU node within practical timeframes.

## B.3 Evaluation Metrics

For the main results in Table 1, we report mean and standard error over 10 independent training runs that differ in random seed. Within each run, we follow a fixed evaluation protocol that uses the sampling temperatures recommended by the InstructPLM (Qiu et al., 2024): rollouts during training use the recommended exploration setting (temperature 0.8, top-$p$ 0.9; Section B), while all test-set evaluation uses the recommended evaluation setting (temperature 0.15, top-$p$ 0.9). For each run, we generate 64 sequences per backbone under this evaluation setting and score them with the oracle metrics; these scores are then averaged across all test-set backbones and the 64 sequences per backbone to yield a single per-run scalar per metric. The 10 resulting per-run scalars yield the mean and standard error reported in Table 1.

### B.3.1 Structural Accuracy

- TM Score: Measures the topological similarity between predicted and target structures, with values ranging from 0 to 1 (higher is better) (Zhang & Skolnick, 2004).

- PLDDT (Predicted Local Distance Difference Test): Assesses the confidence in local structure prediction (Jumper et al., 2021; Abramson et al., 2024).

- scRMSD (Self-consistency RMSD of structures): Measures the deviation of side chain positions, with percentage below 2 Å reported as an additional quality indicator (Qiu et al., 2024; Park et al., 2024).

### B.3.2 Stability Metrics

- Fast-ddG (Jiao et al., 2025): Predicted change in Gibbs free energy, estimated directly from the model.

- FoldX ddG (Schymkowitz et al., 2005): A more rigorous physics-based calculation of stability using the FoldX force field, which better correlates with experimental measurements.

### B.3.3 Sequence Properties

- Recovery: The percentage of amino acids matching reference sequences, indicating how well the model captures natural sequence preferences (Park et al., 2024).

- Diversity: A measure of variation among generated sequences, calculated as the mean normalized Hamming distance between every pair of sequences conditioned on the same backbone (score ranges from 0 for identical sequences to 1 for sequences that differ at every position):

$$D_{\text{Hamming}}(\mathcal{B}) \; = \; \frac{2}{B(B-1)} \sum_{1 \leq i < j \leq B} \left[ \frac{1}{L} \sum_{t=1}^{L} \mathbf{1}\big[y_{i,t} \neq y_{j,t}\big] \right].$$

## B.4 Baseline Methods

We compared ProteinZero against several state-of-the-art methods:

### B.4.1 Supervised Inverse Folding Models

1. ProteinMPNN: A graph-based model that directly predicts amino acid sequences from backbone structures.

2. ESM-IF: A transformer-based inverse folding model trained on substantial structural data.

3. InstructPLM (our base model): A recently developed protein language model fine-tuned to follow structural design instructions.

### B.4.2  Offline RL Baseline

DPO (Direct Preference Optimization): A widely used offline reinforcement learning method that learns from preference data without online interaction.

Multi-Round DPO: An iterative extension of DPO that regenerates preference pairs from the updated policy at each round, allowing for progressive refinement while remaining offline.

For fair comparison, all baseline methods used the same evaluation protocol and metrics. InstructPLM served as our starting model for ProteinZero fine-tuning, establishing a direct comparison between supervised learning and our online RL approach.

## B.5  Reward Model

Traditional methods for evaluating protein designs require minutes to hours per evaluation, making online reinforcement learning impractical. We solve this challenge with two efficient reward models:

### B.5.1  Structural Alignment Reward

We use ESMFold for structural inference instead of the slower AlphaFold2/3 (Jumper et al., 2021; Abramson et al., 2024). The TM-score reward $r_{\mathrm{TM}}(x, y)$ is computed by first folding the generated sequence $y$ using ESMFold, then calculating the TM-score (Zhang & Skolnick, 2004) between the predicted structure and the target structure $x$ with US-align (Zhang et al., 2022), an updated implementation from the original TM-align (Zhang & Skolnick, 2005).

### B.5.2  Design Stability Reward

We calculate $r_{\Delta\Delta\mathrm{G}}(x, y)$, the estimation of $\Delta\Delta G$ by comparing the backbone-conditioned likelihood of each generated sequence with an unconditional sequence prior, $p_\varphi(y)$, provided by pretrained inverse folding models such as ProteinMPNN and InstructPLM, as proposed in (Jiao et al., 2025; Shanker et al., 2024; Widatalla et al., 2024; Cagiada et al., 2025; Bennett et al., 2023): $\Delta\Delta G(x, y) = -k_B T[(\log p_\theta(y \mid x) - \log p_\varphi(y)) - (\log p_\theta(y_{\mathrm{wt}} \mid x) - \log p_\varphi(y_{\mathrm{wt}}))]$, where $y_{\mathrm{wt}}$ represents the PDB wild-type sequence and $k_B T$ represents the thermal energy at 298 K ($0.593\,\mathrm{kcal\,mol^{-1}}$).

Our reward combines both scores after min-max normalization across the candidate pool of inverse folding sequences generated for the same backbone within each reinforcement learning iteration: $\tilde{r}_{\mathrm{TM}} = (r_{\mathrm{TM}} - r_{\mathrm{TM}}^{\min})/(r_{\mathrm{TM}}^{\max} - r_{\mathrm{TM}}^{\min})$ and $\tilde{r}_{\Delta\Delta G}$ analogously, giving $r(x, y) = \lambda_{\mathrm{TM}}\tilde{r}_{\mathrm{TM}}(x, y) + \lambda_{\Delta\Delta G}\tilde{r}_{\Delta\Delta G}(x, y)$. This reward model accelerates evaluation speed by at least $2500\times$ compared to traditional methods, reducing training time from months to days. The effectiveness of this approach is demonstrated through comprehensive evaluation metrics presented in Table 1.

## B.6  Online RL Algorithms

We implemented and evaluated two online reinforcement learning algorithms for ProteinZero:

### B.6.1  ProteinZero$_{\mathsf{RAFT}}$

Our adaptation of Reward-rAnked Fine-Tuning, which generates multiple candidate sequences, evaluates them using our reward models, and retains only the best sequences for supervised fine-tuning. We extended RAFT with our embedding level diversity regularization term.

### B.6.2  ProteinZero$_{\mathsf{GRPO}}$

Our adaptation of Group Relative Policy Optimization, which directly optimizes the policy using relative rewards within each batch. This was further enhanced with our embedding-level diversity regularization.

### B.7 Computational Efficiency and Potential Extensions to De Novo Design

A critical computational challenge in protein structure-conditioned generation stems from the runtime requirements of structural inference during reward computation. As shown in Tables 4 and 5, we comprehensively evaluate the wall-clock time necessary for reward generation across multiple structural prediction frameworks. For our inverse folding framework, which operates with predetermined backbone structures, ESMFold demonstrates substantial efficiency advantages, requiring only 18.7s and 47.5s for proteins in the 0-150 and 150-300 amino acid ranges, respectively. This represents a 26-87× acceleration compared to AlphaFold2, ColabFold, OpenFold, and AlphaFold3. The computational gap widens significantly when considering Multiple Sequence Alignment (MSA), which constitutes essential but time-intensive preprocessing for the AlphaFold family models. For thermal stability prediction, our Fast-ddG approach ($\sim$2s on GPU) achieves a 236-760× speedup over physics-based methods like FoldX. While our current implementation focuses on inverse folding with fixed backbones, these benchmarks establish important computational baselines for future extensions to de novo protein design tasks, where simultaneous optimization of sequence and structure would introduce additional complexity. Notably, as Table 5 demonstrates, our framework's reliance on ESMFold eliminates the computational burden of MSA search, a critical advantage for potential de novo applications where rapid structural evaluation is essential. De novo design presents different challenges, requiring not only the generation of applicable sequences but also the exploration of the vast conformational landscape to discover novel protein folds with targeted functional properties. This expanded search space would require efficient sampling strategies across both sequence and structural domains, while maintaining physically realistic conformations with proper hydrophobic packing, secondary structure formation, and domain-level architectural coherence. The computational efficiency gains demonstrated in our proxy reward models suggest that integrating lightweight structural prediction methods that avoid MSA requirements within a reinforcement learning framework could make online learning feasible even for these more complex design scenarios. The dramatic reduction in evaluation time enabled by our approach makes online reinforcement learning computationally tractable for current inverse folding tasks, while providing insights into the feasibility of extending this paradigm to full de novo design in future work.

Recent GPU-accelerated implementations combining optimized MSA generation and TensorRT-enhanced inference achieve over 130-fold speedups in structure prediction (Didi et al., 2025), suggesting that incorporating more sophisticated structural oracles into online RL frameworks may become computationally feasible in the near future.

## C Additional Experimental Results

### C.1 Independent Validation with AlphaFold3

While our evaluation pipeline employs external US-align to compute TM-score between ESMFold-predicted and target structures rather than relying on ESMFold's internal predicted TM-score (pTM, a confidence metric predicting the alignment quality of the folded structure), we sought to further strengthen our evaluation through comprehensive independent validation using AlphaFold3, the current state-of-the-art structure prediction model. This orthogonal assessment provides additional evidence that our performance improvements represent genuine advances in protein design capability and demonstrates the robustness of our approach across different structure prediction frameworks.

Table 2 presents designability metrics computed using both ESMFold (employed during training) and AlphaFold3 (independent evaluation) for all methods. The improvements observed through ESMFold evaluation are consistently corroborated by AlphaFold3 results. For 0-150 residue proteins, ProteinZero$_{\text{GRPO}}$ achieves 91.56% success rate with AlphaFold3 evaluation, maintaining its substantial advantage over baselines (InstructPLM: 86.98%, DPO: 88.12%, Multi-Round DPO: 88.71%). Similar patterns hold for 150-300 residue proteins, where ProteinZero$_{\text{GRPO}}$ reaches 92.27% success rate with AlphaFold3. Figure 4 provides qualitative examples of representative complex protein architectures evaluated with AlphaFold3, further illustrating the structural fidelity of our designed sequences.

The consistent improvements across both evaluation frameworks indicate that our approach learns design principles that transfer across computational structure predictors. While we selected ESMFold as our re-

Table 6: Supplementary experimental results exploring different hyperparameter configurations for ProteinZero. We evaluate the impact of KL divergence coefficients ($\alpha_{\mathrm{KL}}$) and diversity regularization ($\alpha_{\mathrm{div}}$) on both GRPO and RAFT algorithms across two protein size categories. Best results within each algorithm and size category are highlighted in blue .

| Length | Configuration | InverseFold Acc. Recovery Rate ↑ | Thermal Stability Metrics Fast-ddG ↓ | FoldX ddG ↓ | TM Score ↑ | Designability Metrics PLDDT ↑ | Diversity ↑ | scRMSD ↓ (scRMSD <2Å% ↑) | Overall Success (%) ↑ |
|---|---|---|---|---|---|---|---|---|---|
| | *Additional GRPO Results* | | | | | | | | |
| 0-150 residues | GRPO ($\alpha_{\mathrm{KL}}=0.04, \alpha_{\mathrm{div}}=0.05$) | 0.58 | -22.06 | -22.71 | 0.86 | 82.13 | 0.31 | 1.39 (93%) | 89% |
| | GRPO ($\alpha_{\mathrm{KL}}=0.04, \alpha_{\mathrm{div}}=0.00$) | 0.58 | -22.50 | -24.55 | 0.85 | 82.23 | 0.27 | 1.41 (90%) | 90% |
| | *Additional RAFT Results* | | | | | | | | |
| | RAFT ($\alpha_{\mathrm{KL}}=0.005, \alpha_{\mathrm{div}}=0.05$) | 0.58 | -21.63 | -21.18 | 0.84 | 80.93 | 0.30 | 1.41 (92%) | 88% |
| | RAFT ($\alpha_{\mathrm{KL}}=0.005, \alpha_{\mathrm{div}}=0.00$) | 0.58 | -21.81 | -21.72 | 0.84 | 80.97 | 0.28 | 1.42 (92%) | 88% |
| | RAFT ($\alpha_{\mathrm{KL}}=0.01, \alpha_{\mathrm{div}}=0.05$) | 0.58 | -22.12 | -22.95 | 0.85 | 81.14 | 0.30 | 1.40 (92%) | 89% |
| | RAFT ($\alpha_{\mathrm{KL}}=0.01, \alpha_{\mathrm{div}}=0.00$) | 0.59 | -22.18 | -22.98 | 0.84 | 81.28 | 0.28 | 1.42 (92%) | 89% |
| | RAFT ($\alpha_{\mathrm{KL}}=0.0, \alpha_{\mathrm{div}}=0.05$) | 0.58 | -21.70 | -21.50 | 0.85 | 81.08 | 0.30 | 1.40 (92%) | 87% |
| | RAFT ($\alpha_{\mathrm{KL}}=0.0, \alpha_{\mathrm{div}}=0.00$) | 0.58 | -22.03 | -22.73 | 0.84 | 81.23 | 0.28 | 1.41 (92%) | 87% |
| | *Additional GRPO Results* | | | | | | | | |
| 150-300 residues | GRPO ($\alpha_{\mathrm{KL}}=0.04, \alpha_{\mathrm{div}}=0.05$) | 0.58 | -39.53 | -31.95 | 0.86 | 83.98 | 0.33 | 1.42 (89%) | 90% |
| | GRPO ($\alpha_{\mathrm{KL}}=0.04, \alpha_{\mathrm{div}}=0.00$) | 0.57 | -40.40 | -32.15 | 0.85 | 84.05 | 0.29 | 1.43 (89%) | 90% |
| | *Additional RAFT Results* | | | | | | | | |
| | RAFT ($\alpha_{\mathrm{KL}}=0.005, \alpha_{\mathrm{div}}=0.05$) | 0.57 | -36.61 | -28.26 | 0.84 | 83.24 | 0.33 | 1.43 (89%) | 88% |
| | RAFT ($\alpha_{\mathrm{KL}}=0.005, \alpha_{\mathrm{div}}=0.00$) | 0.58 | -36.79 | -28.86 | 0.83 | 83.53 | 0.30 | 1.44 (88%) | 88% |
| | RAFT ($\alpha_{\mathrm{KL}}=0.01, \alpha_{\mathrm{div}}=0.05$) | 0.58 | -37.23 | -30.08 | 0.84 | 83.57 | 0.33 | 1.43 (89%) | 89% |
| | RAFT ($\alpha_{\mathrm{KL}}=0.01, \alpha_{\mathrm{div}}=0.00$) | 0.58 | -37.48 | -30.47 | 0.84 | 83.67 | 0.31 | 1.44 (88%) | 89% |
| | RAFT ($\alpha_{\mathrm{KL}}=0.0, \alpha_{\mathrm{div}}=0.05$) | 0.57 | -36.53 | -27.65 | 0.84 | 83.46 | 0.33 | 1.43 (89%) | 87% |
| | RAFT ($\alpha_{\mathrm{KL}}=0.0, \alpha_{\mathrm{div}}=0.00$) | 0.58 | -36.95 | -29.47 | 0.84 | 83.52 | 0.31 | 1.44 (88%) | 87% |

ward model for computational efficiency, the self-improved policies demonstrate robust performance when evaluated with AlphaFold3, confirming that ProteinZero discovers genuine improvements that transcend the specific choice of structure predictor used during training. The relative performance rankings remain unchanged across both evaluation methods: ProteinZero methods consistently outperform both offline RL baselines and the base model.

These results establish the methodological rigor required for reinforcement learning applications to protein design. The strong performance under AlphaFold3 evaluation indicates that our approach achieves consistent improvements across computational evaluation frameworks. Whether the learned policies generalize to practical applications remains to be established through prospective experimental validation.

## C.2 Hyperparameter Ablation Studies

Table 6 presents additional experimental results exploring different hyperparameter configurations for ProteinZero, specifically evaluating the impact of KL divergence coefficients ($\alpha_{\mathrm{KL}}$) and diversity regularization ($\alpha_{\mathrm{div}}$) on both ProteinZero$_{\mathrm{GRPO}}$ and ProteinZero$_{\mathrm{RAFT}}$ algorithms across two protein size categories (0-150 and 150-300 residues).

For ProteinZero$_{\mathrm{GRPO}}$, we test configurations with $\alpha_{\mathrm{KL}} = 0.04$ (the original GRPO setting) and varying diversity regularization ($\alpha_{\mathrm{div}} \in \{0.00, 0.05\}$). In the 0-150 residue category, the configuration with $\alpha_{\mathrm{div}} = 0.04, \alpha_{\mathrm{div}} = 0.05$ achieves recovery rate of 0.58, TM Score of 0.86, sequence diversity of 0.31, and overall success rate of 89%, while removing diversity regularization ($\alpha_{\mathrm{div}} = 0.00$) yields enhanced thermal stability (Fast-ddG: -22.50 vs -22.06, FoldX ddG: -24.55 vs -22.71) but significantly degraded sequence diversity (0.27 vs 0.31) and structural accuracy (TM Score: 0.85 vs 0.86), achieving 90% overall success rate. For 150-300 residues, both configurations reach 90% success rates, with $\alpha_{\mathrm{div}} = 0.05$ providing superior sequence diversity (0.33 vs 0.29) and designability metrics (TM Score: 0.86 vs 0.85).

For ProteinZero$_{\mathrm{RAFT}}$, we examine configurations with $\alpha_{\mathrm{KL}} \in \{0.0, 0.005, 0.01\}$ and $\alpha_{\mathrm{div}} \in \{0.00, 0.05\}$. In the 0-150 residue category, the best performing configuration ($\alpha_{\mathrm{KL}} = 0.01, \alpha_{\mathrm{div}} = 0.00$) achieves recovery rate of 0.59, thermal stability of Fast-ddG: -22.18 and FoldX ddG: -22.98, and 89% overall success rate. Weaker KL regularization with $\alpha_{\mathrm{KL}} = 0.005$ consistently underperforms (88% success rate), while completely removing KL constraints ($\alpha_{\mathrm{KL}} = 0.0$) further degrades performance to 87% success rate. For 150-300 residues, similar patterns emerge with $\alpha_{\mathrm{KL}} = 0.01$ configurations achieving 89% success rates compared to 88% for

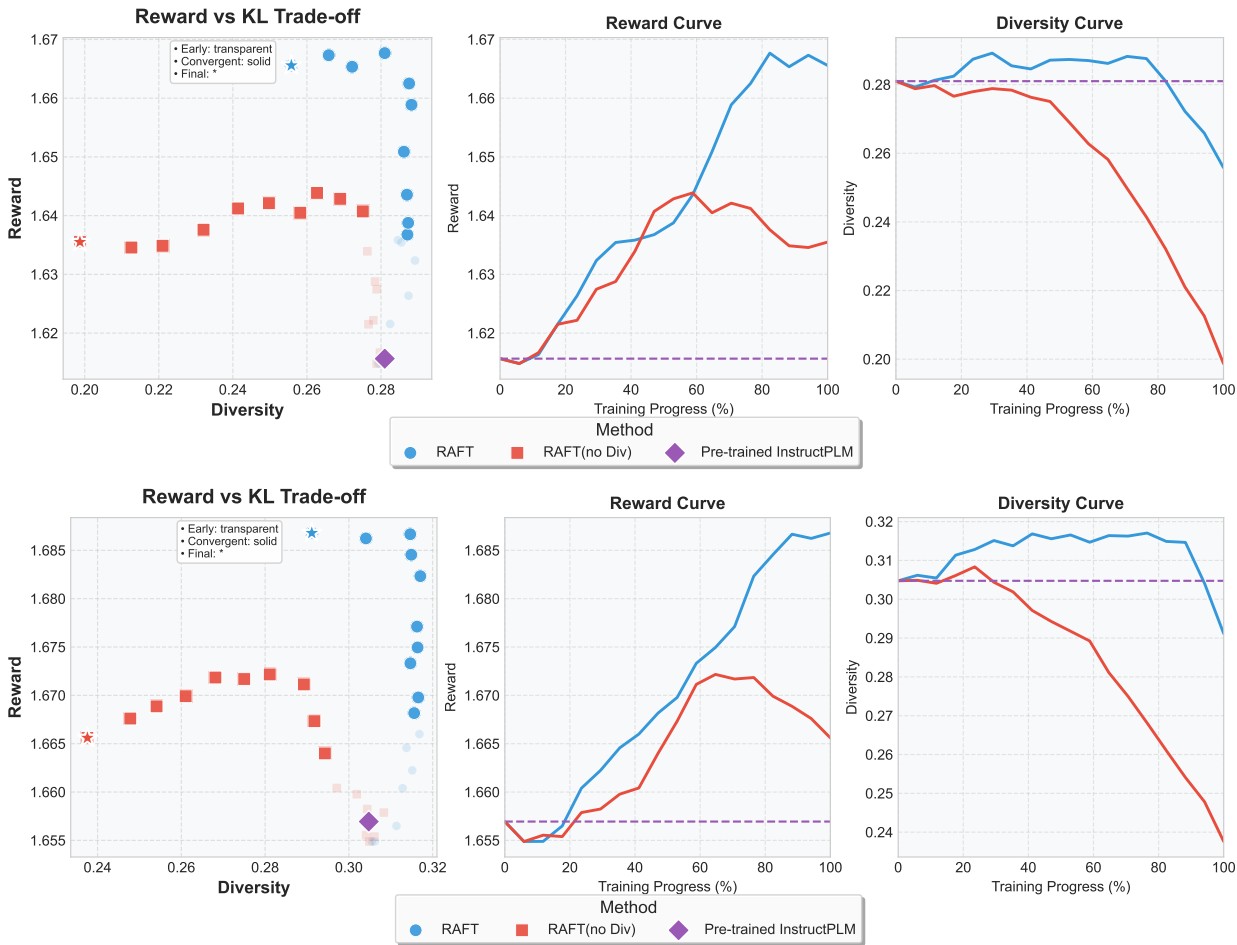

Figure 7: Training dynamics of ProteinZero$_{\mathrm{RAFT}}$ across protein size categories. **Top**: 0-150 residue proteins. **Bottom**: 150-300 residue proteins. Each row shows: (**Left**) Reward-diversity trade-off demonstrating Pareto frontier between final reward and sequence diversity. (**Middle**) Evolution of reward throughout training, showing consistent improvement over InstructPLM baseline. (**Right**) Diversity trajectory revealing how our novel embedding-level diversity regularization $\mathcal{L}_{\mathrm{Div}}$ maintains higher sequence diversity compared to RAFT without this regularization (no div).

$\alpha_{\mathrm{KL}} = 0.005$ and 87% for $\alpha_{\mathrm{KL}} = 0.0$. Importantly, removing diversity regularization consistently reduces sequence diversity across all configurations.

Despite these extensive explorations, all configurations in Table 6 underperform our optimal settings reported in Table 1, where $\alpha_{\mathrm{KL}} = 0.1$ and $\alpha_{\mathrm{div}} = 0.05$ achieve superior results: ProteinZero$_{\mathrm{GRPO}}$ reaches 90.13% and 91.19% overall success rates for 0-150 and 150-300 residues respectively, while ProteinZero$_{\mathrm{RAFT}}$ achieves 89.29% and 89.36%. These results demonstrate that stronger KL regularization and our embedding-level diversity regularization are essential for optimal protein design performance.

## C.3 Training Dynamics and Convergence Analysis

Figure 7 and Figure 8 present comprehensive training dynamics for ProteinZero across different protein size categories, revealing critical insights about online reinforcement learning in protein design and the broader implications for mitigating mode collapse in RLHF systems.

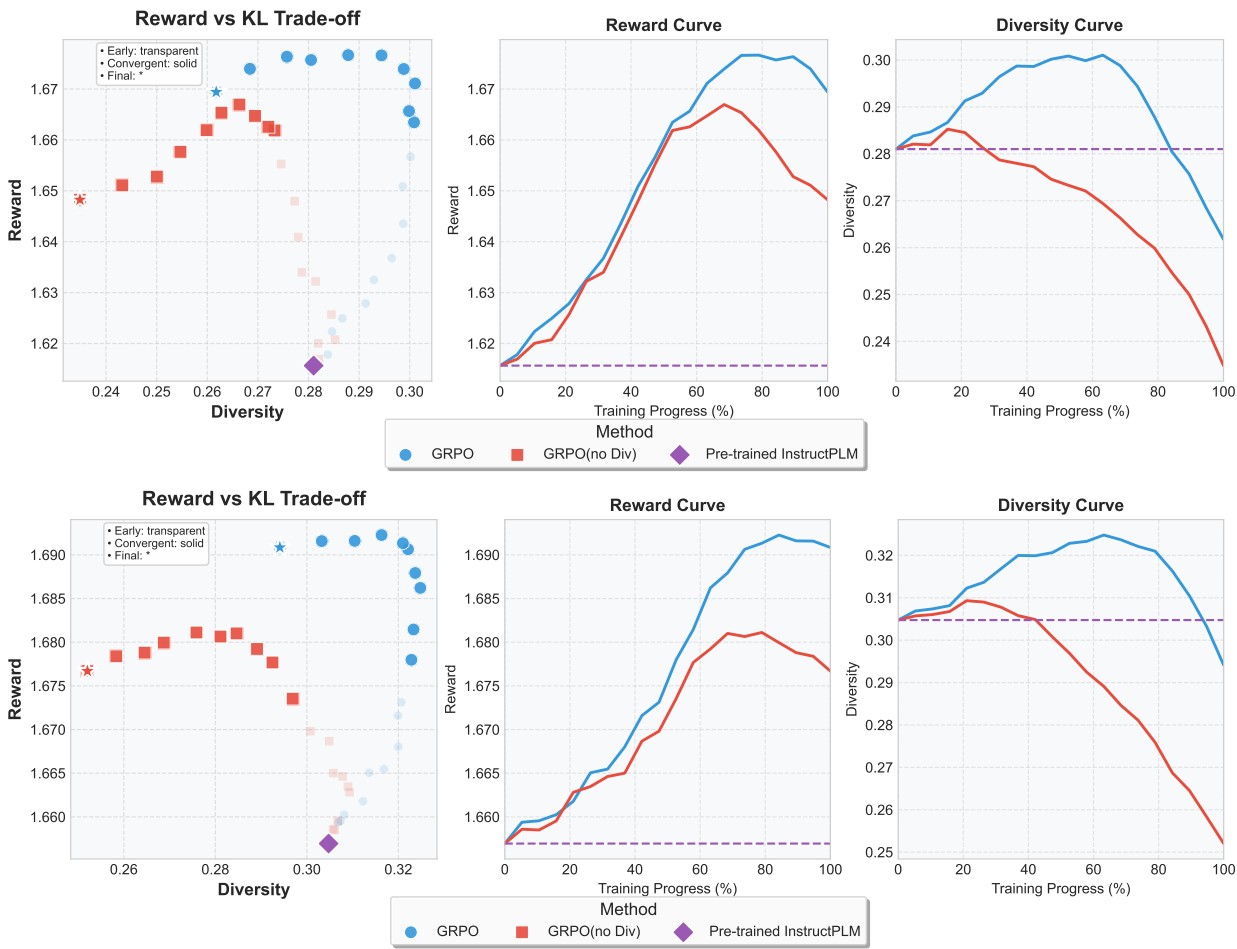

Figure 8: Training dynamics of ProteinZero$_{\text{GRPO}}$ across protein size categories. **Top**: 0-150 residue proteins. **Bottom**: 150-300 residue proteins. Each row shows: (**Left**) Reward-diversity trade-off demonstrating Pareto frontier between final reward and sequence diversity. (**Middle**) Evolution of reward throughout training, showing consistent improvement over InstructPLM baseline. (**Right**) Diversity trajectory revealing how our novel embedding-level diversity regularization $\mathcal{L}_{\text{Div}}$ maintains higher sequence diversity compared to GRPO without this regularization (no div).

Figure 9 further reports the seed-averaged training trajectories of ProteinZero$_{\text{GRPO}}$ over 10 independent training runs, separately for 0–150 and 150–300 residue proteins, with shaded regions indicating $\pm 1$ standard deviation across seeds and the horizontal axis labeled in training steps. Training reward rises rapidly during the early steps and reaches a plateau by approximately step 1300 (0–150 residues) and step 1500 (150–300 residues), after which it oscillates within a narrow band around its peak; the magnitude of this late-stage fluctuation is small, with the final mean reward at most 1.19% below the peak value for 0–150 residues and at most 0.51% below for 150–300 residues. The success-rate trajectory (right column) reaches a stable plateau over the same period and does not degrade in the final steps; across the 10 seeds, the final success rate has a standard deviation of 0.06 percentage points on 0–150 residues and 0.07 percentage points on 150–300 residues, indicating that the training behavior is reproducible across runs rather than seed-dependent.

The training trajectories demonstrate a fundamental challenge in online RL: without explicit diversity maintenance, policies consistently collapse toward narrow, high-reward regions of the solution space. As shown in the diversity curves (right panels of both figures), standard RAFT and GRPO without our diversity regularization $\mathcal{L}_{\text{Div}}$ exhibit monotonic diversity decline, with sequence diversity dropping from initial values of 0.28-0.30 to as low as 0.13-0.18 by iteration 20. This represents a 40-55% reduction in exploration capacity,

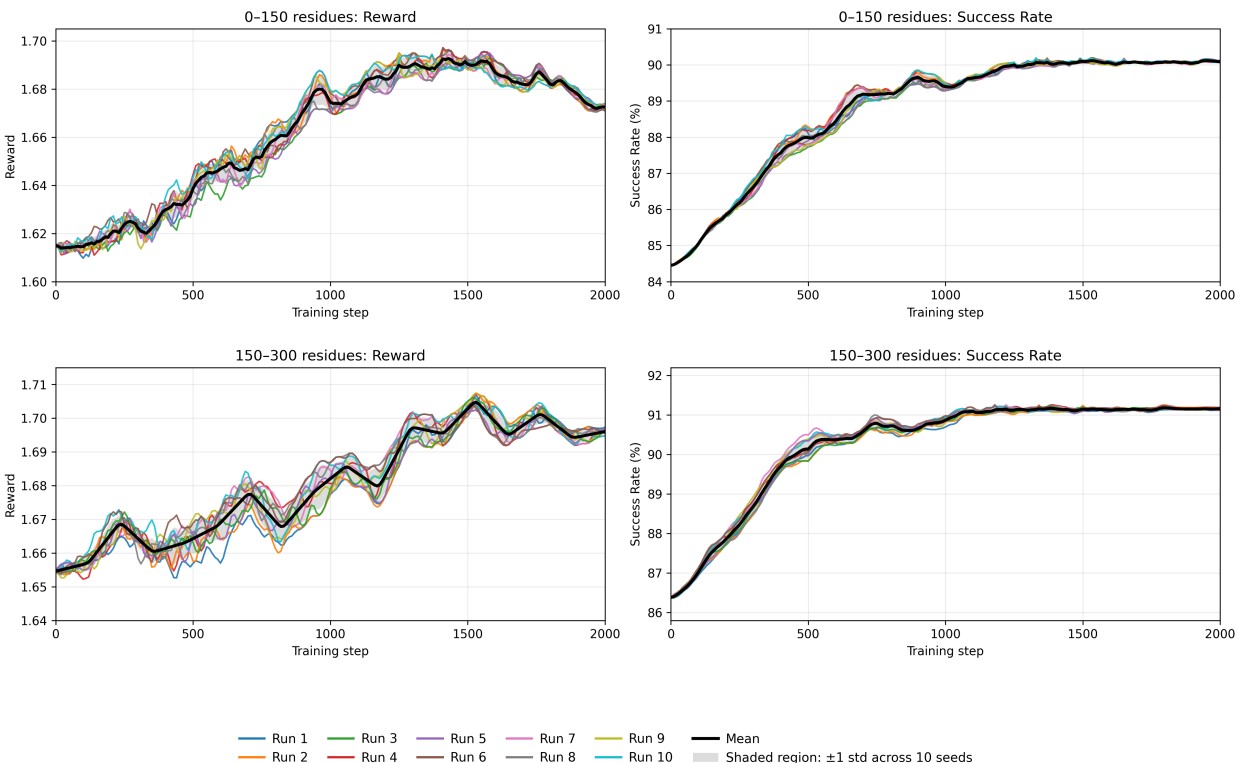

Figure 9: Training-dynamics summary across 10 independent training seeds, showing the evolution of training reward (left column) and held-out test-set success rate (right column) for ProteinZero$_{\text{GRPO}}$ on 0–150 residue proteins (top row) and 150–300 residue proteins (bottom row). In each panel, the solid line traces the seed-averaged trajectory and the shaded region indicates $\pm 1$ standard deviation across the 10 seeds; the x-axis is labeled in training steps. Success rate is defined as the proportion of designs satisfying scRMSD $< 2\,\text{Å}$ and FoldX ddG $< 0$ on the held-out test set. Training reward rises rapidly during the early steps and reaches a plateau by approximately step 1300 (0–150 residues) and step 1500 (150–300 residues), after which it oscillates within a narrow band around its peak. Success rate reaches a stable plateau over the same period.

severely limiting the model's ability to discover novel solutions. In contrast, incorporating our embedding-level diversity regularization maintains sequence diversity above 0.20-0.26 throughout training, preserving 70-85% of initial exploration capacity while still achieving comparable or superior reward values. For 150-300 residue proteins, where the design space is exponentially larger, this effect becomes even more pronounced: models with $\mathcal{L}_{\text{Div}}$ maintain diversity levels of 0.21-0.26 compared to 0.17-0.23 without regularization.

The reward-diversity trade-off plots (left panels) reveal that maintaining diversity through $\mathcal{L}_{\text{Div}}$ creates a favorable Pareto frontier where both high rewards and sequence variety are preserved. This sustained exploration capability translates directly to performance gains. Examining the reward curves (middle panels), ProteinZero with diversity regularization demonstrates more robust convergence, reaching final rewards of 1.644-1.686 for 0-150 residues and 1.686-1.688 for 150-300 residues, compared to more variable performance without regularization. The preservation of diversity enables the model to continue discovering improved solutions rather than prematurely converging to local optima. This phenomenon is particularly evident in iterations 15-20, where models without diversity regularization show reward stagnation or decline (e.g., RAFT dropping from 1.648 to 1.626 for 0-150 residues), while regularized models maintain steady improvement or stability.

These findings extend beyond protein design to general online reinforcement learning from human feedback. Mode collapse represents a critical failure mode in RLHF where policies converge to narrow behavioral pat-

Table 7: Per-protein performance of the Fast-ddG surrogate on Ssym dataset (342 single-point direct mutations, wild-type→mutant direction). We report the number of mutations per PDB ($n_{\text{mut}}$), per-protein RMSE (kcal/mol), and PCC for the pretrained and fine-tuned Fast-ddG variants. Best results highlighted in  blue .

| PDB | $n_{\text{mut}}$ | RMSE (kcal/mol) ↓ | | | PCC ↑ | | |
|---|---|---|---|---|---|---|---|
| | | Pretrained | Fast-ddG only | TM-score + Fast-ddG | Pretrained | Fast-ddG only | TM-score + Fast-ddG |
| 1L63 | 118 | 1.36 | 1.30 | 1.33 | 0.58 | 0.66 | 0.60 |
| 2LZM | 66 | 1.16 | 1.13 | 1.13 | 0.74 | 0.75 | 0.75 |
| 1LZ1 | 61 | 1.22 | 1.12 | 1.18 | 0.76 | 0.78 | 0.79 |
| 1BNI | 13 | 1.09 | 1.03 | 1.25 | 0.62 | 0.70 | 0.59 |

Table 8: Performance comparison on the Ssym dataset (342 single-point direct mutations, wild-type→mutant direction). Lower RMSE and higher Pearson correlation coefficient (PCC) indicate better agreement with experimental $\Delta\Delta G$ values. Best ProteinZero result highlighted in  blue .

| Model | RMSE (kcal/mol) ↓ | PCC ↑ |
|---|---|---|
| ProteinMPNN | 3.38 | 0.26 |
| ThermoMPNN | 1.12 | 0.72 |
| Rosetta | 2.31 | 0.69 |
| FoldX | 1.56 | 0.63 |
| ThermoNet | 1.56 | 0.47 |
| PROSTATA | 1.42 | 0.51 |
| ProteinZero (pretrained) | 1.47 | 0.60 |
| ProteinZero (TM-score + Fast-ddG) | 1.45 | 0.61 |
| ProteinZero (Fast-ddG only) | 1.44 | 0.62 |

terns that maximize immediate rewards but sacrifice long-term adaptability and robustness. Our embedding-level diversity regularization offers a principled solution by operating directly in the latent representation space, encouraging exploration of functionally distinct regions rather than merely surface-level variations. The consistent effectiveness across both RAFT and GRPO algorithms, and across different protein size categories, suggests that this approach addresses a fundamental limitation in online RL optimization.

By maintaining a balance between exploitation (achieving high rewards) and exploration (preserving diversity), our method enables continuous learning and adaptation, essential properties for developing robust, generalizable AI systems. The quantitative results demonstrate that diversity-aware online RL achieves 89-91% success rates while maintaining 2-3× higher sequence diversity compared to non-regularized variants. This simultaneous improvement in both performance and exploration capacity validates that preventing mode collapse through embedding-level regularization is not merely a theoretical benefit but translates to concrete gains in practical applications.

### C.4   Experimental Validation of Fast-ddG on Ssym Benchmark

This section validates Fast-ddG accuracy on experimental thermodynamic measurements from the Ssym benchmark (Pucci et al., 2018), which comprises 684 single-point mutations across multiple protein families with calorimetrically determined $\Delta\Delta G$ values and crystal structures. Following Eq. 4, we evaluate 342 wild-type→mutant transitions by computing stability changes on wild-type backbone geometries.

Table 8 compares our predictor against physics-based oracles (FoldX, Rosetta) and supervised predictors (ThermoMPNN (Dieckhaus et al., 2024), ThermoNet (Li et al., 2020), PROSTATA (Umerenkov et al., 2022)). Across three configurations (pretrained, Fast-ddG-only, TM-score + Fast-ddG), we achieve RMSE 1.44–1.47 kcal/mol and PCC 0.60–0.62, matching FoldX (RMSE: 1.56, PCC: 0.63) at 236–760× speedup (Tables 4–5), which is a 56% RMSE improvement over ProteinMPNN (3.38 kcal/mol, PCC: 0.26). Ther-

Table 9: Representative mutations from the Ssym dataset demonstrating improved prediction accuracy after fine-tuning. Error denotes absolute deviation $|\widehat{\Delta\Delta G} - \Delta\Delta G_{\text{exp}}|$ in kcal/mol. Best results for each mutation highlighted in blue.

| PDB | Mutation | Experimental $\Delta\Delta G$ (kcal/mol) | Prediction Error (kcal/mol) ↓ Pretrained | Fast-ddG only | TM-score + Fast-ddG |
|---|---|---|---|---|---|
| 1CEY | D12A | 2.50 | 3.64 | 1.20 | 1.32 |
| 1LZ1 | V2G | −2.29 | 3.54 | 2.09 | 1.56 |
| 1LZ1 | I23A | −2.50 | 1.70 | 0.60 | 0.64 |
| 1L63 | V149A | −3.20 | 1.49 | 0.50 | 0.49 |
| 1L63 | A98V | −3.20 | 1.52 | 0.61 | 0.18 |
| 1L63 | D20A | −0.30 | 3.55 | 2.64 | 1.66 |
| 1LZ1 | V2A | −1.50 | 2.31 | 1.09 | 1.40 |
| 5PTI | N43G | −5.70 | 3.09 | 2.16 | 2.29 |
| 1IOB | T9G | −2.60 | 1.78 | 1.04 | 0.35 |
| 1BNI | T26A | −1.70 | 1.25 | 0.53 | 0.39 |
| 1L63 | S44R | 0.20 | 1.38 | 0.67 | 0.33 |
| 1BNI | I76A | −1.70 | 0.94 | 0.31 | 0.26 |
| 1VQB | V35I | −0.60 | 0.83 | 0.22 | 0.07 |
| 1L63 | A42V | −2.70 | 2.37 | 1.77 | 0.52 |
| 4LYZ | T40S | −0.30 | 1.45 | 0.23 | 0.90 |
| 1VQB | I47M | −1.70 | 1.15 | 0.61 | 0.62 |
| 1L63 | L46A | −1.90 | 1.33 | 0.68 | 0.81 |
| 1VQB | I47L | −0.40 | 0.91 | 0.40 | 0.35 |
| 2RN2 | D70N | 0.90 | 2.40 | 1.90 | 1.60 |
| 1L63 | I27M | −3.10 | 1.23 | 0.76 | 0.74 |

Table 10: Extended version of Table 1 including wild-type and generated folding energies. Performance comparison of protein inverse folding methods on CATH-4.3 benchmark proteins grouped by length (0-150 and 150-300 residues). Metrics include sequence recovery, thermal stability (Fast-ddG, absolute folding energies, FoldX ddG), and designability (TM-score, pLDDT, diversity, scRMSD). Success rate is defined as scRMSD < 2Å and FoldX ddG < 0. Designability metrics computed using ESMFold; independent AlphaFold3 validation confirms consistent trends (Table 2). Best results highlighted in blue, second-best in green.

| Length | Method | InverseFold Acc. Recovery Rate ↑ | Thermal Stability Metrics Fast-ddG ↓ | WT Energy | Gen. Energy ↓ | FoldX ddG ↓ | Designability Metrics TM Score ↑ | PLDDT ↑ | Diversity ↑ | scRMSD ↓ (<2Å% ↑) | Overall Success (%) ↑ |
|---|---|---|---|---|---|---|---|---|---|---|---|
| | | | | | *Base Model* | | | | | | |
| 0-150 residues | InstructPLM | 0.574 | -21.543 | 27.09 | 6.21 | -20.878 | 0.812 | 79.983 | 0.281 | 1.484 (85.71%) | 84.45% |
| | | | | | *SOTA Inverse Folding Models* | | | | | | |
| | ProteinMPNN | 0.426 | -21.509 | 27.09 | 6.30 | -20.792 | 0.805 | 79.883 | 0.280 | 1.500 (82.14%) | 81.95% |
| | ESM-IF | 0.377 | -17.900 | 27.09 | 12.76 | -14.328 | 0.802 | 78.918 | 0.263 | 1.515 (81.25%) | 80.71% |
| | | | | | *RL Baseline Methods* | | | | | | |
| | DPO | 0.571 | -21.713 | 27.09 | 5.90 | -21.191 | 0.820 | 80.716 | 0.274 | 1.473 (87.58%) | 86.44% |
| | Multi-Round DPO | 0.569 | -21.797 | 27.09 | 5.67 | -21.423 | 0.823 | 80.797 | 0.266 | 1.468 (87.95%) | 86.89% |
| | | | | | *Our Online RL Methods* | | | | | | |
| | ProteinZero$_{\text{RAFT}}$ (Ours) | 0.587 | -22.236 | 27.09 | 3.92 | -23.168 | 0.849 | 81.560 | 0.296 | 1.393 (92.86%) | 89.29% |
| | ProteinZero$_{\text{GRPO}}$ (Ours) | 0.590 | -22.616 | 27.09 | 2.17 | -24.924 | 0.867 | 82.326 | 0.306 | 1.373 (93.55%) | 90.13% |
| | | | | | *Base Model* | | | | | | |
| 150-300 residues | InstructPLM | 0.570 | -36.362 | 36.96 | 9.82 | -27.145 | 0.824 | 83.783 | 0.305 | 1.448 (88.24%) | 86.38% |
| | | | | | *SOTA Inverse Folding Models* | | | | | | |
| | ProteinMPNN | 0.405 | -35.778 | 36.96 | 9.90 | -27.057 | 0.816 | 82.361 | 0.297 | 1.469 (86.64%) | 84.67% |
| | ESM-IF | 0.446 | -32.125 | 36.96 | 12.14 | -24.816 | 0.802 | 82.042 | 0.279 | 1.487 (86.09%) | 82.81% |
| | | | | | *RL Baseline Methods* | | | | | | |
| | DPO | 0.570 | -36.417 | 36.96 | 8.05 | -28.915 | 0.830 | 83.837 | 0.296 | 1.441 (88.97%) | 87.70% |
| | Multi-Round DPO | 0.569 | -36.483 | 36.96 | 7.87 | -29.087 | 0.831 | 83.840 | 0.288 | 1.437 (89.04%) | 88.05% |
| | | | | | *Our Online RL Methods* | | | | | | |
| | ProteinZero$_{\text{RAFT}}$ (Ours) | 0.578 | -37.575 | 36.96 | 6.21 | -30.755 | 0.841 | 83.850 | 0.324 | 1.427 (89.17%) | 89.36% |
| | ProteinZero$_{\text{GRPO}}$ (Ours) | 0.580 | -40.626 | 36.96 | 4.16 | -32.805 | 0.862 | 84.154 | 0.331 | 1.393 (90.43%) | 91.19% |

moMPNN achieves superior performance (RMSE: 1.12, PCC: 0.72) but requires supervised training and

handles only single-residue perturbations, whereas our unsupervised predictor generalizes to multi-mutation redesigns often exceeding 50% sequence divergence.

Table 9 reports errors for 20 representative mutations across eight families (1CEY, 1LZ1, 1L63, 5PTI, 1IOB, 1BNI, 1VQB, 4LYZ, 2RN2), spanning experimental $\Delta\Delta G$ from $-5.70$ to $+2.50$ kcal/mol. Fine-tuning consistently reduces errors: 1L63 A98V improves from 1.52 to 0.18 kcal/mol; 1VQB V35I from 0.83 to 0.07 kcal/mol. This consistency across diverse targets suggests Fast-ddG captures thermodynamic signal that generalizes across these experimentally characterized targets.

These results demonstrate that Fast-ddG, though unsupervised and self-derived and optimized for full-sequence inverse folding, achieves physics-based accuracy on experimental data while maintaining computational efficiency for online RL.

## C.5 Complete Performance Metrics with Absolute Folding Energies

This section provides extended performance metrics complementing Table 1 in the main text. Table 10 presents the complete evaluation including wild-type and generated absolute folding energies, offering deeper insights into thermodynamic stability improvements.

The wild-type (WT) energy represents the average FoldX folding free energy of native structures in each length category: 27.09 kcal/mol for 0-150 residues and 36.96 kcal/mol for 150-300 residues. Generated energy denotes the average absolute folding free energy of designed sequences computed by FoldX. The FoldX ddG column reports the stability change relative to wild-type: ddG = Generated Energy − WT Energy. More negative ddG values indicate enhanced thermodynamic stability relative to native sequences.

ProteinZero achieves substantial predicted stability improvements across both length categories. For 0-150 residues, ProteinZeroGRPO reduces generated energy from 6.21 kcal/mol (InstructPLM baseline) to 2.17 kcal/mol, corresponding to FoldX ddG improvement from -20.878 to -24.924 kcal/mol, a 4.05 kcal/mol enhancement (19.4% relative improvement). For 150-300 residues, generated energy decreases from 9.82 to 4.16 kcal/mol, yielding FoldX ddG improvement from -27.145 to -32.805 kcal/mol, a 5.66 kcal/mol enhancement (20.8% relative improvement). These gains demonstrate that online RL with Fast-ddG optimization transfers effectively to independent physics-based oracles, suggesting that our framework learns thermodynamic signal that transfers across computational oracles rather than overfitting to training proxies.

## C.6 Fine-grained structural features under AlphaFold3

Beyond Table 2's AlphaFold3 cross-oracle validation, this section examines fine-grained structural features computed by refolding every designed sequence with AlphaFold3 (Abramson et al., 2024) and comparing with the target backbone.

The cross-oracle comparisons in this section and in Sections C.7, C.8, C.9, and C.10 share a common evaluation protocol. All comparisons against the InstructPLM base (Qiu et al., 2024) are performed on the same CATH-4.3 held-out test backbones used in the main paper, with both length ranges (0–150 and 150–300 residues). While the main-text results aggregate all designs across the held-out test backbones into a single scalar per metric (for example, each TM-score and FoldX ddG value in Table 1 is one number per model, computed across all test backbones and all 64 designs per backbone) and therefore average away the per-backbone behavior, the analyses in these subsections are instead computed per backbone to make this behavior visible. For each backbone, we pool 640 designs from ProteinZero$_{GRPO}$ (10 independent training runs $\times$ 64 designs each) and a matched set of 640 designs from the InstructPLM base (10 sampling seeds $\times$ 64 designs each), with both models sampled at the test-set evaluation setting (temperature 0.15, top-$p$ 0.9). Each per-backbone metric in Tables 11, 12, 13, 14, and 15 is reported as both mean $\pm$ standard deviation and median $[Q_1, Q_3]$ across backbones, so that the central tendency and dispersion of the per-backbone distribution are both visible. Significance is assessed by the two-sided paired Wilcoxon signed-rank test (Wilcoxon, 1945) applied at the backbone level on per-backbone means; the unit of inference is the backbone, so pooling across runs increases the per-backbone sample size without changing the level of the test. The test is rank-based and therefore consistent with the median view.

Table 11: Cross-oracle structural metrics under AlphaFold3 refolding, reported as mean $\pm$ standard deviation (top line) and median $[Q_1, Q_3]$ (bottom line) across all evaluation backbones in each length range; $p$ values are from the two-sided paired Wilcoxon signed-rank test applied to per-backbone means. In this and subsequent appendix tables, $\uparrow$ marks metrics for which higher values are typically preferred, $\downarrow$ marks metrics for which lower values are typically preferred, and $\approx$ marks metrics for which values closer to native are typically preferred.

| Metric | 0–150 residues | | | 150–300 residues | | |
| --- | --- | --- | --- | --- | --- | --- |
| | InstructPLM | ProteinZero$_{\text{GRPO}}$ | $p$ | InstructPLM | ProteinZero$_{\text{GRPO}}$ | $p$ |
| Global RMSD (Å) $\downarrow$ | $6.20 \pm 5.07$ 4.49 [2.08–9.13] | $4.57 \pm 4.43$ 2.65 [1.16–6.43] | $< 0.001$ | $6.04 \pm 6.83$ 3.13 [1.41–7.66] | $5.12 \pm 5.60$ 2.51 [1.22–6.93] | $< 0.001$ |
| Mean Local RMSD (Å) $\downarrow$ | $5.37 \pm 4.67$ 3.90 [1.57–7.64] | $3.84 \pm 4.04$ 2.10 [0.808–5.16] | $< 0.001$ | $4.78 \pm 5.88$ 2.22 [0.906–5.48] | $3.92 \pm 4.66$ 1.77 [0.810–4.91] | $< 0.001$ |
| Helix RMSD (Å) $\downarrow$ | $5.07 \pm 4.55$ 3.69 [1.19–7.39] | $3.61 \pm 4.10$ 1.70 [0.715–4.48] | $< 0.001$ | $4.70 \pm 5.75$ 2.30 [0.818–5.74] | $3.87 \pm 4.56$ 1.81 [0.742–5.44] | $< 0.001$ |
| Sheet RMSD (Å) $\downarrow$ | $3.94 \pm 4.17$ 2.36 [0.941–5.27] | $2.62 \pm 3.10$ 1.38 [0.529–3.63] | $< 0.001$ | $3.30 \pm 4.14$ 1.45 [0.674–3.41] | $2.64 \pm 3.30$ 1.13 [0.571–2.92] | $< 0.001$ |
| Loop RMSD (Å) $\downarrow$ | $6.53 \pm 5.40$ 5.81 [1.98–10.14] | $4.77 \pm 4.58$ 3.03 [1.22–6.37] | $< 0.001$ | $5.57 \pm 6.88$ 2.54 [1.21–6.57] | $4.68 \pm 5.62$ 2.19 [1.01–6.39] | $< 0.001$ |
| Number of H-bonds $\uparrow$ | $106.36 \pm 43.07$ 99.55 [75.52–127.94] | $108.91 \pm 44.98$ 100.94 [75.16–130.50] | $< 0.001$ | $252.03 \pm 105.86$ 240.35 [169.31–328.53] | $256.27 \pm 110.13$ 248.22 [172.75–332.63] | $< 0.001$ |
| H-bond Retention $\uparrow$ | $0.786 \pm 0.181$ 0.854 [0.685–0.917] | $0.853 \pm 0.142$ 0.910 [0.808–0.952] | $< 0.001$ | $0.829 \pm 0.142$ 0.877 [0.782–0.931] | $0.857 \pm 0.120$ 0.905 [0.830–0.939] | $< 0.001$ |
| Number of Salt Bridges $\uparrow$ | $1.40 \pm 1.12$ 1.02 [0.652–1.84] | $1.56 \pm 1.20$ 1.22 [0.773–2.05] | $< 0.001$ | $6.03 \pm 3.81$ 4.88 [3.68–7.45] | $6.38 \pm 3.82$ 5.52 [4.06–7.82] | $< 0.001$ |
| Salt Bridge Retention $\uparrow$ | $0.381 \pm 0.395$ 0.203 [0.016–0.947] | $0.402 \pm 0.380$ 0.273 [0.059–0.676] | $< 0.001$ | $0.350 \pm 0.250$ 0.330 [0.160–0.501] | $0.396 \pm 0.258$ 0.366 [0.187–0.561] | $< 0.001$ |
| Hydrophobic Core Size $\uparrow$ | $11.51 \pm 7.92$ 10.94 [4.90–18.24] | $12.13 \pm 8.12$ 11.99 [5.13–18.54] | $< 0.001$ | $58.72 \pm 30.87$ 51.49 [35.81–81.93] | $59.95 \pm 30.62$ 53.02 [36.53–83.06] | $< 0.001$ |
| Buried Hydrophobic Fraction $\uparrow$ | $0.546 \pm 0.234$ 0.612 [0.361–0.730] | $0.551 \pm 0.256$ 0.629 [0.401–0.760] | $0.357$ | $0.642 \pm 0.087$ 0.635 [0.587–0.699] | $0.648 \pm 0.086$ 0.644 [0.596–0.700] | $0.011$ |

C$\alpha$ RMSD is computed with Biopython's Superimposer (Cock et al., 2009); secondary structure is assigned with mkdssp (Hekkelman et al., 2025), the reference implementation of the DSSP algorithm (Kabsch & Sander, 1983); hydrogen bonds are defined as backbone N–O contacts within 3.5 Å (adjacent residues excluded); salt bridges as contacts within 4.0 Å between positively (Lys, Arg, His) and negatively (Asp, Glu) charged side chains; and the hydrophobic core as the set of buried hydrophobic residues with SASA $< 25$ Å$^2$ via the Shrake–Rupley algorithm (Shrake & Rupley, 1973). Results for the eleven resulting metrics are reported in Table 11, visualized as paired effect sizes in Figure 10, and as per-backbone box plots in Figure 11.

For several of these metrics the mean and median differ appreciably (Global RMSD in the 0–150 range, for instance, has mean 6.20 Å but median 4.49 Å under InstructPLM, with a similar pattern under ProteinZero), so the median and IQR are reported alongside mean $\pm$ std as a complementary view of the same per-backbone distribution. Median Global RMSD decreases from 4.49 Å to 2.65 Å in the 0–150 range and from 3.13 Å to 2.51 Å in the 150–300 range, with comparable median-level reductions across the per-secondary-structure RMSDs (Helix: $3.69 \rightarrow 1.70$ Å; Sheet: $2.36 \rightarrow 1.38$ Å; Loop: $5.81 \rightarrow 3.03$ Å in the 0–150 range, with smaller same-direction shifts in the 150–300 range). For these RMSD metrics, both $Q_1$ and $Q_3$ are also lower under ProteinZero in both length ranges, indicating that the median-level shift is mirrored by a shift in the surrounding quartiles. A consistent direction appears in the retention metrics: median H-bond retention rises from 0.854 to 0.910 in the 0–150 range and from 0.877 to 0.905 in the 150–300 range, and median salt-bridge retention rises in both length ranges. The count-based metrics (number of hydrogen bonds, number of salt bridges, and hydrophobic core size) shift only slightly at the median level. The observed differences are consistent in direction across both length ranges for the directional metrics, with statistically significant improvements ($p < 0.001$) on the RMSD, hydrogen-bond, salt-bridge, and core-size metrics in both length ranges; the Buried Hydrophobic Fraction differs significantly only in the 150–300 range ($p = 0.011$).

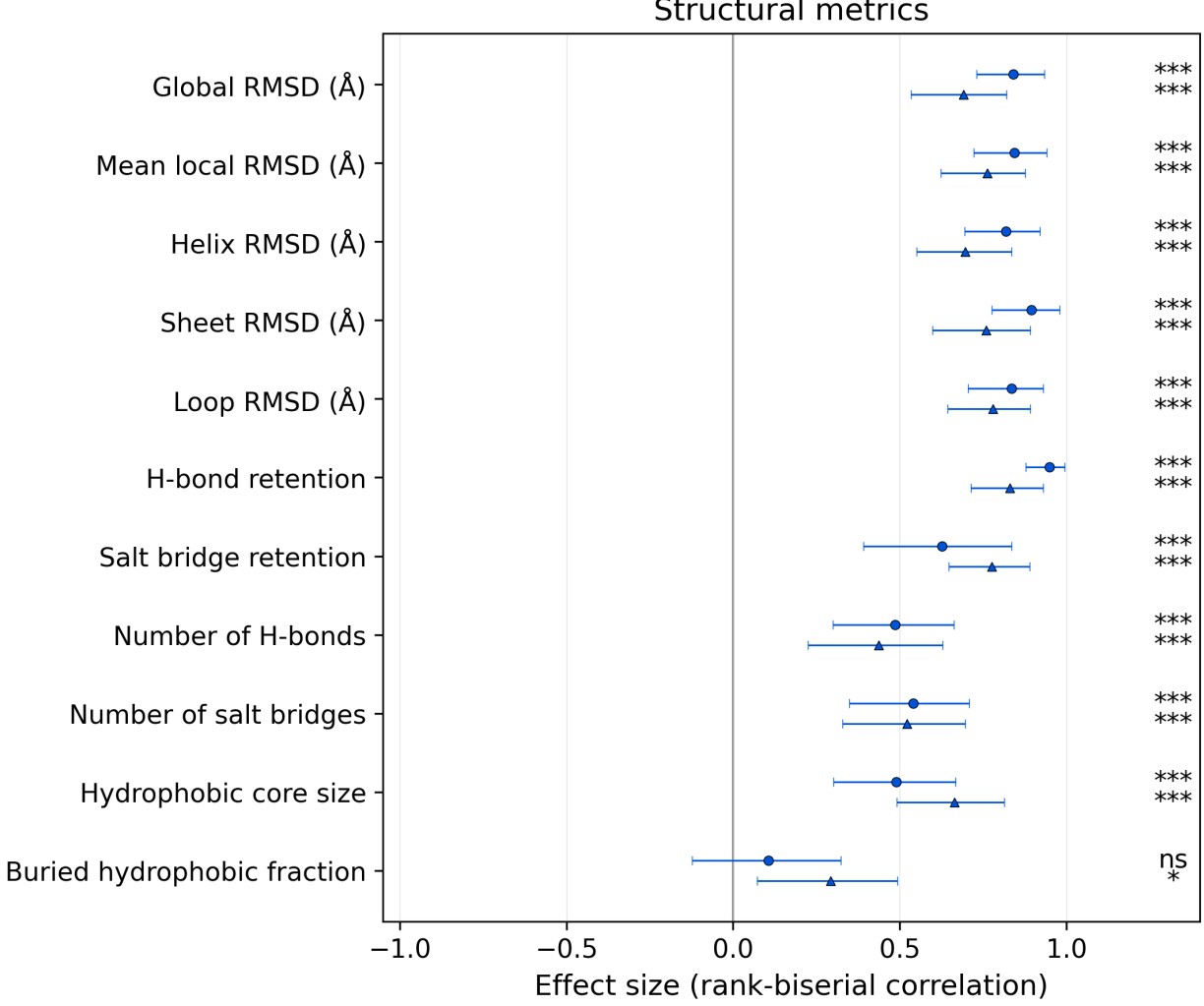

Figure 10: Forest plot comparing ProteinZero$_{\text{GRPO}}$ to InstructPLM on eleven structural metrics computed from AlphaFold3-refolded designs, for 0–150 and 150–300 residue proteins. For each metric, the rank-biserial correlation effect size is computed from per-backbone means, with positive values indicating that ProteinZero$_{\text{GRPO}}$ outperforms InstructPLM. Markers denote effect-size estimates (circles: 0–150; triangles: 150–300), with horizontal bars giving bootstrap 95% confidence intervals (1,000 resamples). Significance is assessed with the two-sided paired Wilcoxon signed-rank test: $^*p < 0.05$, $^{**}p < 0.01$, $^{***}p < 0.001$, ns = not significant. ProteinZero$_{\text{GRPO}}$ shows statistically significant differences favoring its designs on most metrics in both length ranges; these are computational measurements consistent with structural gains transferring across an independent folding oracle, but do not by themselves establish experimental structural correctness.

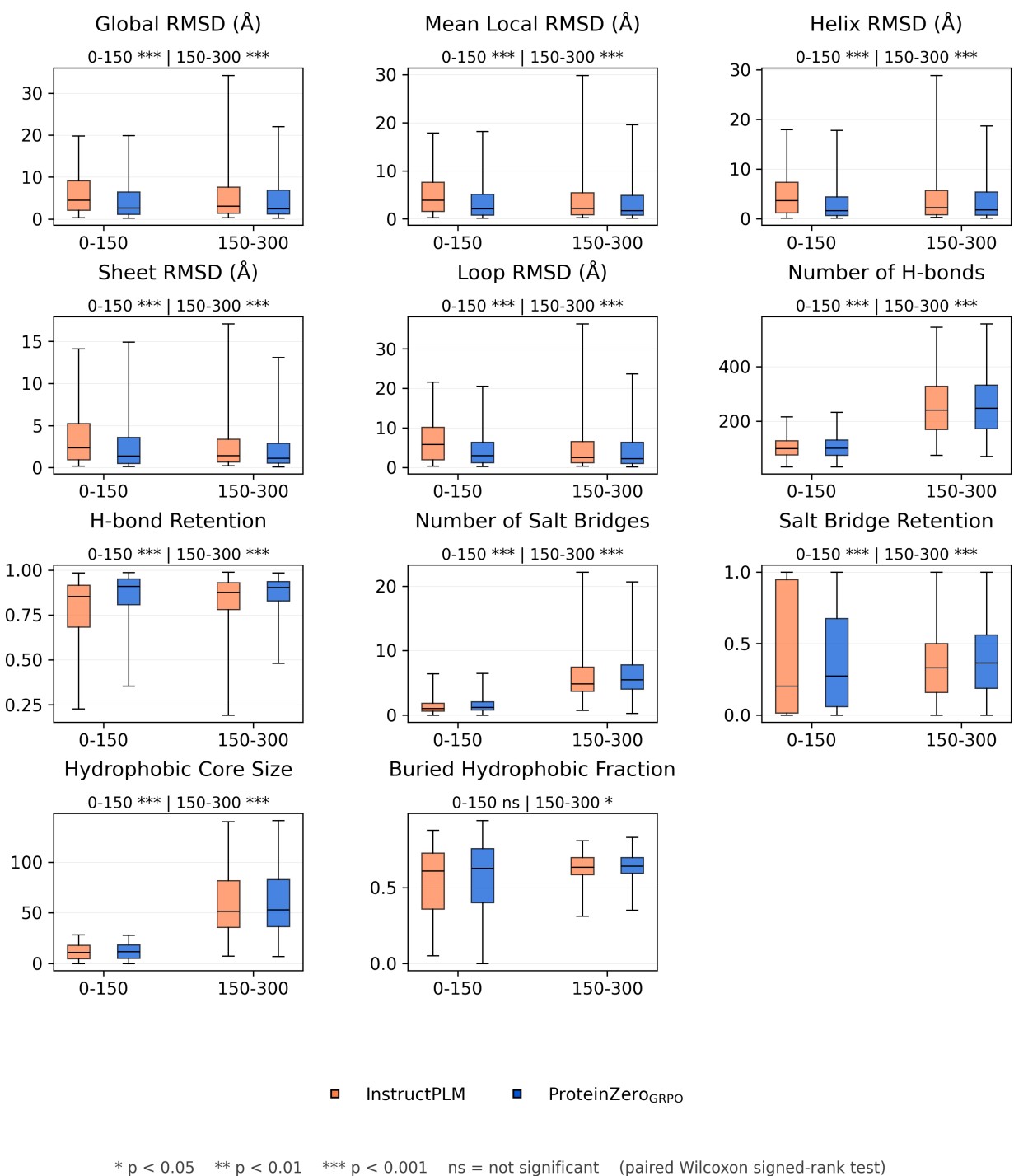

Figure 11: Per-backbone box plots of the eleven metrics in Table 11, RMSDs (global, local, helix, sheet, loop), hydrogen-bond and salt-bridge counts and retention, hydrophobic-core size, and buried hydrophobic fraction, for InstructPLM vs ProteinZero$_{\text{GRPO}}$ at 0–150 and 150–300 residues. Boxes span $Q_1$–$Q_3$ with a median line; whiskers extend to per-backbone min/max. Panel annotations: paired Wilcoxon signed-rank tests within each length category ($^*p < 0.05$, $^{**}p < 0.01$, $^{***}p < 0.001$, ns: $p \geq 0.05$).

Table 12: Sequence-substitution-pattern metrics derived from per-position BLOSUM62 scores relative to native sequences. Values are reported as mean $\pm$ standard deviation (top line) and median $[Q_1, Q_3]$ (bottom line) across backbones; $p$ values are from the two-sided paired Wilcoxon signed-rank test applied to per-backbone means.

| Metric | 0–150 residues | | | 150–300 residues | | |
|---|---|---|---|---|---|---|
| | InstructPLM | ProteinZero$_{\text{GRPO}}$ | $p$ | InstructPLM | ProteinZero$_{\text{GRPO}}$ | $p$ |
| Number of Mutations ↓ | $38.13 \pm 12.47$ $37.28$ [28.08–46.41] | $34.64 \pm 11.40$ $33.20$ [26.52–40.57] | $< 0.001$ | $87.48 \pm 43.01$ $73.88$ [59.11–110.59] | $81.47 \pm 42.27$ $69.38$ [53.57–100.42] | $< 0.001$ |
| Mutation Rate ↓ | $0.518 \pm 0.157$ $0.509$ [0.402–0.630] | $0.470 \pm 0.145$ $0.450$ [0.359–0.573] | $< 0.001$ | $0.457 \pm 0.138$ $0.449$ [0.364–0.542] | $0.424 \pm 0.135$ $0.414$ [0.337–0.490] | $< 0.001$ |
| Mean BLOSUM62 ↑ | $-0.269 \pm 0.380$ $-0.251$ [$-0.502, -0.024$] | $-0.116 \pm 0.358$ $-0.092$ [$-0.355, 0.112$] | $< 0.001$ | $-0.207 \pm 0.372$ $-0.136$ [$-0.475, 0.094$] | $-0.129 \pm 0.372$ $-0.067$ [$-0.365, 0.159$] | $< 0.001$ |
| Conservative Mutation Fraction ↑ | $0.339 \pm 0.096$ $0.321$ [0.277–0.399] | $0.370 \pm 0.097$ $0.357$ [0.304–0.443] | $< 0.001$ | $0.351 \pm 0.087$ $0.354$ [0.297–0.419] | $0.367 \pm 0.089$ $0.366$ [0.314–0.435] | $< 0.001$ |
| Radical Mutation Fraction ↓ | $0.482 \pm 0.099$ $0.481$ [0.420–0.540] | $0.450 \pm 0.095$ $0.446$ [0.392–0.506] | $< 0.001$ | $0.461 \pm 0.090$ $0.452$ [0.391–0.519] | $0.444 \pm 0.089$ $0.443$ [0.373–0.515] | $< 0.001$ |

This pattern indicates that ProteinZero's improvements over InstructPLM extend to fine-grained structural features under AlphaFold3 refolding, which Table 2 does not capture.

## C.7 Substitution-pattern analysis under BLOSUM62

This analysis examines the character of substitutions that ProteinZero$_{\text{GRPO}}$ introduces relative to the native sequence, on the BLOSUM62 axis. Substitutions are scored with the BLOSUM62 matrix (Henikoff & Henikoff, 1992) via Biopython; positive-score substitutions are classified as conservative and negative-score substitutions as radical. Following the common evaluation protocol described in Section C.6, the five resulting metrics are reported per backbone in Table 12; effect sizes are visualized in Figure 12 and per-backbone box plots are shown in Figure 13.

The shift in substitution patterns is concrete: ProteinZero$_{\text{GRPO}}$ produces fewer substitutions per backbone than the InstructPLM base (median $37.28 \rightarrow 33.20$ in the 0–150 range; $73.88 \rightarrow 69.38$ in the 150–300 range), and a correspondingly lower per-position mutation rate (median $0.509 \rightarrow 0.450$ and $0.449 \rightarrow 0.414$, respectively). The substitutions that remain shift toward higher BLOSUM62 scores: median Mean BLOSUM62 rises from $-0.251$ to $-0.092$ in 0–150 and from $-0.136$ to $-0.067$ in 150–300, with both medians remaining below zero but to a smaller degree. The Conservative Mutation Fraction increases in both length ranges (median $0.321 \rightarrow 0.357$ and $0.354 \rightarrow 0.366$) while the Radical Mutation Fraction decreases ($0.481 \rightarrow 0.446$ and $0.452 \rightarrow 0.443$). Mean and median move in the same direction throughout, and all ten paired comparisons reach $p < 0.001$. These are descriptive statistics about substitution patterns; they are consistent with substitutions tending toward evolutionarily tolerated rather than disruptive directions on BLOSUM62-based measures, and are not claims about evolutionary mechanism.

## C.8 Sequence plausibility under ESM-2

This analysis assesses whether ProteinZero$_{\text{GRPO}}$ designs are scored as plausible by a protein language model that was not part of the reinforcement-learning training pipeline. ESM-2 (Lin et al., 2023) is an independently trained protein language model; designs that lie in regions of sequence space favored by ESM-2 receive lower pseudo-perplexity, equivalently higher pseudo log-likelihood. Pseudo-perplexity is computed via the standard masked-language-model procedure (Meier et al., 2021) using ESM-2 (650M parameters). Following the common evaluation protocol described in Section C.6, the two resulting metrics are reported per backbone in Table 13; paired distributions are shown in Figure 14.

Median pseudo-perplexity decreases from 12.57 to 10.07 in the 0–150 range and from 8.26 to 7.45 in the 150–300 range; median pseudo log-likelihood correspondingly rises (becoming less negative) from $-173.86$ to $-163.37$ in 0–150 and from $-349.94$ to $-340.19$ in 150–300. Mean and median move in the same direction in both length ranges, and all four paired comparisons reach $p < 0.001$. These are computational measurements

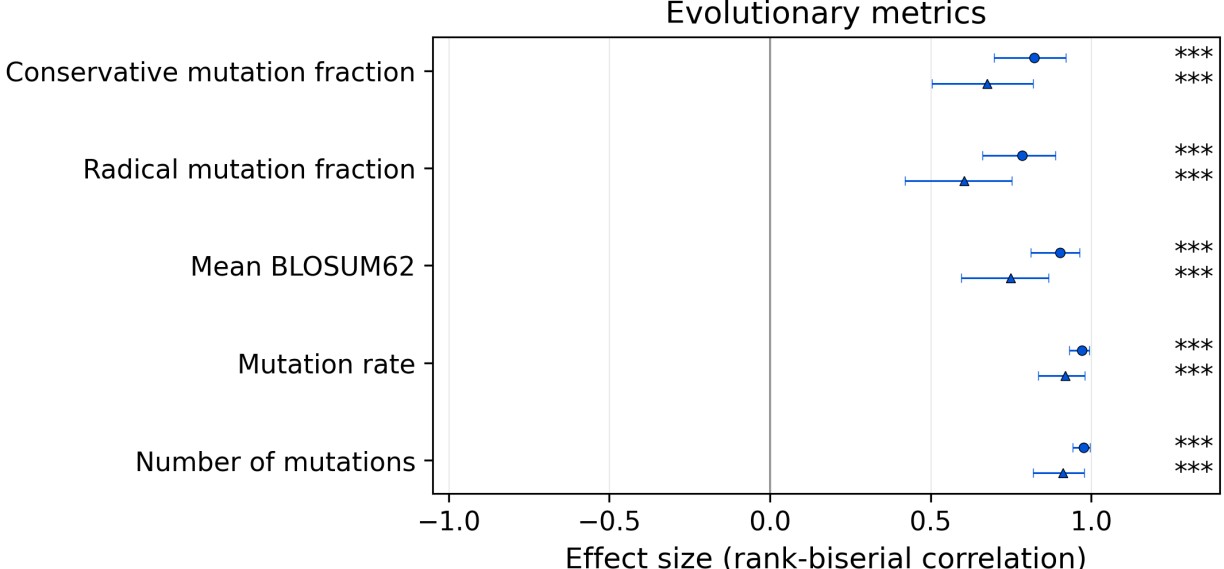

Figure 12: Forest plot of differences in five sequence-substitution-pattern metrics derived from per-position BLOSUM62 substitution scores relative to native sequences, comparing ProteinZero$_{\text{GRPO}}$ against InstructPLM. Effect sizes (rank-biserial correlation) are computed from per-backbone means and oriented so that positive values favor ProteinZero$_{\text{GRPO}}$. Markers denote effect-size estimates (circles: 0–150 residues; triangles: 150–300 residues), and horizontal bars give bootstrap 95% confidence intervals (1,000 resamples). Statistical significance is assessed with the two-sided paired Wilcoxon signed-rank test: $^*p < 0.05$, $^{**}p < 0.01$, $^{***}p < 0.001$, ns = not significant. ProteinZero$_{\text{GRPO}}$ produces sequences with significantly higher BLOSUM62-weighted similarity to native sequences and a significantly larger fraction of conservative substitutions in both length ranges; these are descriptive statistics about substitution patterns and not claims about evolutionary mechanism.

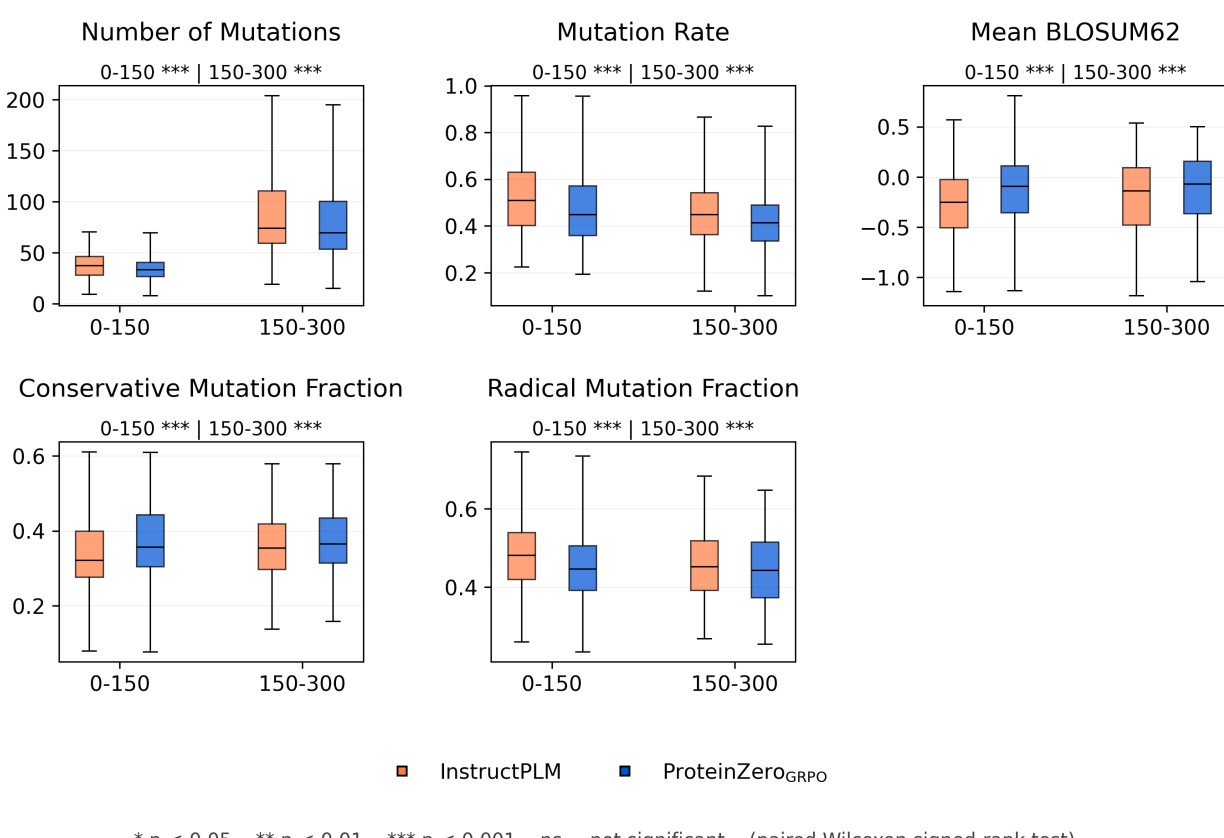

Figure 13: Per-backbone box plots of the five metrics in Table 12, Number of Mutations, Mutation Rate, Mean BLOSUM62, Conservative Mutation Fraction, and Radical Mutation Fraction, for InstructPLM vs ProteinZero$_{\text{GRPO}}$ at 0–150 and 150–300 residues. Boxes span $Q_1$–$Q_3$ with a median line; whiskers extend to per-backbone min/max. Panel annotations: paired Wilcoxon signed-rank tests within each length category ($^*$ $p < 0.05$, $^{**}$ $p < 0.01$, $^{***}$ $p < 0.001$, ns: $p \geq 0.05$).

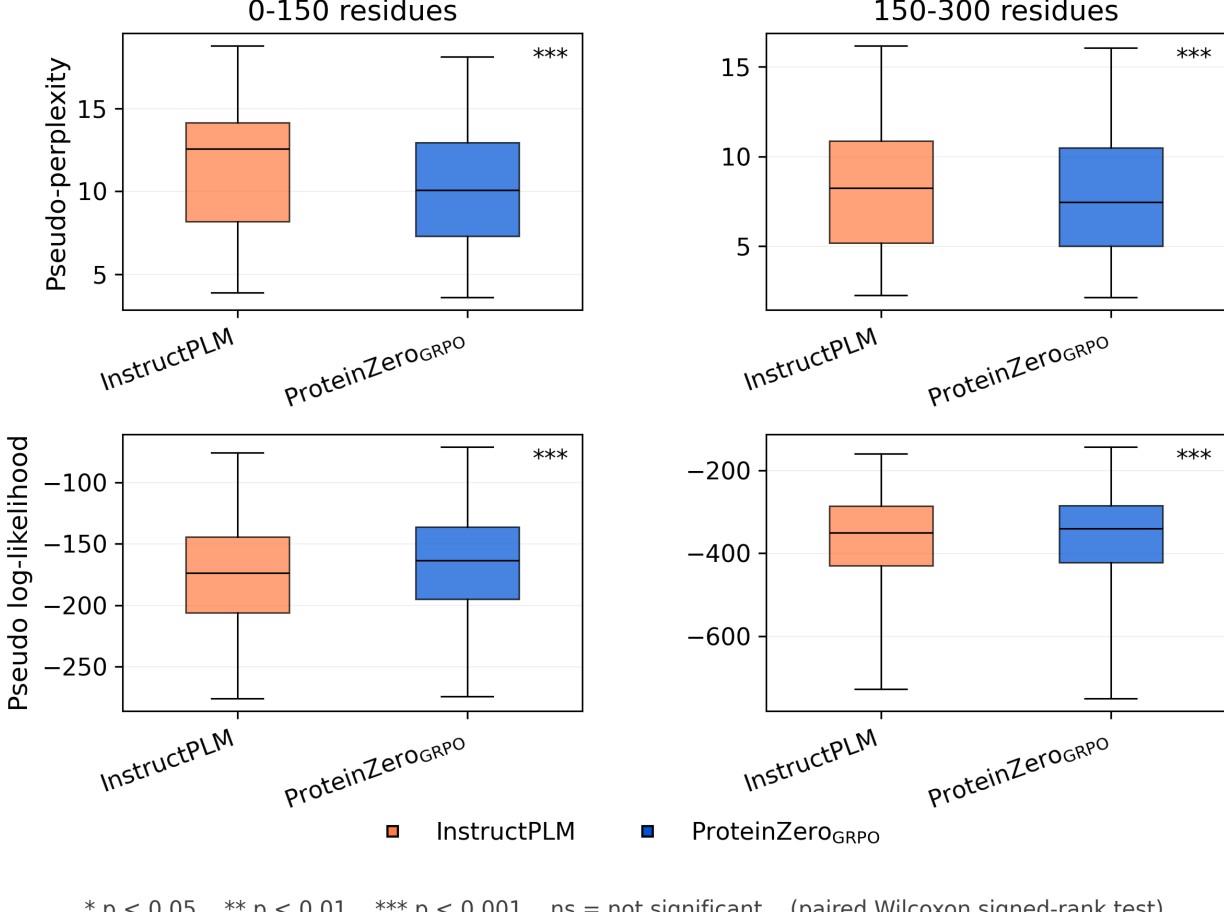

Figure 14: Box plots comparing sequence naturalness between ProteinZero$_{\text{GRPO}}$ and InstructPLM under an external protein language model not used in the reinforcement-learning training pipeline. Each panel summarizes per-backbone pseudo-perplexity (top row, lower is better) and pseudo log-likelihood (bottom row, higher is better) computed by ESM-2 (650M parameters) under the standard masked-language-model scoring procedure, with columns separating the 0–150 and 150–300 residue length ranges. Boxes display the median, interquartile range, and full data range (whiskers extend to the minimum and maximum). Statistical significance is assessed with the two-sided paired Wilcoxon signed-rank test: $^*p < 0.05$, $^{**}p < 0.01$, $^{***}p < 0.001$, ns = not significant. ProteinZero$_{\text{GRPO}}$ achieves significantly lower pseudo-perplexity and significantly higher pseudo log-likelihood in both length ranges, consistent with its designs lying in regions of sequence space to which an independent language model assigns higher likelihood.

Table 13: Sequence plausibility under ESM-2 (650M parameters). Values are reported as mean $\pm$ standard deviation (top line) and median $[Q_1, Q_3]$ (bottom line) across backbones; $p$ values are from the two-sided paired Wilcoxon signed-rank test applied to per-backbone means.

| | 0–150 residues | | | 150–300 residues | | |
|---|---|---|---|---|---|---|
| Metric | InstructPLM | ProteinZero$_{\text{GRPO}}$ | $p$ | InstructPLM | ProteinZero$_{\text{GRPO}}$ | $p$ |
| Pseudo-perplexity ↓ | $11.54 \pm 3.80$ $12.57\ [8.19–14.15]$ | $10.27 \pm 3.64$ $10.07\ [7.31–12.95]$ | $< 0.001$ | $8.48 \pm 3.62$ $8.26\ [5.18–10.86]$ | $8.07 \pm 3.69$ $7.45\ [5.02–10.49]$ | $< 0.001$ |
| Pseudo log-likelihood ↑ | $-174.76 \pm 43.79$ $-173.86\ [-206.19, -144.17]$ | $-166.75 \pm 43.62$ $-163.37\ [-195.26, -136.01]$ | $< 0.001$ | $-371.18 \pm 132.05$ $-349.94\ [-429.42, -286.28]$ | $-362.73 \pm 137.46$ $-340.19\ [-422.57, -284.70]$ | $< 0.001$ |

Table 14: Sequence-derived biophysical descriptors associated with predicted protein stability and predicted solubility. Values are reported as mean $\pm$ standard deviation (top line) and median $[Q_1, Q_3]$ (bottom line) across backbones; $p$ values are from the two-sided paired Wilcoxon signed-rank test applied to per-backbone means.

| | 0–150 residues | | | 150–300 residues | | |
|---|---|---|---|---|---|---|
| Metric | InstructPLM | ProteinZero$_{\text{GRPO}}$ | $p$ | InstructPLM | ProteinZero$_{\text{GRPO}}$ | $p$ |
| Predicted Scaled Solubility ↑ | $0.742 \pm 0.088$ $0.749\ [0.688–0.808]$ | $0.744 \pm 0.090$ $0.757\ [0.688–0.812]$ | $0.472$ | $0.631 \pm 0.131$ $0.639\ [0.520–0.724]$ | $0.631 \pm 0.133$ $0.634\ [0.523–0.726]$ | $0.975$ |
| GRAVY ≈ | $-0.413 \pm 0.418$ $-0.414\ [-0.706, -0.183]$ | $-0.458 \pm 0.456$ $-0.443\ [-0.743, -0.212]$ | $< 0.001$ | $-0.263 \pm 0.291$ $-0.296\ [-0.456, -0.089]$ | $-0.288 \pm 0.289$ $-0.323\ [-0.472, -0.137]$ | $< 0.001$ |
| pI ≈ | $7.04 \pm 1.79$ $6.98\ [5.59–8.33]$ | $7.40 \pm 1.88$ $7.60\ [5.87–8.87]$ | $< 0.001$ | $6.48 \pm 1.35$ $6.18\ [5.39–7.38]$ | $6.59 \pm 1.40$ $6.46\ [5.39–7.51]$ | $< 0.001$ |
| Molecular Weight ≈ | $8440 \pm 1880$ $8730\ [7270–10000]$ | $8470 \pm 1900$ $8730\ [7290–10000]$ | $< 0.001$ | $20800 \pm 6260$ $20400\ [15500–25800]$ | $20900 \pm 6290$ $20400\ [15600–25800]$ | $< 0.001$ |
| Positive Charge Fraction ≈ | $0.162 \pm 0.049$ $0.155\ [0.131–0.196]$ | $0.174 \pm 0.055$ $0.170\ [0.136–0.201]$ | $< 0.001$ | $0.140 \pm 0.040$ $0.135\ [0.116–0.159]$ | $0.143 \pm 0.040$ $0.139\ [0.119–0.164]$ | $< 0.001$ |
| Negative Charge Fraction ≈ | $0.142 \pm 0.050$ $0.138\ [0.109–0.175]$ | $0.141 \pm 0.052$ $0.137\ [0.106–0.174]$ | $0.116$ | $0.132 \pm 0.032$ $0.131\ [0.113–0.153]$ | $0.133 \pm 0.032$ $0.132\ [0.114–0.152]$ | $0.682$ |
| Aromatic Fraction ≈ | $0.076 \pm 0.034$ $0.075\ [0.050–0.096]$ | $0.077 \pm 0.036$ $0.076\ [0.052–0.098]$ | $0.593$ | $0.087 \pm 0.025$ $0.087\ [0.069–0.103]$ | $0.088 \pm 0.025$ $0.089\ [0.070–0.104]$ | $0.016$ |
| Mean Disorder Score ↓ | $0.218 \pm 0.104$ $0.194\ [0.139–0.293]$ | $0.215 \pm 0.109$ $0.184\ [0.128–0.279]$ | $0.357$ | $0.120 \pm 0.060$ $0.106\ [0.077–0.146]$ | $0.113 \pm 0.055$ $0.102\ [0.076–0.132]$ | $0.001$ |
| Disordered Fraction ↓ | $0.101 \pm 0.131$ $0.035\ [0.004–0.165]$ | $0.097 \pm 0.137$ $0.025\ [0.002–0.148]$ | $0.203$ | $0.034 \pm 0.056$ $0.010\ [0.001–0.040]$ | $0.027 \pm 0.052$ $0.005\ [0.000–0.023]$ | $< 0.001$ |

under one external language model; they are consistent with the fine-tuned designs continuing to lie in regions of sequence space to which an independent language model assigns higher likelihood.

## C.9 Biophysical descriptor preservation

This analysis examines whether basic biophysical qualities (hydrophobicity, charge balance, intrinsic disorder, and predicted solubility) remain in a range comparable to the base model after fine-tuning. None of the descriptors below is part of the training reward. GRAVY, isoelectric point, molecular weight, and charge and aromatic-residue fractions are computed via Biopython's `ProtParam`; predicted scaled solubility (0–1) is obtained from Protein-Sol (Hebditch et al., 2017); intrinsic disorder is predicted with metapredict (Emenecker et al., 2021). Following the common evaluation protocol described in Section C.6, per-backbone summary statistics are reported in Table 14; paired distributions are shown in Figure 15.

The directional disorder metrics improve significantly in the 150–300 length range: Mean Disorder Score decreases from 0.120 to 0.113 (median $0.106 \rightarrow 0.102$, $p = 0.001$) and Disordered Fraction decreases from 0.034 to 0.027 (median $0.010 \rightarrow 0.005$, $p < 0.001$); mean and median move in the same direction in both cases. The same disorder metrics show no significant change in the 0–150 length range ($p = 0.357$ and $p = 0.203$, respectively). Predicted scaled solubility, aromatic fraction (in the 0–150 range), and negative-charge fraction show no significant change relative to the base model in their respective length range(s) and are reported here as explicit non-improvements rather than presented as uniform gains; medians and IQRs likewise remain near the base values. The remaining ≈-oriented descriptors (GRAVY, pI, molecular weight, positive-charge fraction) shift by small amounts and remain in a range comparable to the base model.

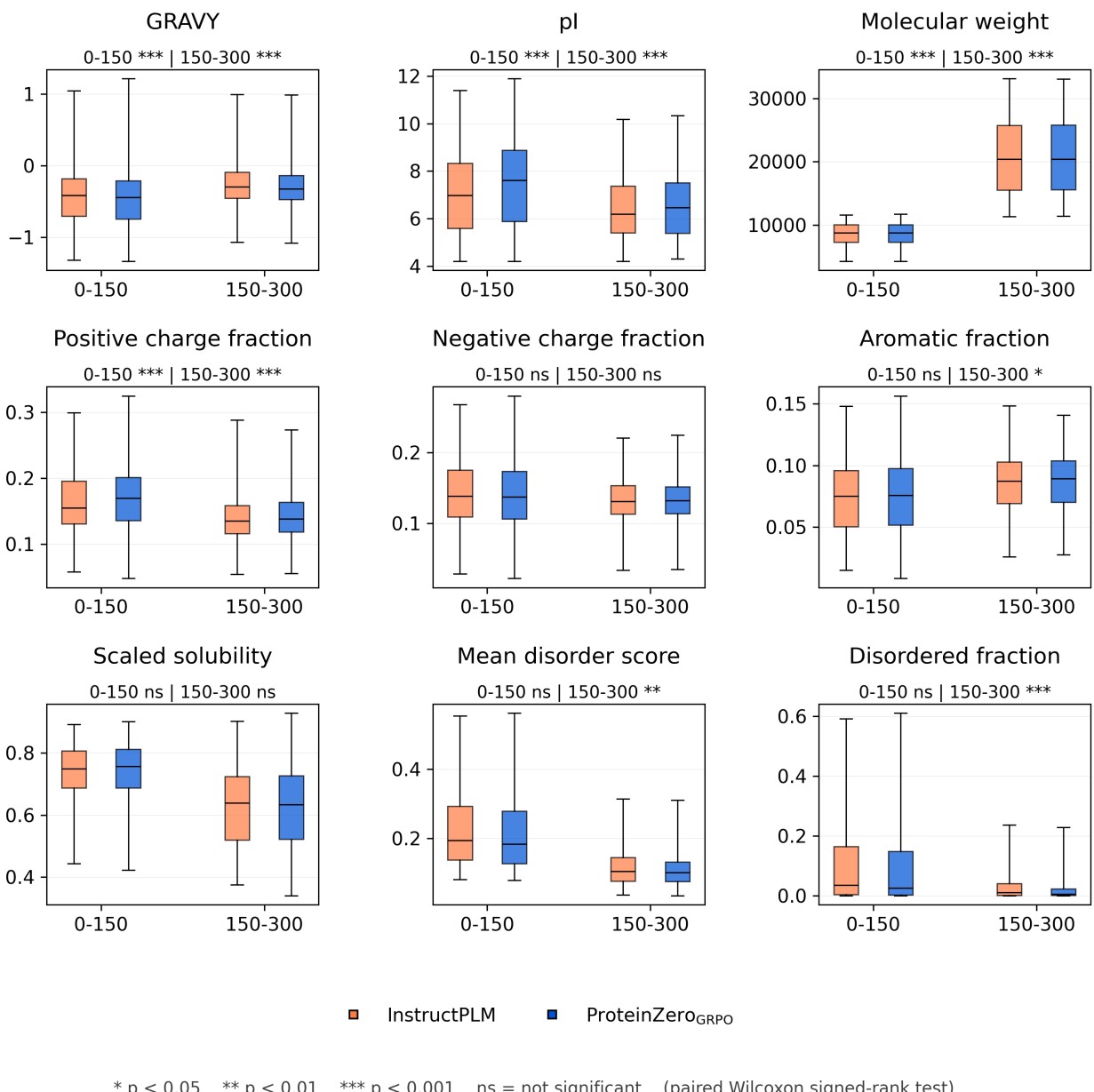

Figure 15: Small-multiples box plots comparing biophysical descriptors between ProteinZero$_{GRPO}$ and InstructPLM across nine sequence-derived descriptors associated with predicted protein stability and predicted solubility. Each panel summarizes per-backbone values for one descriptor; within each panel, box plots are reported separately for the 0–150 and 150–300 residue length ranges. Boxes display the median, interquartile range, and full data range (whiskers extend to the minimum and maximum). Significance symbols above each panel correspond to the two-sided paired Wilcoxon signed-rank test for the two length ranges, respectively: $^*p < 0.05$, $^{**}p < 0.01$, $^{***}p < 0.001$, ns = not significant. The directional disorder metrics (Mean Disorder Score and Disordered Fraction) improve significantly in the 150–300 length range; several other descriptors (predicted scaled solubility, aromatic fraction, negative-charge fraction) show no significant change relative to the base model; the remaining ≈-oriented descriptors shift by small amounts and remain in a range comparable to the base model. The overall pattern is consistent with these biophysical qualities being broadly preserved through fine-tuning, with modest improvements on the disorder axis at longer length and no signs of systematic degradation.

Table 15: UniRef90 best-hit identity profile via MMseqs2 search. Values are reported as mean $\pm$ standard deviation (top line) and median $[Q_1, Q_3]$ (bottom line) of the per-backbone percentage of designs falling in each identity bucket; $p$ values are from the two-sided paired Wilcoxon signed-rank test applied to per-backbone percentages.

| | 0–150 residues | | | 150–300 residues | | |
|---|---|---|---|---|---|---|
| Identity bucket | InstructPLM | ProteinZero$_{\text{GRPO}}$ | $p$ | InstructPLM | ProteinZero$_{\text{GRPO}}$ | $p$ |
| > 70% identity | $19.44 \pm 29.81$ 
 6.25 [0.00–18.75] | $17.19 \pm 27.31$ 
 6.25 [0.00–20.31] | 0.554 | $32.41 \pm 38.76$ 
 12.50 [0.00–68.75] | $31.00 \pm 38.77$ 
 6.25 [0.00–62.50] | 0.290 |
| 30%–70% identity | $60.24 \pm 34.89$ 
 64.06 [30.47–93.75] | $58.68 \pm 31.93$ 
 62.50 [34.38–85.94] | 0.668 | $67.34 \pm 38.84$ 
 87.50 [31.25–100.00] | $68.59 \pm 38.71$ 
 93.75 [37.50–100.00] | 0.283 |
| < 30% identity | $20.31 \pm 32.00$ 
 0.00 [0.00–29.69] | $24.13 \pm 30.93$ 
 4.69 [0.00–49.22] | 0.447 | $0.25 \pm 1.92$ 
 0.00 [0.00–0.00] | $0.41 \pm 3.76$ 
 0.00 [0.00–0.00] | 0.706 |

Concretely, median GRAVY moves from $-0.414$ to $-0.443$ in the 0–150 range and from $-0.296$ to $-0.323$ in the 150–300 range; median pI rises from 6.98 to 7.60 in 0–150 and from 6.18 to 6.46 in 150–300; median molecular weight is essentially unchanged in both length ranges ($8730 \rightarrow 8730$ in 0–150; $20400 \rightarrow 20400$ in 150–300); median positive-charge fraction rises from 0.155 to 0.170 in 0–150 and from 0.135 to 0.139 in 150–300. Mean and median move in the same direction for each of these descriptors. The overall pattern is consistent with these basic biophysical qualities being broadly preserved through fine-tuning, with modest improvements on the disorder axis at longer length and no signs of systematic degradation.

### C.10 Sequence-database identity profile against UniRef90

This analysis examines whether fine-tuning shifts the designs toward closer copies of natural proteins, a concern related to memorization. Each designed sequence is searched against UniRef90 (Suzek et al., 2015) with MMseqs2 (Steinegger & Söding, 2017) under deliberately homolog-favoring settings (maximum sensitivity `-s 7.5`, permissive E-value `-e 10`); sequences in the < 30% identity bucket therefore remain remote or novel (Rost, 1999) even under search settings biased toward finding matches. Following the common evaluation protocol described in Section C.6, for each backbone we compute the percentage of its 640 designs falling in each identity bucket (> 70%, 30%–70%, < 30%); per-backbone percentages are reported in Table 15 and tested with the two-sided paired Wilcoxon signed-rank test, with the corresponding distributions shown in Figure 16.

The per-backbone bucket distribution of ProteinZero$_{\text{GRPO}}$ is not significantly different from that of InstructPLM in any of the three buckets, in either length range (all six paired comparisons give $p > 0.28$). In the > 70% identity bucket, which is directly relevant to the memorization concern, the median per-backbone percentage is unchanged at 6.25% in the 0–150 range and is not increased in the 150–300 range (medians 12.50% for InstructPLM and 6.25% for ProteinZero$_{\text{GRPO}}$; $p = 0.554$ and $p = 0.290$). In the < 30% bucket, which captures designs remaining remote even under homolog-favoring search settings, the median percentage is not decreased in either length range (medians 0.00 and 4.69% in 0–150; 0.00 and 0.00 in 150–300; $p = 0.447$ and $p = 0.706$). For both models in the 150–300 range, $Q_1$, median, and $Q_3$ are all zero in this < 30% bucket, indicating that it is near-empty for the vast majority of backbones at longer length. The 30%–70% middle bucket likewise shows no significant per-backbone shift ($p = 0.668$ and $p = 0.283$). This non-finding is consistent with the absence of memorization-like shifts in UniRef90 sequence identity under the present setup; it is a computational measurement under one homology-search configuration and does not by itself rule out forms of overfitting that would not manifest at the sequence-identity level.

### C.11 Multi-Objective Reward Weighting Analysis

The main-paper Reward Model Designs ablation (Table 3) reports the two single-objective endpoints (100% TM-score and 100% $\Delta\Delta G$) of the reward design space. To characterize the full spectrum of weight choices between these two endpoints, we additionally evaluate five intermediate weightings (10%/90%, 30%/70%, 50%/50%, 70%/30%, 90%/10%), for both 0–150 and 150–300 residue proteins, with all other hyperparame-

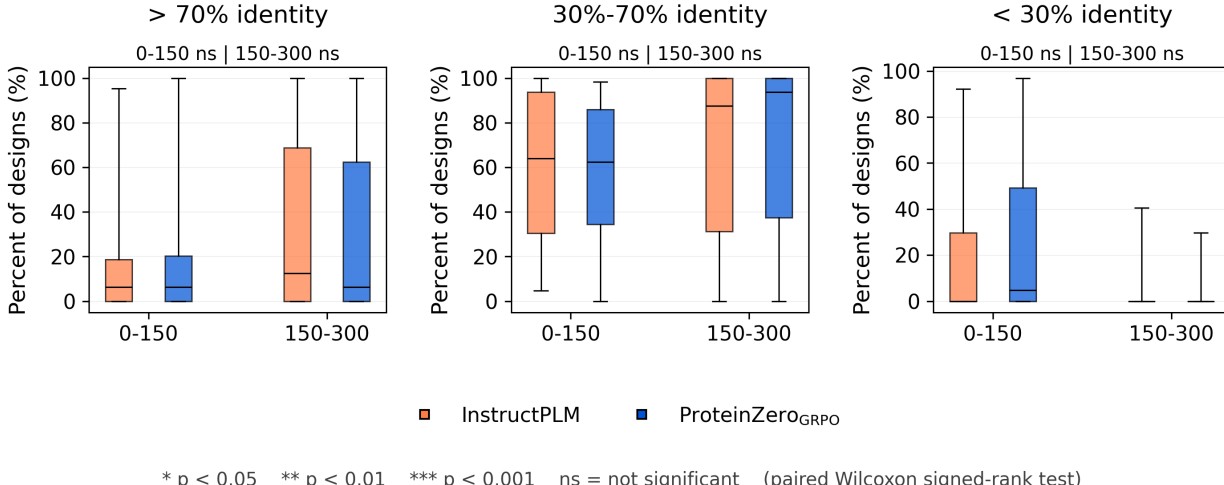

Figure 16: **Sequence-database identity profile against UniRef90.** Per-backbone box plots of the three identity buckets in Table 15 (the percentage of each backbone's designs falling in the $> 70\%$, 30%–70%, or $< 30\%$ identity bucket) for InstructPLM vs ProteinZero$_\text{GRPO}$ at 0–150 and 150–300 residues. Each design was searched against UniRef90 with MMseqs2 under homolog-favoring settings (maximum sensitivity `-s 7.5`, permissive E-value `-e 10`) and assigned to a bucket from its top hit; sequences in the $< 30\%$ bucket remain remote or novel even under settings biased toward finding matches. Boxes span $Q_1$–$Q_3$ with a median line; whiskers extend to per-backbone min/max. Panel annotations: paired Wilcoxon signed-rank tests within each length regime ($^*$ $p < 0.05$, $^{**}$ $p < 0.01$, $^{***}$ $p < 0.001$, ns: $p \geq 0.05$).

Table 16: Multi-objective reward weighting ablation over the full $\lambda_\text{TM}/\lambda_{\Delta\Delta G}$ spectrum for ProteinZero$_\text{GRPO}$. The 0%/100% and 100%/0% endpoints are also reported in Table 3; the five intermediate weightings (10%/90%, 30%/70%, 50%/50%, 70%/30%, 90%/10%) are added here. The 50%/50% row is the default used in the main paper. Best results per column within each length range are highlighted in blue.

| Length | Reward Weighting | InverseFold Acc. | Thermal Stability | | Designability Metrics | | | | Overall |
|---|---|---|---|---|---|---|---|---|---|
| | | Recovery ↑ | Fast-ddG ↓ | FoldX ddG ↓ | TM Score ↑ | PLDDT ↑ | Diversity ↑ | scRMSD ↓ (<2Å% ↑) | Success (%) ↑ |
| 0–150 residues | InstructPLM (base, no RL) | 0.574 | -21.543 | -20.878 | 0.812 | 79.983 | 0.281 | 1.484 (85.71%) | 84.45% |
| | 0% TM + 100% ΔΔG | 0.580 | -22.996 | -25.381 | 0.831 | 82.270 | 0.299 | 1.466 (87.75%) | 85.15% |
| | 10% TM + 90% ΔΔG | 0.583 | -22.943 | -25.341 | 0.836 | 82.273 | 0.302 | 1.458 (88.25%) | 85.92% |
| | 30% TM + 70% ΔΔG | 0.589 | -22.831 | -25.187 | 0.852 | 82.281 | 0.296 | 1.425 (90.18%) | 88.42% |
| | 50% TM + 50% ΔΔG (default) | 0.590 | -22.616 | -24.924 | 0.867 | 82.326 | 0.306 | 1.373 (93.55%) | 90.13% |
| | 70% TM + 30% ΔΔG | 0.581 | -22.247 | -23.618 | 0.869 | 82.485 | 0.304 | 1.376 (93.46%) | 89.96% |
| | 90% TM + 10% ΔΔG | 0.586 | -21.851 | -22.043 | 0.871 | 82.706 | 0.291 | 1.379 (93.18%) | 89.78% |
| | 100% TM + 0% ΔΔG | 0.582 | -21.598 | -21.271 | 0.874 | 82.827 | 0.293 | 1.372 (93.62%) | 89.52% |
| 150–300 residues | InstructPLM (base, no RL) | 0.570 | -36.362 | -27.145 | 0.824 | 83.783 | 0.305 | 1.448 (88.24%) | 86.38% |
| | 0% TM + 100% ΔΔG | 0.574 | -42.769 | -35.927 | 0.831 | 83.540 | 0.327 | 1.447 (88.52%) | 87.38% |
| | 10% TM + 90% ΔΔG | 0.576 | -42.512 | -35.594 | 0.835 | 83.618 | 0.329 | 1.440 (88.74%) | 87.81% |
| | 30% TM + 70% ΔΔG | 0.581 | -41.836 | -34.358 | 0.849 | 83.912 | 0.326 | 1.418 (89.62%) | 89.45% |
| | 50% TM + 50% ΔΔG (default) | 0.580 | -40.626 | -32.805 | 0.862 | 84.154 | 0.331 | 1.393 (90.43%) | 91.19% |
| | 70% TM + 30% ΔΔG | 0.579 | -38.547 | -30.428 | 0.866 | 84.193 | 0.328 | 1.389 (90.84%) | 90.71% |
| | 90% TM + 10% ΔΔG | 0.575 | -36.783 | -27.213 | 0.868 | 84.205 | 0.332 | 1.386 (91.10%) | 89.97% |
| | 100% TM + 0% ΔΔG | 0.577 | -35.793 | -25.905 | 0.870 | 84.237 | 0.333 | 1.384 (91.25%) | 89.76% |

ters held fixed at the main-paper default. The reward takes the form

$$r(x, y) = \lambda_\text{TM} \cdot \tilde{r}_\text{TM}(x, y) + \lambda_{\Delta\Delta G} \cdot \tilde{r}_{\Delta\Delta G}(x, y),$$

where $\tilde{r}_\text{TM}$ and $\tilde{r}_{\Delta\Delta G}$ are min-max normalized so that the two reward components are on a comparable scale, and $\lambda_\text{TM} + \lambda_{\Delta\Delta G} = 1$. The 50%/50% configuration corresponds to the default weighting used throughout the main results (Table 1). Results for both length ranges are reported in Table 16.

The pattern is consistent across both length ranges. Heavily weighting one objective improves the corresponding metric: the 100% TM-score configuration achieves the highest TM Score (0.874 on 0–150 and

0.870 on 150–300), and the 100% $\Delta\Delta G$ configuration achieves the most negative FoldX ddG ($-25.381$ on 0–150 and $-35.927$ on 150–300). The highest overall Success Rate, however, is attained at the 50%/50% default weighting in both length ranges (90.13% on 0–150 and 91.19% on 150–300), exceeding the best single-objective configuration by 0.61 percentage points on 0–150 (90.13% vs. 89.52% at 100% TM-score) and by 1.43 percentage points on 150–300 (91.19% vs. 89.76% at 100% TM-score). This is consistent with the joint nature of our success criterion (scRMSD $< 2$ Å and FoldX ddG $< 0$): neither sub-objective alone is sufficient, and the equal-weighting default balances the two competing rewards in our setup. Across the full sweep, Success Rate remains above the InstructPLM base in every configuration in both length ranges, indicating that the framework retains a useful operating range across the weight choices explored.

## D   Additional Related Work

### D.1   Classical RL vs. RLHF Fine-tuning for Biological Sequence Design

Classical reinforcement learning approaches to biological sequence design emerged before the advent of powerful pre-trained protein models, representing a fundamentally different paradigm from modern RLHF fine-tuning. These methods, developed when large-scale protein language models were not yet available, train task-specific policies from scratch, optimizing sequences directly for defined reward signals. Early work formulated sequence design as Markov decision processes where agents construct or modify sequences step-by-step. Angermueller et al. (2020) employed PPO with model-based variants (DyNA-PPO) to optimize DNA binding sites and antimicrobial peptides, achieving improved sample efficiency through learned simulators. Runge et al. (2019) introduced LEARNA for RNA inverse folding, using PPO to build sequences nucleotide-by-nucleotide with meta-learning across large-scale structure datasets. Even recent planning-based approaches continue this paradigm: Lutz et al. (2023) developed AlphaZero-style MCTS for protein nanomaterial design, discovering assemblies with atomic-precision geometry verified by cryo-EM, while Wang et al. (2023b) proposed EvoPlay, treating amino acid mutations as moves in single-player games for efficient variant exploration. Classical RL methods typically rely on physics-based or learned oracles (e.g., Rosetta energies, AlphaFold predictions, docking scores) within optimization loops. Skwark et al. (2020) used Rosetta-based binding energy to evolve ACE2 variants against SARS-CoV-2, demonstrating substantial improvements in binding affinity with significantly reduced computational requirements compared to traditional design algorithms. These approaches perform online optimization, iteratively querying oracles which can require thousands to millions of evaluations for complex objectives. Model-based variants help address computational costs: DyNA-PPO trains surrogate models between experimental rounds, while Jain et al. (2022) combines GFlowNets with active learning to sample diverse high-fitness sequences proportional to reward, achieving enhanced diversity compared to standard RL baselines. Wang et al. (2025) introduced DRAKES for reward optimization in discrete diffusion models. Their approach enables direct reward backpropagation through diffusion trajectories via Gumbel-Softmax approximations when rewards are differentiable; for non-differentiable rewards, they resort to standard policy gradient methods (PPO) or reward-weighted maximum likelihood estimation. ProteinZero targets protein inverse folding models, employing online RL with policy gradients designed from the outset for non-differentiable scalar rewards from structure predictors (ESMFold, US-align) and stability oracles (Fast-ddG, FoldX). A key distinction lies in diversity handling: DRAKES reports sequence entropy as a post-hoc metric without explicit regularization, whereas ProteinZero incorporates embedding-level diversity regularization with theoretical guarantees (Appendix F) to actively prevent mode collapse during training. While both advance reward-guided protein design, they address complementary model classes: DRAKES for discrete diffusion, ProteinZero for protein inverse folding.

Classical methods often focus on specific objectives such as binding affinity, folding accuracy, or assembly geometry, learning the necessary biophysical constraints through exploration. In contrast, RLHF fine-tuning, enabled by the recent emergence of powerful pre-trained models, operates in a different problem setting: leveraging these foundation models that already encode extensive biophysical knowledge, we refine competent generators rather than training naive policies. This setting enables holistic multi-objective optimization for generalizable improvements, promotes sequence realism without hard-coded penalties, and achieves sample-efficient learning as every oracle query refines an already capable model rather than teaching basic constraints

from scratch. The pre-trained foundation facilitates generation of biologically plausible candidates that satisfy RL objectives, fundamentally changing the optimization landscape compared to classical approaches that must discover these constraints through extensive exploration from scratch.

## D.2 Mode Collapse in Online Reinforcement Learning

Mode collapse represents a critical failure mode in online reinforcement learning where policies converge to narrow output distributions despite diverse valid solutions existing. Kirk et al. (2024) demonstrate that RLHF significantly reduces output diversity compared to supervised fine-tuning, with models producing uniform responses across different inputs. Cui et al. (2025) reveal that policy entropy plummets early in training, causing exploration to vanish and performance to saturate. As models converge to limited outputs, policy distributions become highly peaked, creating a vicious cycle where reduced diversity leads to overconfidence, further limiting exploration. The standard KL penalty in PPO-style RLHF only partially alleviates this issue, as reverse KL is inherently mode-seeking, which favors single high-probability solutions.

Various mitigation strategies have emerged. Entropy regularization directly adds bonuses to maintain broader distributions: Shekhar et al. (2024) integrate self-entropy into preference optimization, while Wang et al. (2024) show forward KL and Jensen-Shannon divergences achieve better alignment-diversity trade-offs than reverse KL. Diversity-reinforced objectives explicitly incorporate variety into rewards, with Li et al. (2025) using semantic clustering as diversity bonuses to achieve simultaneous improvements in quality and novelty. Data mixing strategies like SimpleMix (Li & Khashabi, 2025) combine on-policy and off-policy data to prevent collapse by maintaining broader training distributions.

Our embedding-level diversity regularization represents a novel contribution. Unlike existing approaches operating on output probabilities or rewards, we directly encourage semantic diversity in latent representation space. By penalizing similarity between hidden states of generated sequences, our method captures meaningful variation beyond surface differences. This complements traditional regularization: KL maintains proximity to reference distributions, entropy encourages probabilistic exploration, while our embedding regularizer ensures exploration of functionally distinct sequence regions. For protein design with expensive oracles, maintaining diversity is critical to maximize information per query. Our approach enables covering more possibilities with fewer oracle calls, avoiding redundant evaluations. The combination provides robust protection against mode collapse while maintaining alignment with design objectives, as demonstrated by simultaneous improvements in diversity and performance metrics.

## D.3 Relation to Fully Atomistic Generative Models

Recent sequence-structure co-generation models include fully atomistic generators (Chroma (Ingraham et al., 2023), Protpardelle (Chu et al., 2024), ProteinGenerator (Lisanza et al., 2023)) and backbone-level co-design methods (MultiFlow (Campbell et al., 2024)). These models learn joint distributions over three-dimensional backbone geometries and amino acid sequences for *de novo* fold sampling with compatible sequences. They combine continuous backbone representations (residue frames or atomic coordinates) with discrete or relaxed sequence representations; ProteinGenerator performs diffusion in continuous sequence space coupled to structure prediction networks for atomic coordinates.

ProteinZero addresses the complementary problem of backbone-conditioned inverse folding. In practical engineering workflows, enzyme optimization or epitope-specific binder design, backbone geometry is predetermined by experimental structures, docking simulations, or motif grafting and must be preserved as a hard constraint. The objective is identifying sequences maximizing stability and foldability for fixed geometries rather than generating novel backbone shapes. ProteinZero provides sequence refinement on fixed backbones where structural template preservation is essential.

A fundamental distinction lies in computational tractability for online RL. Applying online RL to joint sequence-structure generators entails repeated sampling in high-dimensional continuous coordinate space ($\mathbb{R}^{3 \times N}$, often including side-chain atoms), with reward evaluation requiring expensive physics-based simulations or slow structural oracles for geometric validity. ProteinZero operates in discrete sequence space with efficient proxy rewards (Fast-ddG, ESMFold), demonstrating that multi-objective, online RL is tractable for

Table 17: Comparison of diversity incorporation strategies. We report success rate, FoldX ddG (kcal/mol), and TM-score for proteins of different lengths.

| Strategy | Success Rate (%) | | FoldX ddG (kcal/mol) | | TM-score | |
|---|---|---|---|---|---|---|
| | 0–150 | 150–300 | 0–150 | 150–300 | 0–150 | 150–300 |
| (1) Embedding reward | 78.65 | 81.71 | -18.681 | -23.967 | 0.836 | 0.831 |
| (2) Hamming reward | 74.63 | 80.29 | -11.135 | -23.228 | 0.836 | 0.831 |
| (3) Embedding regularization | 90.13 | 91.19 | -24.924 | -32.805 | 0.867 | 0.867 |

sequence optimization. Our evaluation focuses on standard backbone-conditioned inverse folding benchmarks (CATH-4.3) rather than direct comparison with *de novo* atomistic generators. This isolates the online RL algorithm's contribution: fixed backbones ensure the observed 36–48% reduction in design failure rates is attributable to policy optimization and diversity regularization rather than backbone sampling or flexibility.

# E   Additional Results on Diversity Regularization Strategies

In the main text, we discuss the impact of incorporating diversity through different strategies. For completeness, Table 17 reports the detailed numerical results, including success rate, FoldX ddG, and TM-score for both protein length categories. These results further illustrate that embedding-based diversity applied as a regularizer preserves stability and structural accuracy, while reward-based variants lead to significant degradation in performance.

# F   Diversity Regularizer: Theoretical Foundation for Preventing Mode Collapse

We provide a theoretical analysis of our embedding-level diversity regularizer, demonstrating how it helps prevent mode collapse in online reinforcement learning. We formalize mode collapse for a conditional policy $p_\theta(y \mid x)$ as a sharp decrease in policy entropy $H_\theta(Y \mid X{=}x)$ and a contraction of its effective support. This perspective aligns with maximum-entropy RL, where entropy encourages stochasticity and prevents brittle policies (Haarnoja et al., 2018; Levine, 2018; Geist et al., 2019). The standard KL-regularized objective, $\mathbb{E}[r] - \alpha_{\mathrm{KL}}\mathrm{KL}(p\|p_{\mathrm{ref}})$, yields the Boltzmann distribution $p^*(y \mid x) \propto p_{\mathrm{ref}}(y \mid x) \exp(r(x,y)/\alpha_{\mathrm{KL}})$. A small $\alpha_{\mathrm{KL}}$ or highly peaked rewards can drive concentration and an entropy drop, a known mode-seeking behavior (Todorov, 2006; Levine, 2018).

## F.1   Mean-Field Objective and Properties

Let $Z = \psi_\theta(X, Y) \in \mathbb{S}^{d-1}$ denote the unit-norm embeddings of generated sequences, as constructed in Section 3.1.1. For a fixed input $x$, we simplify notation by considering probability measures $p(\cdot) \equiv p_\theta(\cdot \mid x)$ on sequences $y$. We define a symmetric kernel $c(y,y') = \cos(\psi_\theta(x,y), \psi_\theta(x,y'))$.

**Assumption 1** (Absolute continuity and i.i.d. pairing)**.** *For each $x$, the feasible set is $\{p \in \Delta: \ p(\cdot \mid x) \ll p_{\mathrm{ref}}(\cdot \mid x)\}$, ensuring $\mathrm{KL}(p\|p_{\mathrm{ref}})$ is finite. Expectations over pairs $(y, y')$ are taken w.r.t. the product measure $p(\cdot \mid x) \otimes p(\cdot \mid x)$ (i.i.d. draws).*

**Remark 1** (Setting and scope of analysis)**.** *All variational arguments below fix $\theta$ and the conditioning input $x$, and treat $p(\cdot) \equiv p_\theta(\cdot \mid x)$ as the optimization variable. We work with discrete sequence policies, so $p_{\mathrm{ref}}(y \mid x) > 0$ on the feasible support, making atomic distributions $\delta_{y^\star}$ admissible whenever $p_{\mathrm{ref}}(y^\star \mid x) > 0$.*

*Coefficient sign convention.* Throughout we assume nonnegative coefficients, in particular $\alpha_{\mathrm{div}} \geq 0$ (and $\alpha_{\mathrm{KL}} \geq 0$), so that the diversity term acts as a repulsive regularizer.

At the population level, we analyze the regularized functional for any fixed $x$:

$$\max_{p \in \Delta: \ p \ll p_{\mathrm{ref}}(\cdot|x)} \mathcal{J}[p] := \mathbb{E}_{y \sim p}[r(x,y)] - \alpha_{\mathrm{KL}} \, \mathrm{KL}\big(p \, \| \, p_{\mathrm{ref}}(\cdot \mid x)\big) - \frac{\alpha_{\mathrm{div}}}{2} \, \mathbb{E}_{y,y' \sim p}\big[c(y,y')\big]. \quad (8)$$

**Remark 2.** *Writing the diversity term as $+\frac{\alpha_{\mathrm{div}}}{2}\big(1-\mathbb{E}[c]\big)$ is equivalent up to an additive constant and yields the same optimizer.*

**Lemma 1** (Diversity as a penalty on the embedding mean). *Under Assumption 1, with $Z = \psi_\theta(X, Y)$ on the unit sphere, the diversity term is the squared norm of the mean embedding: $\mathbb{E}_{y,y'\sim p}\big[c(y,y')\big] = \big\|\mathbb{E}_{y\sim p}[Z]\big\|_2^2$. Consequently, the objective*

$$\mathcal{J}[p] = \mathbb{E}_p[r] - \alpha_{\mathrm{KL}}\mathrm{KL}(p\|p_{\mathrm{ref}}) - \frac{\alpha_{\mathrm{div}}}{2}\,\|\mathbb{E}_p[Z]\|_2^2$$

*is concave in $p$. It is strictly concave on the relative interior if $\alpha_{\mathrm{KL}} > 0$.*

**Proposition 1** (Interior fixed point with a non-local repulsive potential). *Assume $\alpha_{\mathrm{KL}} > 0$. Any interior stationary point $p^*$ of Eq. 8 (where $p^*(y) > 0$ for all feasible $y$) satisfies*

$$p^*(y \mid x) \;\propto\; p_{\mathrm{ref}}(y \mid x)\,\exp\!\left(\frac{1}{\alpha_{\mathrm{KL}}}r(x,y) - \frac{\alpha_{\mathrm{div}}}{\alpha_{\mathrm{KL}}}\,\Phi_\theta\big(y; p^*\big)\right), \tag{9}$$

*where the potential $\Phi_\theta(y; p) := \mathbb{E}_{y'\sim p}\big[c(y,y')\big]$ is a **non-local repulsive term**. Placing mass on a sequence $y$ increases the "energy" of other sequences $y'$ with similar embeddings, discouraging collapse.*

## F.2 Guarantees Against Collapse to a Single Mode

With $\alpha_{\mathrm{KL}} > 0$, the KL term alone rules out collapse to a point mass (delta distribution). The diversity term adds a non-local repulsion that discourages uni-directional concentration in representation space.

**Theorem 1** (KL barrier to deterministic collapse). *Suppose $\alpha_{\mathrm{KL}} > 0$ and Assumption 1 holds. Let $y^\star$ be any sequence with $p_{\mathrm{ref}}(y^\star \mid x) > 0$. If another sequence $y' \neq y^\star$ exists with $p_{\mathrm{ref}}(y' \mid x) > 0$, then the point mass $p = \delta_{y^\star}$ is not a stationary point of Eq. 8.*

*Proof.* Consider a perturbation $p_\varepsilon = (1-\varepsilon)\delta_{y^\star} + \varepsilon\delta_{y'}$ for a small $\varepsilon > 0$. The change in the reward term is $\Delta\mathcal{J}_{\mathrm{reward}} = \varepsilon\big(r(x,y') - r(x,y^\star)\big) + O(\varepsilon^2)$. For the diversity term, let $c = c(y^\star, y')$. The change is $\Delta\mathcal{J}_{\mathrm{div}} = \alpha_{\mathrm{div}}\varepsilon(1-c) + O(\varepsilon^2)$. For the KL divergence, the change is

$$\Delta\mathrm{KL} := \mathrm{KL}(p_\varepsilon\|p_{\mathrm{ref}}) - \mathrm{KL}(\delta_{y^\star}\|p_{\mathrm{ref}})$$

$$= (1-\varepsilon)\log(1-\varepsilon) \;+\; \varepsilon\log\varepsilon \;+\; \varepsilon\log\frac{p_{\mathrm{ref}}(y^\star \mid x)}{p_{\mathrm{ref}}(y' \mid x)}.$$

Using $(1-\varepsilon)\log(1-\varepsilon) = -\varepsilon + O(\varepsilon^2)$, we find that $-\alpha_{\mathrm{KL}}\,\Delta\mathrm{KL}$ is dominated by the term $-\alpha_{\mathrm{KL}}\varepsilon\log\varepsilon$. Combining these, the directional derivative of the full objective is:

$$\lim_{\varepsilon\to 0^+}\frac{\mathcal{J}[p_\varepsilon] - \mathcal{J}[\delta_{y^\star}]}{\varepsilon} = \underbrace{r(x,y') - r(x,y^\star)}_{\text{reward}} + \underbrace{\alpha_{\mathrm{div}}(1-c)}_{\text{diversity}}$$

$$+ \underbrace{\alpha_{\mathrm{KL}}\left(1 - \log\varepsilon - \log\frac{p_{\mathrm{ref}}(y^\star|x)}{p_{\mathrm{ref}}(y'|x)}\right)}_{\text{KL barrier}} + o(1).$$

As $\varepsilon \to 0^+$, the $-\log\varepsilon$ term drives the quotient to $+\infty$. Moving probability mass away from any single point mass $\delta_{y^\star}$ thus always increases the objective, meaning $\delta_{y^\star}$ cannot be a stationary point. $\qquad\square$

**Proposition 2** (No-KL case: finite condition that rules out a delta optimum). *If $\alpha_{\mathrm{KL}} = 0$ and there exists $y' \neq y^\star$ such that*

$$r(x,y') - r(x,y^\star) \;+\; \alpha_{\mathrm{div}}\big(1 - c(y^\star, y')\big) \;>\; 0,$$

*then $p = \delta_{y^\star}$ is not a (local) maximizer of Eq. 8.*

**Corollary 1** (Readable sufficient condition). *For any $y^\star, y'$ with $p_{\mathrm{ref}}(y^\star \mid x), p_{\mathrm{ref}}(y' \mid x) > 0$:*

- *If $\alpha_{\mathrm{KL}} > 0$, then $p(\cdot \mid x) = \delta_{y^\star}$ is never stationary (Theorem 1).*

- If $\alpha_{\mathrm{KL}} = 0$, *a sufficient condition for non-stationarity is* $r(x, y^\star) - r(x, y') < \alpha_{\mathrm{div}}\big(1 - c(y^\star, y')\big)$, *which is the finite, reward–diversity tradeoff stated in Proposition 2.*

**Remark 3** (Scope of the diversity term)**.** *Since* $\mathbb{E}_{y,y'}[c(y, y')] = \|\mathbb{E}[Z]\|^2$, *the regularizer mainly discourages* uni-directional *concentration (single-mode collapse aligned with one embedding direction). It may not penalize symmetric few-mode collapse where* $\mathbb{E}[Z] \approx 0$.

### F.3 Entropy Lower Bound and Implementation

The diversity regularizer also yields a conservative lower bound on policy entropy. Let $Z = \psi_\theta(X, Y) \in \mathbb{S}^{d-1}$ and define the cosine kernel $k(z, z') = (1 + \cos(z, z'))/2 \in [0, 1]$. The *information potential* of the embedding distribution $\nu_\theta(\cdot \mid x)$ is

$$I_k(Z \mid X{=}x) := \mathbb{E}\big[k(Z, Z') \mid X{=}x\big] = 1 - \tfrac{1}{2}\,\bar{D}_{\cos}(\theta; x),$$
$$\bar{D}_{\cos}(\theta; x) := 1 - \mathbb{E}[\cos(Z, Z') \mid X{=}x].$$

Since $H(Y \mid X) \geq H_2(Y \mid X) \geq H_2(Z \mid X)$ and $H_2(Z \mid X) \geq -\log I_k(Z \mid X)$, we obtain the lower bound on policy entropy and perplexity:

$$H_\theta(Y \mid X{=}x) \geq -\log\big(1 - \tfrac{1}{2}\bar{D}_{\cos}(\theta; x)\big), \qquad \mathrm{Perp}_\theta(x) \geq \frac{1}{1 - \tfrac{1}{2}\bar{D}_{\cos}(\theta; x)}. \tag{10}$$

By Lemma 1, $\mathbb{E}[\cos(Z, Z')] = \|\mathbb{E}[Z]\|_2^2 \in [0, 1]$, hence $I_k = \tfrac{1}{2}\big(1 + \|\mathbb{E}[Z]\|_2^2\big) \in [1/2, 1]$. Thus the bound is conservative and cannot exceed $\log 2$ (equivalently, the perplexity lower bound is at most 2). It should be viewed as a safety valve rather than a strong guarantee.

**Remark 4** (Mini-batch estimator)**.** *In practice we estimate* $\bar{D}_{\cos}$ *using off-diagonal pairs to avoid upward bias:*

$$\widehat{\mathbb{E}[\cos]} = \frac{1}{m(m-1)} \sum_{i \neq j} \cos(z_i, z_j) = \frac{m\|\bar{z}\|_2^2 - 1}{m - 1} \in \Big[-\frac{1}{m-1}, 1\Big],$$
$$\widehat{\bar{D}}_{\cos} = 1 - \widehat{\mathbb{E}[\cos]} \in \Big[0, 1 + \frac{1}{m-1}\Big].$$

*When* $m \geq 3$, $1 - \widehat{\bar{D}}_{\cos}/2 > 0$ *holds automatically; for* $m = 2$, *a tiny truncation can be applied before evaluating* $-\log(1 - \widehat{\bar{D}}_{\cos}/2)$.

The objective in Eq. 8 is implemented in our ProteinZero$_{\mathrm{RAFT}}$ and ProteinZero$_{\mathrm{GRPO}}$ algorithms by appending the diversity loss term, which induces the repulsive fixed point from Eq. 9 and benefits from the entropy guarantees of Eq. 10.

