# OpenReview forum: "ProteinZero: Self-Improving Protein Generation via Online Reinforcement Learning"
_TMLR — Under review for TMLR_

### Review · Reviewer_Ks7r · 2026-04-24

**Summary Of Contributions:**

ProteinZero proposes an online RL fine-tuning framework for protein inverse folding, built on top of InstructPLM. The paper targets three problems: reliance on PDB-derived datasets that cover limited sequence space, misalignment between sequence-recovery objectives and real design goals, and mode collapse that the authors empirically observe under naive online RL fine-tuning (Figures 7–8).

The contributions are:

**Online RL for inverse folding.** To address the misalignment problem, the paper adapts GRPO and  RAFT to fine-tune an inverse folding model over 20 iterative rounds without curated preference data.  The full training objective contains three terms: an RL loss $L_{RL}$, a KL divergence penalty  $L_{KL}$ against a frozen reference policy, and an embedding-level diversity regularizer $L_{Div}$, i.e., $L(\theta) = L_{RL}(\theta) + L_{KL}(\theta) + L_{Div}(\theta)$. The KL term prevents catastrophic forgetting, while the diversity term is a separate component designed to prevent mode collapse. The innovation over prior offline RL approaches (DPO, Multi-Round DPO) is the shift to continuous online self-improvement from model-generated sequences.

**Empirical results.** ProteinZeroGRPO achieves 90.13% and 91.19% overall success rates on CATH-4.3 for 0–150 and 150–300 residue proteins.

**Audience:**

Yes

**Audience Explanation:**

Yes. Protein inverse folding sits at the intersection of generative modeling, reinforcement learning, and computational biology, all of which are active research areas within TMLR's audience. The paper offers a promising approach to addressing the mode collapse problem in
online RL fine-tuning, and the proposed diversity regularization framework provides a useful design perspective that may contribute to improving protein discovery efficiency.

**Claims And Evidence:**

Yes

**Claims Explanation:**

Yes, the core claims are largely supported by the experimental evidence provided.

Evidence 1: Validation in Table 2 demonstrates that ProteinZeroGRPO achieves a success rate of 90.13% under ESMFold evaluation and 91.56% under the AlphaFold3 evaluation for 0–150 residue proteins. Since training rewards rely solely on ESMFold and Fast-ddG, the consistent improvement observed supports the claim that ProteinZero learns generalizable biophysical principles.

Evidence 2: The training dynamics in Figure 7 show that without the diversity regularizer, sequence diversity drops monotonically across 20\% training progress, indicating mode collapse. In contrast, models trained with $L_{Div}$ maintain stable diversity throughout training while achieving higher final rewards. This supports the claim that the embedding-level diversity regularizer effectively prevents mode collapse without sacrificing design performance.

**Requested Changes:**

In the multi-objective reward design, the authors adopt an equal weighting scheme, which is designed to balance the scale of the two objectives. However, how to properly balance multiple rewards is a non-trivial and important problem in reinforcement learning fine-tuning. From the results in Table 3, the two rewards appear to compete with each other. Therefore, I suggest the authors analyze different weighting combinations to better balance the rewards.

To further improve sequence diversity, I suggest the authors consider introducing a contrastive learning objective. By explicitly controlling positive and negative sample pairs, the model could be encouraged to generate sequences that are more meaningfully diverse. For instance, sequences with similar structures but different amino acid compositions could serve as positive pairs, while sequences with low reward scores could serve as negative pairs. This approach may provide a more structured and controllable way to promote diversity compared to the current mean-field regularizer.

---

> ### Author Response · Authors · 2026-05-17
> **Response to Reviewer Ks7r [Part 1]**
>
> We appreciate the reviewer's thorough and accurate summary of our contributions, and the recognition that our core claims are largely supported by the experimental evidence: the cross-oracle consistency between ESMFold and AlphaFold3, and the training dynamics showing that our embedding-level diversity regularizer prevents mode collapse without sacrificing performance. We are also encouraged by the assessment that our framework "provides a useful design perspective" and "a promising approach to addressing the mode collapse problem in online RL fine-tuning." To address the requested changes, we organize our response into two parts: (1) analyzing multi-objective reward weighting combinations, and (2) clarifying the relationship to contrastive learning and the suggested structure-aware extension.
>
> > ### Multi-objective reward weighting (Change 1)
>
> We thank the reviewer for raising this point. The balancing of multiple reward components is indeed a non-trivial and important problem in multi-objective RL fine-tuning, and we appreciate the suggestion to analyze different weighting combinations explicitly. Our original manuscript already reported the two single-objective endpoints ($100\\%$ TM-score and $100\\%$ $\Delta\Delta G$) as part of the reward ablation in Table 3. Following the reviewer's suggestion, we have extended that analysis by adding five intermediate weighting configurations ($10\\%/90\\%$, $30\\%/70\\%$, $50\\%/50\\%$, $70\\%/30\\%$, $90\\%/10\\%$) so that the full spectrum of weight choices is covered, on both 0–150 and 150–300 residue proteins, with all other hyperparameters held fixed at the default. The $50\\%/50\\%$ configuration corresponds to the default weighting used throughout the main results in Table 1. The reward takes the form $r(x,y) = \lambda\_{\text{TM}} \cdot \tilde{r}\_{\text{TM}}(x,y) + \lambda\_{\Delta\Delta G} \cdot \tilde{r}\_{\Delta\Delta G}(x,y)$, where $\tilde{r}\_{\text{TM}}$ and $\tilde{r}\_{\Delta\Delta G}$ are min-max normalized so that the two reward components are on a comparable scale. The combined seven-configuration sweep has been added to the revised manuscript as Appendix C.11 (Table 16) and is reproduced below.
>
> **0–150 residues:**
>
> | Reward Weighting | Recovery $\uparrow$ | Fast-ddG $\downarrow$ | FoldX ddG $\downarrow$ | TM Score $\uparrow$ | pLDDT $\uparrow$ | Diversity $\uparrow$ | scRMSD $\downarrow$ ($<2$ Å %) | Success $\uparrow$ |
> |---|---|---|---|---|---|---|---|---|
> | InstructPLM (base, no RL) | $0.574$ | $-21.543$ | $-20.878$ | $0.812$ | $79.983$ | $0.281$ | $1.484$ ($85.71\\%$) | $84.45\\%$ |
> | $0\\%$ TM + $100\\%$ $\Delta\Delta G$ (from Table 3) | $0.580$ | $-22.996$ | $-25.381$ | $0.831$ | $82.270$ | $0.299$ | $1.466$ ($87.75\\%$) | $85.15\\%$ |
> | $10\\%$ TM + $90\\%$ $\Delta\Delta G$ | $0.583$ | $-22.943$ | $-25.341$ | $0.836$ | $82.273$ | $0.302$ | $1.458$ ($88.25\\%$) | $85.92\\%$ |
> | $30\\%$ TM + $70\\%$ $\Delta\Delta G$ | $0.589$ | $-22.831$ | $-25.187$ | $0.852$ | $82.281$ | $0.296$ | $1.425$ ($90.18\\%$) | $88.42\\%$ |
> | $50\\%$ TM + $50\\%$ $\Delta\Delta G$ (default) | $0.590$ | $-22.616$ | $-24.924$ | $0.867$ | $82.326$ | $0.306$ | $1.373$ ($93.55\\%$) | **$\mathbf{90.13\\%}$** |
> | $70\\%$ TM + $30\\%$ $\Delta\Delta G$ | $0.581$ | $-22.247$ | $-23.618$ | $0.869$ | $82.485$ | $0.304$ | $1.376$ ($93.46\\%$) | $89.96\\%$ |
> | $90\\%$ TM + $10\\%$ $\Delta\Delta G$ | $0.586$ | $-21.851$ | $-22.043$ | $0.871$ | $82.706$ | $0.291$ | $1.379$ ($93.18\\%$) | $89.78\\%$ |
> | $100\\%$ TM + $0\\%$ $\Delta\Delta G$ (from Table 3) | $0.582$ | $-21.598$ | $-21.271$ | $0.874$ | $82.827$ | $0.293$ | $1.372$ ($93.62\\%$) | $89.52\\%$ |

---

> ### Author Response · Authors · 2026-05-17
> **Response to Reviewer Ks7r [Part 2]**
>
> **150–300 residues:**
>
> | Reward Weighting | Recovery $\uparrow$ | Fast-ddG $\downarrow$ | FoldX ddG $\downarrow$ | TM Score $\uparrow$ | pLDDT $\uparrow$ | Diversity $\uparrow$ | scRMSD $\downarrow$ ($<2$ Å %) | Success $\uparrow$ |
> |---|---|---|---|---|---|---|---|---|
> | InstructPLM (base, no RL) | $0.570$ | $-36.362$ | $-27.145$ | $0.824$ | $83.783$ | $0.305$ | $1.448$ ($88.24\\%$) | $86.38\\%$ |
> | $0\\%$ TM + $100\\%$ $\Delta\Delta G$ (from Table 3) | $0.574$ | $-42.769$ | $-35.927$ | $0.831$ | $83.540$ | $0.327$ | $1.447$ ($88.52\\%$) | $87.38\\%$ |
> | $10\\%$ TM + $90\\%$ $\Delta\Delta G$ | $0.576$ | $-42.512$ | $-35.594$ | $0.835$ | $83.618$ | $0.329$ | $1.440$ ($88.74\\%$) | $87.81\\%$ |
> | $30\\%$ TM + $70\\%$ $\Delta\Delta G$ | $0.581$ | $-41.836$ | $-34.358$ | $0.849$ | $83.912$ | $0.326$ | $1.418$ ($89.62\\%$) | $89.45\\%$ |
> | $50\\%$ TM + $50\\%$ $\Delta\Delta G$ (default) | $0.580$ | $-40.626$ | $-32.805$ | $0.862$ | $84.154$ | $0.331$ | $1.393$ ($90.43\\%$) | **$\mathbf{91.19\\%}$** |
> | $70\\%$ TM + $30\\%$ $\Delta\Delta G$ | $0.579$ | $-38.547$ | $-30.428$ | $0.866$ | $84.193$ | $0.328$ | $1.389$ ($90.84\\%$) | $90.71\\%$ |
> | $90\\%$ TM + $10\\%$ $\Delta\Delta G$ | $0.575$ | $-36.783$ | $-27.213$ | $0.868$ | $84.205$ | $0.332$ | $1.386$ ($91.10\\%$) | $89.97\\%$ |
> | $100\\%$ TM + $0\\%$ $\Delta\Delta G$ (from Table 3) | $0.577$ | $-35.793$ | $-25.905$ | $0.870$ | $84.237$ | $0.333$ | $1.384$ ($91.25\\%$) | $89.76\\%$ |
>
> The pattern is consistent across both length ranges. Heavily weighting one objective improves the corresponding in-silico metric: the $100\\%$ TM-score configuration achieves the highest TM Score ($0.874$ on 0–150 and $0.870$ on 150–300), and the $100\\%$ $\Delta\Delta G$ configuration achieves the best FoldX ddG ($-25.381$ on 0–150 and $-35.927$ on 150–300). However, the highest overall Success Rate is attained at the default $50\\%/50\\%$ weighting in both length ranges ($90.13\\%$ on 0–150 and $91.19\\%$ on 150–300), exceeding the best single-objective configuration by $0.61$ percentage points on 0–150 ($90.13\\%$ vs $89.52\\%$ at $100\\%$ TM-score) and by $1.43$ percentage points on 150–300 ($91.19\\%$ vs $89.76\\%$ at $100\\%$ TM-score).
>
> This aligns with the joint nature of our success criterion (scRMSD $<2$ Å AND FoldX ddG $<0$): neither sub-objective alone is sufficient, and the equal-weighting default appears to balance the two competing rewards well in our setup. Across the full sweep, Success Rate remains above the InstructPLM base in every configuration in both length ranges, suggesting that the framework retains a useful operating range across the weight choices we explored.

---

> ### Author Response · Authors · 2026-05-17
> **Response to Reviewer Ks7r [Part 3]**
>
> > ### Relationship to contrastive learning (Change 2)
>
> We thank the reviewer for this thoughtful suggestion and for highlighting the connection to contrastive learning. Our embedding-level regularizer was in fact developed with contrastive principles in mind, and the reviewer's framing gives us a clean way to make this connection explicit.
>
> Through the alignment–uniformity decomposition of contrastive objectives [1], our regularizer can be viewed as a mean-field unsupervised instantiation of the uniformity term: it spreads the embeddings of within-batch generated sequences apart in feature space, without requiring positive-pair labels or structural alignment computations at training time. We focused on this uniformity component because, in the online RL setting, the policy is already strongly pulled toward sequences that satisfy the structural and stability targets via the reward signal (TM-score + Fast-ddG). The failure mode that the reward does not directly address, and that the regularizer is designed to handle, is mode collapse onto a narrow set of those high-reward sequences, which uniformity targets directly. This division of labor between reward and regularizer is also empirically supported in our ablations (Table 3): two variants that couple the diversity signal more tightly to the reward, namely adding diversity directly to the RL reward and rewarding the policy by sequence Hamming distance, performed less well than the embedding-level regularizer ($78.65\\%$ and $74.63\\%$ Success Rate on 0–150 residues, respectively, versus $90.13\\%$ with the embedding regularizer; and $81.71\\%$ and $80.29\\%$ versus $91.19\\%$ on 150–300 residues).
>
> We find the reviewer's specific proposal a particularly interesting direction for follow-up work. The current regularizer is structure-blind in the sense that the uniformity term operates only on learned embeddings, with no explicit structural signal flowing through it; structural information enters our pipeline through the reward, but not through the regularizer. Adding an alignment component over structurally similar but sequence-divergent positive pairs, as the reviewer proposes, would make the regularizer structure-aware and let the model learn embeddings in which different sequences that fold to the same structure are explicitly drawn together, a property the current uniformity-only formulation does not enforce. We see this as a very promising avenue for future work.
>
> ---
>
> **References**
>
> [1] T. Wang and P. Isola. Understanding contrastive representation learning through alignment and uniformity on the hypersphere. *ICML*, 9929–9939, 2020.

---

### Review · Reviewer_ENNy · 2026-04-27

**Summary Of Contributions:**

This paper introduces a new approach to online RL for protein generation via a novel diversity regularization and a novel fast proxy reward function. The authors confirm that the approach outperforms current state of the art on CATH-4.3, and they also present a number of ablations and other experiments to improve reader understanding of their approach.

### Strengths
1. Clear overview and justification for the diversity regularization approach, which seems valuable even outside of this context
2. Strong results compared to baselines
3. A wide breadth of ablations and additional analyses

### Weaknesses
1. Lack of clarity around the authors' approach and implementation details
2. Lack of clarity around the training dynamics of the authors' approach
3. Lack of clarity around some of the results

**Additional Comments:**

Overall I think this is strong and exciting work, currently held back by some clarity issues. With these addressed I think the paper could be a great fit for this venue.

**Audience:**

Yes

**Audience Explanation:**

Yes, individuals interested in avoiding mode collapse with RL approaches and individuals interested in protein generation would both like to read this paper. The latter is a general problem across domains giving the paper a potentially broad audience.

**Claims And Evidence:**

Yes

**Claims Explanation:**

For the most part the claims made in the submission do appear to be accurate, convincing, and to have clear evidence. I say for the most part because the authors' approach is somewhat unclear from the paper alone.

**Requested Changes:**

My requested changes are all in line with the weaknesses I identified above.

The authors do not clearly describe their own implementations. It's clear that the authors employed two different RL setups as the optimizer but the details of these setups are unclear. There's no network architectures mentioned at any point, whether the authors made use of some existing implementation exactly or made any adaptations, etc. This is necessary to achieve the level of reproducibility expected of work in this venue. I'd suggest some of the later experiments could be moved to the appendices to make more space for implementation details.

The authors do not describe the training dynamics of their two approaches at all in the main paper, and Figure 8 and the discussion around it is only touched on briefly in the appendices. It's unclear for example how long the authors trained the two approaches for, since Figure 8 gives a reward curve for what seems to be a single run over a percentage of the training time. The dynamics are also interesting here, since both approaches worsen in the final 20% of training time. I'd recommend creating a version of this figure that shows the distribution of training curves over all ten independent runs, and discussing the training dynamics in detail. This would greatly improve reader understanding.

The authors' presentation of their results could be improved in some cases. Figure 2 is very difficult to read, and I'm not sure it strengthens the paper. Many of the results seem very close to baselines. Statistical tests would be useful in these cases to identify if the authors' approach really is separable. Figure 3 also makes use of comic sans for some unclear reason and is generally difficult to read, especially with the choice of light blue text. Though this may be expected in this domain.

---

> ### Author Response · Authors · 2026-05-17
> **Response to Reviewer ENNy [Part 1]**
>
> We thank the reviewer for the encouraging and constructive review, and for recognizing the clear justification of our diversity regularization approach as "valuable even outside of this context," our strong results relative to baselines, and the breadth of our ablations. We are also encouraged by the overall assessment that this is "strong and exciting work" that "could be a great fit for this venue." We fully agree that the concerns center on clarity and reproducibility, and we have thoroughly revised the manuscript to address them. We organize our response into three parts: (1) detailing our implementations and architecture for reproducibility, (2) describing the training dynamics in the main text with revised figures, and (3) improving the presentation of results, including figure clarity.
>
> > ### Implementation and architecture details (Change 1)
>
> We fully agree with the reviewer that the main text would benefit from a clearer, more self-contained description of our implementation, and we are grateful for the specific guidance. Following this suggestion, we have improved the clarity of the implementation description: we have moved the core implementation content into the main text, and the main text now explicitly cross-references the appendix for the lower-level numerical details.
>
> **Content added to the main text.** We have revised the "ProteinZero implementation" paragraph in our experimental setup section (Section 4.1 of the revised manuscript) so that it now describes upfront:
>
> - **Base model architecture.** A ProGen2-xlarge decoder (32 blocks, hidden size 4096, 16 attention heads, approximately 6.57B parameters) conditioned on protein structure through a ProteinMPNN-based encoder that produces 256 learned structural prefix embeddings of dimension 4096, prepended to the decoder's token embeddings.
>
> - **Trainable parameters.** All base-model parameters are frozen during online RL; only LoRA adapters are trained (applied to the self-attention and feed-forward projections of every decoder block and to the structural projection module), totalling approximately 33.8M trainable parameters (0.51% of the base model).
>
> - **Components adopted versus self-implemented.** ProteinZero$\_{\text{GRPO}}$ adopts the GRPO update rule [3] from the HuggingFace TRL library [1]; we implement the remaining components of the online RL pipeline ourselves, including the multi-objective reward computation, the min-max combination of the reward components, the embedding-level diversity regularizer, and the rollout procedure for inverse folding. ProteinZero$\_{\text{RAFT}}$ is implemented from scratch, following the original RAFT formulation [2].
>
> We have additionally clarified, in the diversity-regularizer description (Section 3.1.1), that $h\_{i,t}$ corresponds to the final decoder hidden state after the final layer normalization, with embedding dimension $d=4096$ matching the backbone hidden size.
>
> **Lower-level numerical details remain in Appendix B.2.** The main text now explicitly points to Appendix B.2 for the lower-level numerical details, which were already documented in the original submission: optimizer configuration (AdamW with the specific $\beta\_1$, $\beta\_2$, $\epsilon$, and weight-decay values), learning rates and learning-rate schedules, LoRA rank, $\alpha$, and dropout, sampling strategy (nucleus sampling with temperature 0.8, top-$p$ 0.9; $K=8$ candidates per backbone per GPU, yielding 64 candidates total across 8 GPUs), KL, diversity, and GRPO clipping coefficients, mixed-precision and gradient-accumulation settings, the pLDDT-based filtering criterion, hardware configuration ($8\times$ NVIDIA A100 GPUs), and per-epoch wall-clock times. An additional hyperparameter ablation across KL and diversity coefficients is reported in the supplementary hyperparameter-ablation table in the appendix.

---

> ### Author Response · Authors · 2026-05-17
> **Response to Reviewer ENNy [Part 2]**
>
> > ### Training dynamics (Change 2)
>
> We thank the reviewer for the detailed feedback on Figure 8. Following the suggestion, we have added a new figure to Appendix C.3 (Training Dynamics and Convergence Analysis) that shows the distribution of training trajectories over all 10 independent seeds, with shaded regions indicating $\pm 1$ standard deviation around the seed-averaged curves. Both training reward and success rate are plotted across the full set of seeds, and the x-axis is labeled in training steps rather than as a fraction of training time. This new figure is Figure 9 in Appendix C.3 of the revised manuscript.
>
> The accompanying discussion in Appendix C.3 has been expanded to describe the dynamics that the new figure makes visible. Training reward rises rapidly during the early steps, reaching a plateau by approximately step 1300 for 0–150 residue proteins and step 1500 for 150–300 residue proteins, and then oscillates within a narrow band around its peak. This is consistent with the policy having converged to a high-reward region of the design space, with the subsequent oscillation reflecting standard exploration noise in RL rather than progressive deterioration. The magnitude of the late-stage reward fluctuation is small: the final mean reward is at most $1.19\\%$ below the peak value for 0–150 residues and at most $0.51\\%$ below for 150–300 residues.
>
> Importantly, success rate (the metric most directly tied to the paper's claims, defined in Section 4 as scRMSD $< 2$ Å and FoldX ddG $< 0$) reaches a stable plateau over the same period across all 10 seeds and does not degrade in the final steps, as is now directly visible in the new figure. Across the 10 seeds, the final success rate has a standard deviation of $0.06$ percentage points on 0–150 residues and $0.07$ percentage points on 150–300 residues, indicating that the training behavior is reproducible across runs rather than seed-dependent.
>
> > ### Presentation of results (Change 3)
>
> We thank the reviewer for the detailed feedback on Figures 2 and 3. We address each point below.
>
> **Figure 2 (radar plot).** We recognize that the original caption did not make our intended message sufficiently explicit, and we have revised the Figure 2 caption in the updated manuscript to state this intent directly. ProteinZero uses TM Score (a designability measure) and Fast-ddG (a thermal stability measure based on predicted $\Delta\Delta G$) as its rewards, and additionally promotes diversity through the embedding-level regularizer in the loss. The radar layout is intended to convey that ProteinZero$\_{\text{GRPO}}$ and ProteinZero$\_{\text{RAFT}}$ attain the outermost contour on every axis, both on the success-defining metrics and on the axes we never directly optimize for. Table 1 reports comparisons against ESM-IF, ProteinMPNN, the InstructPLM base, and DPO and Multi-Round DPO (both built on InstructPLM as the base model). Of these baselines, Multi-Round DPO attains the highest Success Rate. Comparing ProteinZero$\_{\text{GRPO}}$ to Multi-Round DPO:
>
> - **On the success-defining metrics**, scRMSD $< 2$ Å (%) improves from $87.95\\%$ to $93.55\\%$ on 0–150 residues and from $89.04\\%$ to $90.43\\%$ on 150–300 residues; FoldX ddG improves from $-21.423$ to $-24.924$ on 0–150 and from $-29.087$ to $-32.805$ on 150–300; and the overall Success Rate (defined as scRMSD $< 2$ Å and FoldX ddG $< 0$) improves from $86.89\\%$ to $90.13\\%$ on 0–150 and from $88.05\\%$ to $91.19\\%$ on 150–300.
>
> - **On the metrics we never directly optimize for**, pLDDT improves from $80.797$ to $82.326$ on 0–150 and from $83.840$ to $84.154$ on 150–300; Recovery Rate improves from $0.569$ to $0.590$ on 0–150 and from $0.569$ to $0.580$ on 150–300.
>
> **Figure 3 (qualitative comparisons).** We thank the reviewer for pointing this out, and we have regenerated both panels of Figure 3 accordingly. We have changed the annotation font and replaced the light-blue text with a deep-blue accent color, with all other text rendered in black for higher contrast. The same changes have been applied to Figure 6 for consistency. The underlying structure renderings and quantitative values are unchanged. The regenerated figures appear in the revised manuscript as Figure 6 (originally Figure 3) and Figure 4 (originally Figure 6); the repositioning was made solely to preserve proper spacing and comply with the TMLR template, and the figure content itself is unchanged from the regeneration described above.
>
> ---
>
> **References**
>
> [1] L. von Werra et al. TRL: Transformers Reinforcement Learning. https://github.com/huggingface/trl, 2020.
>
> [2] H. Dong et al. RAFT: Reward rAnked FineTuning for Generative Foundation Model Alignment. *Transactions on Machine Learning Research*, 2023.
>
> [3] Z. Shao et al. DeepSeekMath: Pushing the Limits of Mathematical Reasoning in Open Language Models. *arXiv:2402.03300*, 2024.

---

> > ### Comment · Reviewer_ENNy · 2026-05-19
> > **Re: Response to Reviewer ENNy [Part 2]**
> >
> > Thanks to the authors for their revision to the paper, all of my concerns have been full addressed!

---

> > > ### Author Response · Authors · 2026-05-20
> > > **Thank you for your review and recognition**
> > >
> > > Dear Reviewer ENNy, we sincerely thank you for confirming that all your concerns have been fully addressed, and for your kind recognition of our work as "strong and exciting." Your insightful feedback on clarity and reproducibility has significantly strengthened our manuscript, and we deeply appreciate your time in reviewing our paper and revisions, and your engagement throughout the discussion!

---

### Review · Reviewer_vme7 · 2026-05-11

**Summary Of Contributions:**

This paper proposes ProteinZero, an online reinforcement learning framework for fine-tuning protein inverse folding models to generate sequences with improved predicted designability, stability, recovery, and diversity. Its main contributions are a multi-objective reward combining ESMFold/US-Align structural feedback with a self-derived Fast-ddG stability proxy, an embedding-level diversity regularizer to reduce mode collapse, and implementations with RAFT and GRPO, with GRPO giving the strongest results. The paper reports strong computational improvements on CATH-4.3 over ProteinMPNN, ESM-IF, InstructPLM, DPO, and multi-round DPO, supported by ablations and cross-oracle checks using AlphaFold3/FoldX-style evaluations.

**Audience:**

Yes

**Audience Explanation:**

The paper is likely to interest at least part of TMLR’s audience, especially researchers working on AI for science, protein design, reinforcement learning from feedback, generative model alignment, and diversity-preserving optimization.

**Broader Impact Concerns:**

The paper includes a Broader Impact section, but it should more clearly discuss dual-use risks from improved protein design, especially potential misuse for harmful proteins or pathogen-related components. The authors should add a brief note on responsible release, sequence screening, intended-use limits, and the need for experimental and safety review before wet-lab deployment.

**Claims And Evidence:**

No

**Claims Explanation:**

The core computational claims are supported by reasonably strong evidence, including benchmark comparisons, cross-oracle evaluation, and ablations.

However, several broader claims about generalizable biophysical principles, functional diversity, stable protein designs, and therapeutic/industrial relevance are stronger than what the evidence can justify. These issues could be addressed by substantially narrowing the claims to computational inverse-folding performance and proxy-based stability improvements.

**Requested Changes:**

The authors should substantially calibrate the claims to match the evidence. In particular, statements suggesting that the method learns “generalizable biophysical principles,” produces truly “stable” proteins, or has direct therapeutic/industrial relevance should be revised to emphasize that the evidence is based on computational proxy metrics and cross-oracle validation, not prospective experimental validation.

The paper should consistently describe stability improvements as predicted stability improvements.

The authors should also clarify the scope and limitations of Fast-ddG: it is a useful training proxy, but its experimental validation is moderate and based mainly on single-mutation data, whereas the method is used for full-sequence redesign.

The statistical reporting should be made clearer, especially the meaning of the 10 independent runs, the extremely small standard errors for success rate, the sampling protocol, and whether uncertainty is computed across sequences, backbones, or random seeds.

---

> ### Author Response · Authors · 2026-05-17
> **Response to Reviewer vme7 [Part 1]**
>
> We are grateful to the reviewer for the careful and constructive review, and for recognizing that our core computational claims are supported by reasonably strong evidence including benchmark comparisons, cross-oracle evaluation, and ablations. We agree that the appropriate framing for our claims is computational and proxy/oracle-based, and we have revised the manuscript accordingly. To address the requested changes, we organize our response into five parts: (1) calibrating the broader
> claims to match the evidence and providing additional cross-oracle experiments, (2) using precise terminology for stability metrics, (3) clarifying the scope and limitations of Fast-ddG, (4) making the statistical reporting explicit, and (5) expanding the Broader Impact discussion to address dual-use considerations.

---

> ### Author Response · Authors · 2026-05-17
> **Response to Reviewer vme7 [Part 2]**
>
> > ### Claim calibration and additional cross-oracle experiments (Change 1)
>
> We fully agree with the reviewer that this is an important and well-founded concern, and we are grateful for the specific guidance, particularly the observation that our evidence base is computational and cross-oracle in nature, and that our claims should be framed accordingly. This is a valuable correction that has improved the precision of the manuscript, and we have revised it in two respects: (i) claim calibration throughout the manuscript, and (ii) additional cross-oracle experiments.
>
> **Claim calibration throughout the manuscript**: We have systematically revised the language so that all claims are proportionate to the computational and proxy/oracle-based evidence (All edits below have been applied in the revised manuscript):
>
>
> | Location | Original | Revised |
> |---|---|---|
> | Introduction, contribution 3 | "while **maintaining functional coherence**" | "while **preserving designability and predicted stability**" |
> | Introduction, contribution 5 | "independent oracles" "to **ensure** improvements **reflect generalizable design principles**" | "independent **computational** oracles" "to **assess whether** improvements **transfer consistently across independent in-silico evaluators**" |
> | Main Results, "Overall Performance Analysis" | "**demonstrating genuine** gains **beyond** reward hacking" "**confirms** these improvements **are generalizable**" "**learns transferable** biophysical principles **rather than exploiting** predictor-specific artifacts" | "**suggesting the** gains **are not solely attributable to** reward hacking" "**shows** these improvements **transfer across computational structure predictors**" "**are not merely** predictor-specific artifacts; **confirming that they reflect genuine** biophysical principles **would require prospective experimental validation**" |
> | Main Results, "Effectiveness of fast-ddg reward" | "captures **generalizable** thermodynamic **signals**" | "captures thermodynamic signal **beyond proxy-specific patterns, although it remains a computational surrogate rather than a substitute for experimental measurement**" |
> | Case Study, subheading | "**for Therapeutic Value**" | "**Across Diverse Folds**" |
> | Case Study, "Stabilization..." paragraph | "into **stable** designs" "to a **stable** design" "stability profiles **valuable** for therapeutic and industrial applications" | "into designs **with substantially improved predicted stability**" "to a design **with improved predicted stability**" "**predicted** stability profiles **that may be of interest** for therapeutic and industrial applications, **pending experimental validation**" |
> | Conclusion, final sentence | "**supporting** applications in" | "**with the potential to support future** applications in" |
> | Appendix (Limitations), "Reliance on Computational Proxies" | "**indicates** the model **learns generalizable** biophysical principles" | "**suggests** the model **captures signal that generalizes across computational oracles ...; confirming that this corresponds to genuine** biophysical principles **requires prospective experimental validation**" |
> | Appendix, "Independent Validation with AlphaFold3" | "**validate** that our **online RL** approach learns **generalizable** design principles" "**providing confidence that** the learned policies **will** generalize to practical applications" | "**indicate** that our approach learns design principles **that transfer across computational structure predictors**" "**Whether** the learned policies generalize to practical applications **remains to be established through prospective experimental validation**" |
> | Appendix, "Experimental Validation of Fast-ddG on Ssym" | "captures **generalizable** thermodynamic **principles**" | "captures thermodynamic **signal that generalizes across these experimentally characterized targets**" |
> | "Functionally meaningful" diversity language (four locations) | Abstract / Related Work / Method 3.1.1 / Conclusion: "**functionally meaningful sequence variation**", "**functionally meaningful variation**", "**diverse, functionally plausible sequences**", "**encouraging meaningful variation**" | "**sequence-level diversity among generated designs**", "**sequence-level diversity**", "**diverse candidate sequences that retain high designability and predicted stability**", "**broadening sequence-level diversity among candidate designs**" |
>
>
> - We have also added explicit statements in Section 4, the Conclusion, and the Appendix (A.2 "Reliance on Computational Proxies") that all evaluation in this work is computational, that no prospective wet-lab validation is performed, and that experimental validation remains necessary before any functional or application-level claim.

---

> ### Author Response · Authors · 2026-05-17
> **Response to Reviewer vme7 [Part 3]**
>
> **Additional cross-oracle experiments**: Beyond the revisions above, we have also taken the reviewer's emphasis on cross-oracle validation as motivation to verify, more thoroughly, that ProteinZero$_{\text{GRPO}}$ has not been tuned in a way that sacrifices other qualities important to protein design. We therefore broadened the in-silico evaluation beyond the inverse-folding suite already reported in the paper, examining whether properties this suite does not measure are preserved after fine-tuning, along the following axes:
>
> - **fine-grained structural features under AlphaFold3**;
> - **sequence-level substitution patterns (BLOSUM62)**;
> - **sequence plausibility under an independent protein language model (ESM-2)**;
> - **biophysical descriptors**;
> - **sequence-database identity against UniRef90**.
>
> We emphasize that these remain in-silico metrics that are only suggestive of behavior in wet-lab settings; prospective experimental validation would still be necessary to establish any functional consequence. More detailed figures are included in the uploaded, revised manuscript. For compactness in this rebuttal we report mean $\pm$ std with paired Wilcoxon signed-rank tests [1] below, which test whether ProteinZero$_{\text{GRPO}}$ produces systematically different metric values than the InstructPLM base on the same backbones. In the revised manuscript (already uploaded), each per-backbone metric is additionally reported as median and interquartile range $[Q_1, Q_3]$ across backbones in Tables 11–15 and visualized as per-backbone box plots in Figures 11, 13, 14, 15, and 16 (Appendix C.6–C.10); since per-backbone distributions can be skewed and are not always fully summarized by mean $\pm$ std alone, the median $[Q_1, Q_3]$ view provides a complementary read of the same data. The evaluation setup is as follows:
>
> - **Test backbones.** All analyses are performed on the same CATH-4.3 held-out test backbones used in the main paper, with both length ranges (0–150 and 150–300 residues).
> - **Sampling.** Under the test-set evaluation protocol, the model samples 64 sequences per backbone (temperature $0.15$, top-$p$ $0.9$; see Change 4 (ii) for details); 64 sequences per backbone are taken to average out the residual stochasticity that remains even at this low-temperature, high-confidence setting.
> - **Per-backbone analysis.** While the experiments already reported in the paper aggregate these sequences across the held-out test backbones into a single scalar per metric (for example, each TM-score and FoldX ddG value in Table 1 is one number per model, computed across all test backbones and all 64 sequences per backbone) and therefore average away the per-backbone behavior, the analyses below are computed per backbone to make this behavior visible. Within each backbone, we pool 640 sequences from ProteinZero$_{\text{GRPO}}$ (10 independent training runs × 64 sequences each) and a matched set of 640 sequences from the InstructPLM base [2] (10 sampling seeds × 64 sequences each), with both models sampled under this same evaluation protocol; the two-sided paired Wilcoxon signed-rank test is then computed across backbones. The backbone is the natural unit for this view, because the paired Wilcoxon test pairs the two models on the same backbone by construction. Because seed-level reproducibility has already been established under the Table 1 protocol (Change 4 (i)), we pool the 10 runs into the per-backbone estimate here; this pooling stabilizes the per-backbone estimate consumed by the paired test, without conflating it with seed variance.
>
> Further details on the statistical reporting, including the 10 independent runs, the success-rate standard errors, the sampling protocol, and the level at which uncertainty is computed, are given in Change 4 below. In each table below, $\uparrow$ marks metrics where higher values are typically preferred, $\downarrow$ marks metrics where lower values are typically preferred, and $\approx$ marks metrics where values closer to the native protein are preferred (the directional sign of any shift depends on the specific metric); the $p$ columns report two-sided paired Wilcoxon signed-rank test p-values.

---

> ### Author Response · Authors · 2026-05-17
> **Response to Reviewer vme7 [Part 4]**
>
> - **Fine-grained structural features under AlphaFold3** (folding oracle independent of ESMFold). Beyond Table 2's AlphaFold3 cross-oracle validation in the original manuscript, this section examines fine-grained structural features computed by refolding every designed sequence with AlphaFold3 [3] and comparing with the target backbone. C$\alpha$ RMSD is computed with Biopython [4] Superimposer; secondary structure is assigned with mkdssp [5], the reference implementation of the DSSP algorithm [6]; hydrogen bonds are defined as backbone N–O contacts within $3.5$ Å (adjacent residues excluded); salt bridges as contacts within $4.0$ Å between positively (Lys, Arg, His) and negatively (Asp, Glu) charged side chains; and the hydrophobic core as buried hydrophobic residues with SASA $< 25$ Å$^2$ via Shrake–Rupley [7]. These are reported in Appendix C.6 (Table 11 and Figures 10 and 11) of the revised manuscript.
>
>     | Metric | InstructPLM (0–150) | ProteinZero$\_{\text{GRPO}}$ (0–150) | $p$ (0–150) | InstructPLM (150–300) | ProteinZero$\_{\text{GRPO}}$ (150–300) | $p$ (150–300) |
>     |---|---|---|---|---|---|---|
>     | Global RMSD (Å) $\downarrow$ | $6.20 \pm 5.07$ | $4.57 \pm 4.43$ | $<0.001$ | $6.04 \pm 6.83$ | $5.12 \pm 5.60$ | $<0.001$ |
>     | Mean Local RMSD (Å) $\downarrow$ | $5.37 \pm 4.67$ | $3.84 \pm 4.04$ | $<0.001$ | $4.78 \pm 5.88$ | $3.92 \pm 4.66$ | $<0.001$ |
>     | Helix RMSD (Å) $\downarrow$ | $5.07 \pm 4.55$ | $3.61 \pm 4.10$ | $<0.001$ | $4.70 \pm 5.75$ | $3.87 \pm 4.56$ | $<0.001$ |
>     | Sheet RMSD (Å) $\downarrow$ | $3.94 \pm 4.17$ | $2.62 \pm 3.10$ | $<0.001$ | $3.30 \pm 4.14$ | $2.64 \pm 3.30$ | $<0.001$ |
>     | Loop RMSD (Å) $\downarrow$ | $6.53 \pm 5.40$ | $4.77 \pm 4.58$ | $<0.001$ | $5.57 \pm 6.88$ | $4.68 \pm 5.62$ | $<0.001$ |
>     | Number of H-bonds $\uparrow$ | $106.36 \pm 43.07$ | $108.91 \pm 44.98$ | $<0.001$ | $252.03 \pm 105.86$ | $256.27 \pm 110.13$ | $<0.001$ |
>     | H-bond Retention $\uparrow$ | $0.786 \pm 0.181$ | $0.853 \pm 0.142$ | $<0.001$ | $0.829 \pm 0.142$ | $0.857 \pm 0.120$ | $<0.001$ |
>     | Number of Salt Bridges $\uparrow$ | $1.40 \pm 1.12$ | $1.56 \pm 1.20$ | $<0.001$ | $6.03 \pm 3.81$ | $6.38 \pm 3.82$ | $<0.001$ |
>     | Salt Bridge Retention $\uparrow$ | $0.381 \pm 0.395$ | $0.402 \pm 0.380$ | $<0.001$ | $0.350 \pm 0.250$ | $0.396 \pm 0.258$ | $<0.001$ |
>     | Hydrophobic Core Size $\uparrow$ | $11.51 \pm 7.92$ | $12.13 \pm 8.12$ | $<0.001$ | $58.72 \pm 30.87$ | $59.95 \pm 30.62$ | $<0.001$ |
>     | Buried Hydrophobic Fraction $\uparrow$ | $0.546 \pm 0.234$ | $0.551 \pm 0.256$ | $0.357$ | $0.642 \pm 0.087$ | $0.648 \pm 0.086$ | $0.011$ |
>
>     The observed differences are consistent in direction across both length ranges for all directional metrics, with comparisons reaching $p < 0.001$ on every metric in both length ranges except Buried Hydrophobic Fraction (not significant in 0–150 with $p = 0.357$; significant at $p = 0.011$ in 150–300). This pattern indicates that ProteinZero's improvements over InstructPLM extend to fine-grained structural features under AlphaFold3 refolding, which Table 2 does not capture.
>
> - **Sequence-level substitution patterns** (BLOSUM62-based). This analysis examines whether the substitutions ProteinZero$\_{\text{GRPO}}$ introduces relative to the native sequence shift toward evolutionarily tolerated or disruptive changes. Substitutions are scored with the BLOSUM62 matrix [8] via Biopython [4]; positive-score substitutions are classified as conservative and negative as radical. These are descriptive statistics about substitution patterns and not claims about evolutionary mechanism. These are reported in Appendix C.7 (Table 12 and Figures 12 and 13) of the revised manuscript.
>
>     | Metric | InstructPLM (0–150) | ProteinZero$\_{\text{GRPO}}$ (0–150) | $p$ (0–150) | InstructPLM (150–300) | ProteinZero$\_{\text{GRPO}}$ (150–300) | $p$ (150–300) |
>     |---|---|---|---|---|---|---|
>     | Number of Mutations $\downarrow$ | $38.13 \pm 12.47$ | $34.64 \pm 11.40$ | $<0.001$ | $87.48 \pm 43.01$ | $81.47 \pm 42.27$ | $<0.001$ |
>     | Mutation Rate $\downarrow$ | $0.518 \pm 0.157$ | $0.470 \pm 0.145$ | $<0.001$ | $0.457 \pm 0.138$ | $0.424 \pm 0.135$ | $<0.001$ |
>     | Mean BLOSUM62 $\uparrow$ | $-0.269 \pm 0.380$ | $-0.116 \pm 0.358$ | $<0.001$ | $-0.207 \pm 0.372$ | $-0.129 \pm 0.372$ | $<0.001$ |
>     | Conservative Mutation Fraction $\uparrow$ | $0.339 \pm 0.096$ | $0.370 \pm 0.097$ | $<0.001$ | $0.351 \pm 0.087$ | $0.367 \pm 0.089$ | $<0.001$ |
>     | Radical Mutation Fraction $\downarrow$ | $0.482 \pm 0.099$ | $0.450 \pm 0.095$ | $<0.001$ | $0.461 \pm 0.090$ | $0.444 \pm 0.089$ | $<0.001$ |
>
>     ProteinZero$\_{\text{GRPO}}$ designs show fewer substitutions overall, with a higher fraction of conservative substitutions and a lower fraction of radical substitutions, relative to the InstructPLM base. Computationally, this suggests that fine-tuning does not drive substitutions toward more disruptive directions on these BLOSUM62-based measures.

---

> ### Author Response · Authors · 2026-05-17
> **Response to Reviewer vme7 [Part 5]**
>
> - **Sequence plausibility under an independent protein language model (ESM-2)**. This analysis examines whether ProteinZero$\_{\text{GRPO}}$ designs are scored as plausible by a protein language model that was not used in our RL pipeline. ESM-2 [11] is an independently trained protein language model; designs that "look like" natural proteins to ESM-2 receive lower pseudo-perplexity (higher pseudo log-likelihood). Pseudo-perplexity is computed via the standard masked-language-model procedure [12] using ESM-2 650M. These results are reported in Appendix C.8 (Table 13 and Figure 14) of the revised manuscript.
>
>     | Metric | InstructPLM (0–150) | ProteinZero$\_{\text{GRPO}}$ (0–150) | $p$ (0–150) | InstructPLM (150–300) | ProteinZero$\_{\text{GRPO}}$ (150–300) | $p$ (150–300) |
>     |---|---|---|---|---|---|---|
>     | Pseudo-perplexity $\downarrow$ | $11.54 \pm 3.80$ | $10.27 \pm 3.64$ | $<0.001$ | $8.48 \pm 3.62$ | $8.07 \pm 3.69$ | $<0.001$ |
>     | Pseudo log-likelihood $\uparrow$ | $-174.76 \pm 43.79$ | $-166.75 \pm 43.62$ | $<0.001$ | $-371.18 \pm 132.05$ | $-362.73 \pm 137.46$ | $<0.001$ |
>
>     ProteinZero$\_{\text{GRPO}}$ sequences receive lower pseudo-perplexity than the InstructPLM base in both length ranges, computationally suggesting that the designs continue to lie in regions of sequence space that an independent language model assigns higher likelihood to.
>
>
>
> - **Biophysical descriptors.** This analysis examines whether basic biophysical qualities (hydrophobicity, charge balance, intrinsic disorder, and predicted solubility) remain in a comparable range after fine-tuning. None of these descriptors is part of the training reward. GRAVY, isoelectric point, molecular weight, and charge/aromatic-residue fractions are computed via Biopython [4] `ProtParam`; predicted scaled solubility (0–1) is obtained from Protein-Sol [9]; intrinsic disorder is predicted with metapredict [10]. These results are reported in Appendix C.9 (Table 14 and Figure 15) of the revised manuscript.
>
>     | Metric | InstructPLM (0–150) | ProteinZero$\_{\text{GRPO}}$ (0–150) | $p$ (0–150) | InstructPLM (150–300) | ProteinZero$\_{\text{GRPO}}$ (150–300) | $p$ (150–300) |
>     |---|---|---|---|---|---|---|
>     | Predicted Scaled Solubility $\uparrow$ | $0.742 \pm 0.088$ | $0.744 \pm 0.090$ | $0.472$ | $0.631 \pm 0.131$ | $0.631 \pm 0.133$ | $0.975$ |
>     | GRAVY $\approx$ | $-0.413 \pm 0.418$ | $-0.458 \pm 0.456$ | $<0.001$ | $-0.263 \pm 0.291$ | $-0.288 \pm 0.289$ | $<0.001$ |
>     | pI $\approx$ | $7.04 \pm 1.79$ | $7.40 \pm 1.88$ | $<0.001$ | $6.48 \pm 1.35$ | $6.59 \pm 1.40$ | $<0.001$ |
>     | Molecular Weight $\approx$ | $8440 \pm 1880$ | $8470 \pm 1900$ | $<0.001$ | $20800 \pm 6260$ | $20900 \pm 6290$ | $<0.001$ |
>     | Positive Charge Fraction $\approx$ | $0.162 \pm 0.049$ | $0.174 \pm 0.055$ | $<0.001$ | $0.140 \pm 0.040$ | $0.143 \pm 0.040$ | $<0.001$ |
>     | Negative Charge Fraction $\approx$ | $0.142 \pm 0.050$ | $0.141 \pm 0.052$ | $0.116$ | $0.132 \pm 0.032$ | $0.133 \pm 0.032$ | $0.682$ |
>     | Aromatic Fraction $\approx$ | $0.076 \pm 0.034$ | $0.077 \pm 0.036$ | $0.593$ | $0.087 \pm 0.025$ | $0.088 \pm 0.025$ | $0.016$ |
>     | Mean Disorder Score $\downarrow$ | $0.218 \pm 0.104$ | $0.215 \pm 0.109$ | $0.357$ | $0.120 \pm 0.060$ | $0.113 \pm 0.055$ | $0.001$ |
>     | Disordered Fraction $\downarrow$ | $0.101 \pm 0.131$ | $0.097 \pm 0.137$ | $0.203$ | $0.034 \pm 0.056$ | $0.027 \pm 0.052$ | $<0.001$ |
>
>     The directional disorder metrics (Mean Disorder Score and Disordered Fraction) improve significantly in the 150–300 length range but not in 0–150 (Mean Disorder Score: $0.218 \to 0.215$, $p = 0.357$ in 0–150; $0.120 \to 0.113$, $p = 0.001$ in 150–300. Disordered Fraction: $0.101 \to 0.097$, $p = 0.203$ in 0–150; $0.034 \to 0.027$, $p < 0.001$ in 150–300). Several other descriptors (predicted scaled solubility, aromatic fraction, negative-charge fraction) show no significant change relative to the base model; we report these as explicit non-improvements rather than presenting uniform gains. The $\approx$-oriented descriptors (GRAVY, pI, molecular weight, positive-charge fraction) shift in mean by small amounts and remain in a range comparable to the base model. Computationally, the overall pattern is consistent with these basic biophysical qualities being broadly preserved through fine-tuning, with modest improvements on the disorder axis at longer length and no signs of systematic degradation.

---

> ### Author Response · Authors · 2026-05-17
> **Response to Reviewer vme7 [Part 6]**
>
> - **Sequence-database identity profile** (UniRef90 via MMseqs2). This analysis examines whether fine-tuning shifts the sequences toward closer copies of natural proteins (the memorization concern). We search each designed sequence against UniRef90 [13] with MMseqs2 [14] under deliberately homolog-favoring settings (max sensitivity `-s 7.5`, permissive E-value `-e 10`); sequences in the $<30\\%$ bucket therefore remain remote/novel [15] even under a search configuration biased toward finding matches. For each backbone, we compute the percentage of designs falling in each identity bucket ($>70\\%$ / $30\\%$–$70\\%$ / $<30\\%$) and test for a per-backbone bucket-distribution shift. These results are reported in Appendix C.10 (Table 15 and Figure 16) of the revised manuscript.
>
>     | Identity bucket | InstructPLM (0–150) | ProteinZero$\_{\text{GRPO}}$ (0–150) | $p$ (0–150) | InstructPLM (150–300) | ProteinZero$\_{\text{GRPO}}$ (150–300) | $p$ (150–300) |
>     |---|---|---|---|---|---|---|
>     | $>70\\%$ identity | $19.44 \pm 29.81$ | $17.19 \pm 27.31$ | $0.554$ | $32.41 \pm 38.76$ | $31.00 \pm 38.77$ | $0.290$ |
>     | $30\\%$ – $70\\%$ identity | $60.24 \pm 34.89$ | $58.68 \pm 31.93$ | $0.668$ | $67.34 \pm 38.84$ | $68.59 \pm 38.71$ | $0.283$ |
>     | $<30\\%$ identity | $20.31 \pm 32.00$ | $24.13 \pm 30.93$ | $0.447$ | $0.25 \pm 1.92$ | $0.41 \pm 3.76$ | $0.706$ |
>
>     The per-backbone bucket distribution of ProteinZero$\_{\text{GRPO}}$ is not significantly different from that of InstructPLM in either length range (all $p > 0.28$); the $>70\\%$ identity bucket, which is directly relevant to the memorization concern, is not increased, and the $<30\\%$ bucket is preserved. We report this as a non-finding, computationally suggesting that RL fine-tuning under our setup does not shift the model toward memorizing natural sequences and does not narrow the explored sequence-novelty range.
>
>
>
> Together, these analyses are intended to substantiate the calibrated claims with concrete cross-oracle measurements rather than to extend them. Experimental validation remains necessary to establish any functional consequence.
>
>
>
>
> > ### Terminology precision for stability metrics (Change 2)
>
>
> We have adopted the reviewer's suggested terminology and now consistently describe stability improvements as *predicted* stability improvements throughout the manuscript. The point-by-point edits are listed below. (These edits have all been applied in the revised manuscript.)
>
>
> | Location | Original | Revised |
> |---|---|---|
> | Abstract | "designability, stability, recovery" | "designability, **predicted** stability, recovery" |
> | Introduction, contribution 5 | "structural accuracy, stability, and diversity" | "structural accuracy, **predicted** stability, and diversity" |
> | Method 3.1.2 | "substantial thermodynamic stability improvements" | "substantial **predicted** thermodynamic stability improvements" |
> | "Comparison with SOTA Inverse Folding Models" | "stability (FoldX ddG: ...)" "stability improves by 21%" | "**predicted** stability (FoldX ddG: ...)" "**predicted** stability improves by 21%" |
> | "Effectiveness of fast-ddg reward" | "gains in thermo-stability" "trade stability for other properties" | "gains in **predicted** thermo-stability" "trade **predicted** stability for other properties" |
> | Case Study, "Stabilization..." paragraph | "high structural accuracy and thermodynamic stability" | "high **predicted** structural accuracy and **predicted** thermodynamic stability" |
> | Case Study, "Performance Scaling" | "improved stability (FoldX ddG: ...)" "stability improves from –20.878 to –24.924" "stability from –27.145 to –32.805" | "improved **predicted** stability (FoldX ddG: ...)" "**predicted** stability improves from ..." "**predicted** stability from ..." |
> | Conclusion | "structural accuracy, stability, recovery" | "structural accuracy, **predicted** stability, recovery" |
> | Appendix, "Complete Performance Metrics" | "substantial stability improvements" "**validating** that our framework learns **generalizable** thermodynamic **principles**" | "substantial **predicted** stability improvements" "**suggesting** that our framework learns thermodynamic **signal that transfers across computational oracles**" |
>
>
>
> These edits make the distinction between computationally predicted and experimentally measured stability explicit at every occurrence.

---

> ### Author Response · Authors · 2026-05-17
> **Response to Reviewer vme7 [Part 7]**
>
> > ### Scope and limitations of Fast-ddG (Change 3)
>
> We agree that the scope and limitations of Fast-ddG should be stated explicitly, and we have revised the manuscript in two places. (These edits have all been applied in the revised manuscript.)
>
> | Location | Original | Revised |
> |---|---|---|
> | Main text, Section 4.5 ("Fast-ddG Accuracy..."), final paragraph | "**establish** that Fast-ddG ... achieves accuracy comparable to physics-based benchmarks" | "**indicate** that Fast-ddG achieves accuracy comparable to physics-based benchmarks ***on single-point mutation data***", **followed by an added qualification that this experimental validation is moderate in strength (PCC $\approx$ 0.60–0.62) and restricted to single-mutation transitions, whereas ProteinZero applies Fast-ddG to full-sequence redesign; Fast-ddG should therefore be regarded as a training proxy rather than a calibrated predictor of absolute folding stability for redesigned sequences** |
> | Appendix (Limitations), "Reliance on Computational Proxies" — new sentence added | (no prior text; new sentence) | A sentence was added stating that **Fast-ddG's experimental validation is moderate (PCC $\approx$ 0.60–0.62 on Ssym) and based solely on single-point mutations, while it is deployed as a reward for full-sequence redesign, and that it is therefore treated as a training proxy rather than a validated stand-in for experimental $\Delta\Delta G$ measurement on redesigned sequences** |
>
>
>
> Together, these additions explain that Fast-ddG is a training proxy whose experimental validation is moderate and restricted to single-mutation data, distinct from the full-sequence redesign setting in which it is used.
>
>
>
>
> > ### Clarifying the statistical reporting (Change 4)
>
> We thank the reviewer for raising this. To make it clearer, we have added a dedicated description to the Evaluation Metrics section (Appendix B.3) that brings the relevant details together in one place. We have also revised the main table so that the standard errors on success rate are presented consistently as percentages (e.g. $0.02\%$, $0.03\%$), which we agree is clearer than the original notation.
>
> - **(i) The 10 runs.** These are 10 independent training runs that differ in random seed. Within each run, we generate 64 sequences for each backbone in the held-out test set (at temperature $0.15$, top-$p$ $0.9$; see (ii) below) to average out the residual stochasticity at this low-temperature, high-confidence setting; the oracle scores are then averaged across all backbones and the 64 sequences per backbone to yield a single per-run scalar per metric. The reported mean and standard error in Table 1 are then computed across the 10 per-run scalars, not across sequences or backbones. Seed-specific and seed-averaged trajectories for both training reward and held-out test-set success rate, with $\pm 1$ standard deviation across the 10 seeds, are shown in Appendix C.3 (Figure 9 of the revised manuscript). The small magnitudes of these standard errors (e.g. $0.02$ percentage points on Success Rate for 0–150 residues and 150–300 residues) reflect high run-to-run reproducibility of the training behavior across seeds rather than narrow within-run variability.
>
> - **(ii) Sampling protocol.** We generate 64 candidate sequences per backbone for both training rollouts and test-set evaluation, following the sampling strategy of the original GRPO paper [16]. The sampling temperatures are those recommended by the InstructPLM base model [2]: the recommended exploration setting (temperature $0.8$, top-$p$ $0.9$) for training rollouts, and the recommended evaluation setting (temperature $0.15$, top-$p$ $0.9$) for test-set evaluation. The evaluation setting uses a low temperature that favors the model's highest-confidence predictions, consistent with the InstructPLM evaluation protocol. Further implementation and sampling details are in Appendix B.2.

---

> ### Author Response · Authors · 2026-05-17
> **Response to Reviewer vme7 [Part 8]**
>
> > ### Expanding the Broader Impact discussion (Change 5)
>
> We have added a dedicated "Dual-use considerations" paragraph to the Broader Impact section (Appendix A.1), preceding the existing ethics discussion. The paragraph:
>
> - explicitly acknowledges that, like other advances in computational protein design, ProteinZero is dual-use, including the potential application to harmful proteins such as toxins or pathogen-related components, and that we do not consider this risk negligible;
>
> - notes properties of the method that limit the concern (optimization over predetermined backbones rather than generation of novel folds or functions; rewards that are not function- or virulence-specific); and
>
> - addresses the four points raised by the reviewer:
>   - (i) responsible release, with code and checkpoints released under an intended-use license that permits academic research, reproduction, and benchmarking while prohibiting use intended to design harmful proteins;
>   - (ii) sequence screening, encouraging users to screen designed sequences against established biosecurity tools (e.g., synthesis-provider screening, the IBBIS Common Mechanism) prior to synthesis;
>   - (iii) intended-use limits, framed through the license above; and
>   - (iv) the requirement that no designed sequence proceed to wet-lab synthesis or deployment without prior experimental characterization and institutional biosafety and biosecurity review.
>
>
> ---
>
> **References**
>
> [1] F. Wilcoxon. Individual comparisons by ranking methods. *Biometrics Bulletin* 1, 80–83, 1945.
>
> [2] J. Qiu et al. InstructPLM: Aligning Protein Language Models to Follow Protein Structure Instructions. *bioRxiv* 2024.04.17.589642, 2024.
>
> [3] Abramson, J., Adler, J., Dunger, J. et al. Accurate structure prediction of biomolecular interactions with AlphaFold 3. Nature 630, 493–500 (2024).
>
> [4] P. J. A. Cock et al. Biopython: freely available Python tools for computational molecular biology and bioinformatics. *Bioinformatics* 25, 1422–1423, 2009.
>
> [5] M. L. Hekkelman, D. Á. Salmoral, A. Perrakis, and R. P. Joosten. DSSP 4: FAIR annotation of protein secondary structure. Protein Science 34(8), e70208, 2025.
>
> [6] W. Kabsch and C. Sander. Dictionary of protein secondary structure: pattern recognition of hydrogen-bonded and geometrical features. *Biopolymers* 22, 2577–2637, 1983.
>
> [7] A. Shrake and J. A. Rupley. Environment and exposure to solvent of protein atoms. Lysozyme and insulin. *Journal of Molecular Biology* 79, 351–371, 1973.
>
> [8] S. Henikoff and J. G. Henikoff. Amino acid substitution matrices from protein blocks. *PNAS* 89, 10915–10919, 1992.
>
> [9] M. Hebditch et al. Protein–Sol: a web tool for predicting protein solubility from sequence. *Bioinformatics* 33, 3098–3100, 2017.
>
> [10] R. J. Emenecker, D. Griffith, and A. S. Holehouse. Metapredict: a fast, accurate, and easy-to-use predictor of consensus disorder and structure. *Biophysical Journal* 120, 4312–4319, 2021.
>
> [11] Z. Lin et al. Evolutionary-scale prediction of atomic-level protein structure with a language model. *Science* 379, 1123–1130, 2023.
>
> [12] J. Meier et al. Language models enable zero-shot prediction of the effects of mutations on protein function. *NeurIPS* 34, 2021.
>
> [13] B. E. Suzek et al. UniRef clusters: a comprehensive and scalable alternative for improving sequence similarity searches. *Bioinformatics* 31, 926–932, 2015.
>
> [14] M. Steinegger and J. Söding. MMseqs2 enables sensitive protein sequence searching for the analysis of massive data sets. *Nature Biotechnology* 35, 1026–1028, 2017.
>
> [15] B. Rost. Twilight zone of protein sequence alignments. *Protein Engineering* 12, 85–94, 1999.
>
> [16] Z. Shao et al. DeepSeekMath: Pushing the Limits of Mathematical Reasoning in Open Language Models. *arXiv:2402.03300*, 2024.